# Fractional relaxation noises, motions and the fractional energy balance equation

Shaun Lovejoy

Physics, McGill University,

3600 University st.

Montreal, Que. H3A 2T8

Canada

## Abstract:

We consider the statistical properties of solutions of the stochastic fractional relaxation equation and its fractionally integrated extensions that are models for the Earth's energy balance. In these equations, the highest order derivative term is fractional and it models the energy storage processes that are scaling over a wide range. When driven stochastically, the system is a Fractional Langevin Equation (FLE) that has been considered in the context of random walks where it yields highly nonstationary behaviour. An important difference with the usual applications is that I instead consider the stationary solutions of the Weyl fractional relaxation equations whose domain is $-\infty$ to $t$ rather than $0$ to $t$.

An additional key difference is that unlike the (usual) FLEs - where the highest order term is of integer order and the fractional term represents a scaling damping - in the fractional relaxation equation, the fractional term is of highest order. When its order is less than ½ (this is the main empirically relevant range), the solutions are noises (generalized functions) whose high frequency limits are fractional Gaussian noises (fGn). In order to yield physical processes, they must be smoothed and this is conveniently done by considering their integrals. Whereas the basic processes are (stationary) fractional relaxation noises (fRn), their integrals are (nonstationary) fractional Relaxation motions (fRm) that generalize both fractional Brownian motion, (fBm) as well as Ornstein-Uhlenbeck processes.

Since these processes are Gaussian, their properties are determined by their second order statistics; using Fourier and Laplace techniques, we analytically develop corresponding power series expansions for fRn, fRm and their fractionally integrated extensions needed to model energy storage processes. We show extensive analytic and numerical results on the autocorrelation functions, Haar fluctuations and spectra. We display sample realizations.

Finally, we discuss the predictability of these processes which – due to long memories - is a *past* value problem, not an *initial* value problem (that is used for example in highly skillful monthly and seasonal temperature forecasts). We develop an analytic formula for the fRn forecast skills and compare it to fGn skill. The large scale white noise and fGn limits are attained in a slow power law manner so that when the temporal resolution of the series is small compared to the relaxation time (of the order of a few years in the Earth), fRn and its extensions can mimic a long memory process with a range of

exponents wider than possible with fGn or fBm. We discuss the implications for monthly,
seasonal, annual forecasts of the Earth's temperature as well as for projecting the
temperature to 2050 and 2100.

## 1. Introduction:

Over the last decades, stochastic approaches have rapidly developed and have spread
throughout the geosciences. From early beginnings in hydrology and turbulence,
stochasticity has made inroads in many traditionally deterministic areas. This is notably
illustrated by stochastic parametrisations of Numerical Weather Prediction models, e.g.
[*Buizza et al.*, 1999], and the "random" extensions of dynamical systems theory, e.g.
[*Chekroun et al.*, 2010].
In parallel, pure stochastic approaches have developed primarily along two distinct
lines. One is the classical (integer ordered) stochastic differential equation approach based
on the Itô or Stratonivch calculii that goes back to the 1950's (see the useful review
[*Dijkstra*, 2013]). The other is the scaling strand that encompasses both linear (monofractal,
[*Mandelbrot*, 1982]) and nonlinear (multifractal) models (see the review [*Lovejoy and*
*Schertzer*, 2013]) that are based on phenomenological scaling models, notably cascade
processes. These and other stochastic approaches have played important roles in nonlinear
Geoscience.
Up until now, the scaling and differential equation strands of stochasticity have had
surprisingly little overlap. This is at least partly for technical reasons: integer ordered
stochastic differential equations have exponential Green's functions that are incompatible
with wide range scaling. However, this shortcoming can – at least in principle - be easily
overcome by introducing at least some derivatives of fractional order. Once the (typically)
ad hoc restriction to integer orders is dropped, the Green's functions are based on
"generalized exponentials" that are in turn are based on fractional powers (see the review
[*Podlubny*, 1999]). The integer-ordered stochastic equations that have received most
attention are thus the exceptional, nonscaling special cases. In physics they correspond to
classical Langevin equations; in geophysics and climate modelling, they correspond to the
Linear Inverse Modelling (LIM) approach that goes back to [*Hasselmann*, 1976] later
elaborated notably by [*Penland and Magorian*, 1993], [*Penland*, 1996], [*Sardeshmukh et*
*al.*, 2000], [*Sardeshmukh and Sura*, 2009] and [*Newman*, 2013]. Although LIM is not the
only stochastic approach to climate, in two recent representative multi-author collections
([*Palmer and Williams*, 2010] and [*Franzke and O'Kane*, 2017]), all 32 papers shared the
integer ordered assumption (a single exception being [*Watkins*, 2017], see also [*Watkins*
*et al.*, 2020]).
Under the title "Fractal operators" [*West et al.*, 2003], reviews and emphasizes that
in order to yield scaling behaviours, it suffices that stochastic differential equations contain
fractional derivatives. However, when it is the time derivatives of stochastic variables that
are fractional - fractional Langevin equations (FLE) - then the relevant processes are
generally non-Markovian [*Jumarie*, 1993], so that there is no Fokker-Planck (FP) equation
describing the corresponding probabilities. Even in the relatively few cases where the FLE
has been studied, the fractional terms are generally models of viscous damping so that the
highest order terms are still integer ordered (an exception is [*Watkins et al.*, 2020] who
mentions "fractionally integrated FLE" of the type studied here but without investigating
its properties). Integer ordered terms have the convenient consequence of regularizing the
solutions so that they are at least root mean square continuous; in this paper the highest
order derivatives are fractional so that when the highest order terms are ≤1/2, the solutions
are "noises" i.e. generalized functions that must be smoothed in order to represent
physically meaningful quantities.
An additional obstacle is that - as with the simplest scaling stochastic model –
fractional Brownian motion (fBm, [*Mandelbrot and Van Ness*, 1968]) - we expect that the
solutions will not be semi-martingales and hence that the Itô calculus used for integer
ordered equations will not be applicable (see [*Biagini et al.*, 2008]). This may explain the
relative paucity of mathematical literature on stochastic fractional equations (see however
[*Karczewska and Lizama*, 2009]). In statistical physics, starting with [*Mainardi and Pironi*,
1996], [*Metzler and Klafter*, 2000], [*Lutz*, 2001] and helped with numerics, the FLE (and
a more general "Generalized Langevin Equation" [*Kou and Sunney Xie*, 2004], [*Watkins
et al.*, 2019]) has received a little more attention as a model for (nonstationary) particle
diffusion (see [*West et al.*, 2003] for an introduction, or [*Vojta et al.*, 2019] for a more
recent example). These technical aspects may explain why the statistics of the resulting
processes are not available in the literature.
Technical difficulties may also explain the apparent paradox of Continuous Time
Random Walks (CTRW) and other approaches to anomalous diffusion that involve
fractional equations. While CTRW probabilities are governed by the deterministic
fractional ordered Generalized Fractional Diffusion equation (e.g. [*Hilfer*, 2000], [*Coffey
et al.*, 2012]), the walks themselves are based on specific particle jump models rather than
(stochastic) Langevin equations. Alternatively, a (spatially) fractional ordered Fokker-
Planck equation may be derived from an integer-ordered but nonlinear Langevin equation
for a diffusing particle driven by an (infinite variance) Levy motion [*Schertzer et al.*, 2001].
In nonlinear geoscience, it is all too common for mathematical models and techniques
developed primarily for mathematical reasons, to be subsequently applied to the real world.
This approach - effectively starting with a solution and then looking for a problem -
occasionally succeeds, yet historically the converse has generally proved more fruitful.
The proposal that an understanding of the Earth's energy balance requires the Fractional
Energy Balance Equation (FEBE, [*Lovejoy et al.*, 2021], announced in [*Lovejoy*, 2019a])
is an example of the latter. First, the scaling exponent of macroweather (monthly, seasonal,
interannual) temperature stochastic variability was determined ($H_I \approx$ -0.085±0.02) and
shown to permit skillful global temperature predictions, [*Lovejoy*, 2015b], [*Lovejoy et al.*,
2015], [*Del Rio Amador and Lovejoy*, 2019], and then it was extended to regional
temperatures (at 2°x2° resolution) [*Del Rio Amador and Lovejoy*, 2019; *Del Rio Amador
and Lovejoy*, 2021a; *Del Rio Amador and Lovejoy*, 2021b]. The latter papers showed how
the long memory high frequency approximation to the FEBE can not only make state of
the art multi-month temperature forecasts, but the corresponding simulations generate
emergent properties such as realistic El Nino events.
In parallel, the multidecadal deterministic response to external (anthropogenic,
deterministic) forcing was shown to also obey a scaling law but with a different exponent
[*Hebert*, 2017], [*Lovejoy et al.*, 2017], [*Procyk et al.*, 2020], [*Procyk*, 2021; *Procyk et al.*,
2022], ($H_F \approx$ -0.5±0.2). It was only then was realized that the order $h$ FEBE naturally
accounts for both the high and low frequency global temperature exponents with $h = H_I +$
1/2 and $H_F = -h$ with both empirical exponents recovered with a FEBE of order $h \approx$
0.38±0.03. The realization that the FEBE fit these basic empirical facts motivated the
present research into its statistical properties including its predictability.
In the EBE, energy storage is modelled by a uniform slab of material implying that
when perturbed, the temperature exponentially relaxes to a new thermodynamic
equilibrium. However, as reviewed in [*Lovejoy and Schertzer*, 2013]), both conventional
Global Circulation Models and observations show that atmospheric, oceanic and surface
(e.g. topographic) structures are spatially scaling. A consequence is that the temperature
relaxes to equilibrium in a power law manner. This motivated earlier approaches ([*van
Hateren*, 2013], [*Rypdal*, 2012], [*Hebert*, 2017], [*Lovejoy et al.*, 2017]) to postulate that
the climate response function (CRF) itself is scaling. However, these models require either
ad hoc truncations or imply infinite sensitivity to small perturbations [*Rypdal*, 2015],
[*Hébert and Lovejoy*, 2015].
The FEBE instead situates the scaling in the energy storage processes; this is the
physical basis for the phenomenological derivation of the FEBE proposed in [*Lovejoy et
al.*, 2021] and the zeroth order term determines guarantees that equilibrium is reached after
long enough times. The scaling of the basic physical quantities in both time and space
motivates the study of the FEBE and its fractionally integrated extensions discussed below
temperature treated as a stochastic variable. The FEBE determines the Earth's global
temperature when the energy storage processes are scaling and modelled by a fractional
time derivative term. Recently, analysis of the atmospheric radiation budget has shown
that at least over some regions, the internal component of the radiative forcing may itself
be scaling, this justifies the consideration of the extensions to fGn forcing.
The FEBE differs from the classical energy balance equation (EBE) in several ways.
Whereas the EBE is integer ordered and describes the deterministic, exponential relaxation
of the Earth's temperature to equilibrium, the FEBE is of fractional order and because it is
both deterministic and stochastic it unites all the forcings and responses into a single model.
Whereas the former represents the forcing and response to the unresolved degrees of
freedom - the "internal variability" - and is treated as a zero mean Gaussian noise, the latter
represents the external (e.g. anthropogenic) forcing and the forced response modelled by
the (deterministic) total external forcing. Complementary work [*Procyk et al.*, 2020],
[*Procyk*, 2021; *Procyk et al.*, 2022] uses the deterministic FEBE as the basic model for the
response to external forcing, but its Bayesian parameter estimation uses the stochastic
FEBE to characterize the likelihood function of the residuals assumed to be the responses
to stochastic internal forcing and governed by the same equation. It thus avoids the ad hoc
error models involved in conventional Bayesian parameter estimation. The result is a
parsimonious, FEBE projection of the Earth's temperature to 2100 that has much lower
uncertainty than the classical Global Circulation Model alternative. This is the first time
that classical General Circulation Model climate projections have been confirmed by an
independent, qualitatively different, approach.
An important but subtle EBE - FEBE difference is that whereas the former is an
*initial* value problem whose initial condition is the Earth's temperature at $t = 0$, the FEBE
is effectively a *past* value problem whose prediction skill improves with the amount of
available past data and - depending on the parameters - it can have an enormous memory
[*Del Rio Amador and Lovejoy*, 2021b]. To understand this, recall that an important aspect
of fractional derivatives is that they are defined as convolutions over various domains. To
date, the main one that has been applied to physical problems is the Riemann-Liouville
(and the related Caputo) fractional derivative specialized to convolutions over the interval
between an initial time = 0 and a later time $t$. With one or two exceptions, this is the domain
considered in Podlubny's mathematical monograph on deterministic fractional differential
equations [*Podlubny*, 1999] as well as in the stochastic fractional physics discussed in
[*West et al.*, 2003], [*Herrmann*, 2011], [*Atanackovic et al.*, 2014], and most of the papers
in [*Hilfer*, 2000] (with the partial exceptions of [*Schiessel et al.*, 2000], and [*Nonnenmacher*
*and Metzler*, 2000]). A key point of the FEBE is that it is instead based over semi-infinite
domains - here from $-\infty$ to $t$ - often called "Weyl" fractional derivatives. This is the
natural range to consider for the Earth's energy balance and it is needed to obtain
statistically stationary responses. Random walk problems involve fractional equations over
the domain 0 to $t$ can be dealt with using Laplace transform techniques. In comparison the
Earth's temperature balance involves statistically stationary stochastic forcings that are
more conveniently dealt with using Fourier techniques.
We have mentioned that the FEBE can be derived phenomenologically where the
fractional derivative of order $h$ term representing the energy storage processes [*Lovejoy et*
*al.*, 2021]. In this approach the order $h$ is an empirically determined parameter with $h = 1$
corresponding to the classical (exponential) exception. Alternatively it may derived from
a more fundamental starting point, the classical heat equation – the same starting point as
the classical Budyko-Sellers energy balance models ([*Budyko*, 1969], [*Sellers*, 1969]).
Recently it was shown that with the help of Babenko's operator method that the special $h$
= 1/2 FEBE - the Half-ordered Energy Balance Equation (HEBE) - could be derived
analytically from the classical heat equation [*Lovejoy*, 2021a; b].
To obtain the HEBE, it is sufficient to follow the Budyko-Sellers approach, but to
avoid one of their key approximations. The Earth's atmosphere and ocean are driven by
local imbalances in radiative fluxes. While Budyko-Sellers models simply redirect this
flux away from the equator, the HEBE improvement ([*Lovejoy*, 2021a; b]) is to instead use
the mathematically correct radiative-conductive surface boundary conditions. When this
is done in the classical energy transport equation, one obtains an important $h = 1/2$ special
case of the FEBE, the Half-order EBE or HEBE. The use of half-order derivatives in the
heat equation is completely classical and goes back to at least [*Oldham*, 1973; *Oldham and*
*Spanier*, 1972], [*Babenko*, 1986], [*Magin et al.*, 2004] [*Sierociuk et al.*, 2013]. The
extension to $h \neq 1/2$ can be obtained using the same mathematical techniques by starting
with the fractional generalization of the classical heat equation, the fractional heat equation.
Further generalizations are also possible and will be reported elsewhere.
The choice of a Gaussian white noise forcing was made not so much for its theoretical
simplicity but for its physical realism. Using scaling to divide atmospheric dynamics into
dynamical ranges ([*Lovejoy*, 2013], [*Lovejoy*, 2015a], [*Lovejoy*, 2019b]), the main ones are
weather, macroweather and climate. While the temperature variability in both space and
in time is generally highly intermittent (multifractal), there is one exception: the temporal
macroweather regime (starting at the lifetime of planetary structures - roughly ten days –
up until the climate regime at much longer scales). Macroweather is the regime over which
the FEBE applies and it has exceptionally low intermittency: temporal (but not spatial)
temperature anomalies are not far from Gaussian ([*Lovejoy*, 2018]). Responses to
multifractal or Levy process FEBE forcings may however be of interest elsewhere.
This paper is structured as follows. In section 2 we present the fractional relaxation
equation, forced by a Gaussian white noise as a natural generalization of classical fractional
Brownian motion, fractional Gaussian noise and Ornstein-Uhlenbeck processes (sections
2.1, 2.2). When forced by Gaussian white noises, the solutions define the corresponding
fractional Relaxation motions (fRm) and fractional Relaxation noises (fRn). We consider
further extensions to the case where the equation is forced by a scaling noise fGn (section
2.3, eqs. 21, 22). This is equivalent to considering the fractionally integrated fractional
relaxation equation with white noise forcing. In section 2, we first solve the equations in
terms of Green's functions, and then introduce powerful Fourier techniques that yield
integral representations of the second order statistics including autocorrelations, structure
functions (eqs. 33, 35), Haar fluctuations and spectra (with many details in appendix A, in
appendix B, we derive the properties of the HEBE special case). In section 3, we develop
both short and long time (asymptotic) series expansions for the statistics (eqs. 49, 51) and
we display and discuss sample fRn, fRm processes. In section 4 we discuss the problem
of prediction – important for macroweather forecasting – and derive expressions for the
optimum predictor (eq. 63) and its theoretical prediction skill as a function of forecast lead
time (eq. 68). In section 5 we conclude.
I could note that the paper is somewhat complex due to the necessity of developing
several approaches: Fourier for the main integral representations (section 2), Laplace for
the asymptotic expansions (section 3), and real space for the predictability results (section
242 4).

## 2. The fractional relaxation equation

### 2.1 fRn, fRm, fGn and fBm

In the introduction, we outlined physical arguments that the Earth's global energy
balance could be well modelled by the fractional energy balance equation. Taking $T$ as the
globally averaged temperature, $\tau$ as the characteristic time scale for energy
storage/relaxation processes, $F$ as the (stochastic) forcing (energy flux; power per area),
and $s$ the climate sensitivity (temperature increase per unit flux of forcing) the FEBE can
be written in Langevin form as:

$$\tau^h\left( {}_aD_t^hT\right)+T=sF \qquad , \tag{1}$$

where the Riemann-Liouville fractional derivative symbol ${}_aD_t^h$ is defined as:

$$_aD_t^hT=\frac{1}{\Gamma\left(1-h\right)}\frac{d}{dt}\int_a^t\left(t-s\right)^{-h}T\left(s\right)ds;\quad 0<h<1 \quad, \tag{2}$$

Where $\Gamma$ is the standard gamma function. Derivatives of order $\nu>1$ can be obtained using
$\nu = h+m$ where $m$ is the integer part of $\nu$, and then applying this formula to the $m^{\text{th}}$ ordinary
derivative. The main case studied in applications (e.g. random walks) is $a = 0$ so that
Laplace transform techniques are often used (alternatively, the somewhat different Caputo
fractional derivative is used). However, here we will be interested in $a=-\infty$ : the Weyl
fractional derivative $_{-\infty}D_t^h$ which is naturally handled by Fourier techniques (section 2.4
and appendices A, B), and in this case, this distinction is unimportant.
Since equation 1 is linear, by taking ensemble averages, it can be decomposed into
deterministic and random components with the former driven by the mean forcing external
to system $<F>$, and the latter by the fluctuating stochastic component $F - <F>$ representing
the internal forcing driving the internal variability.  The deterministic part has been used to
project the Earth's temperature throughout the 21$^{st}$ century ([*Procyk et al.*, 2020], [*Procyk*
*et al.*, 2022]); in the following we consider the simplest purely stochastic model in which
$<F> = 0$ and $F = \gamma$ where $\gamma$ is a Gaussian "delta correlated" and unit amplitude white noise:
$$\langle \gamma(v) \rangle = 0; \quad \langle \gamma(v)\gamma(u) \rangle = \delta(u-v) \ .$$  (3)
In [*Hebert*, 2017], [*Lovejoy et al.*, 2017], [*Hébert et al.*, 2021] it was argued on the
basis of an empirical study of ocean- atmosphere coupling that $\tau_r \approx 2$ years while recent
work indicates a value somewhat higher, $\approx 5$ years, [*Procyk et al.*, 2022].  At high
frequencies, [*Lovejoy et al.*, 2015] and [*Del Rio Amador and Lovejoy*, 2019], [*Del Rio*
*Amador and Lovejoy*, 2021a] that the value $h \approx 0.4$ reproduced both the Earth's temperature
both at scales $< \tau$ as well as for macroweather scales (longer than the weather regime
scales of about 10 days) but still $< \tau$. [*Procyk et al.*, 2020] also used the FEBE to estimate
(the global) $s = [0.45, 0.67]$ K/(W/m$^2$) (90% confidence interval) and  the amplitude of
the radiative forcing at monthly resolution was: $[0.89; 1.42]$ W/m$^2$ (90% confidence
interval).
When $0 < h < 1$, eq. 1 with $\gamma(t)$ replaced by a deterministic forcing is a fractional
generalization of the usual ($h = 1$) relaxation equation; when $1 < h < 2$, it is the "fractional
oscillation equation", a generalization of the usual ($h = 2$) oscillation equation, [*Podlubny*,
1999].
To simplify the development, we use the relaxation time $\tau$ to nondimensionalize time
i.e. to replace time by $t/\tau$ to obtain the canonical Weyl fractional relaxation equation:
$$\left( {}_{-\infty}D_t^h + 1 \right)U_h = \gamma; \quad Q_h(t) = \int_0^t U_h(v)dv$$  (4)
for the nondimensional process $U_h$. The dimensional solution of eq. 1 with nondimensional
$\gamma = sF$  is simply $T(t) = \tau^{-1} U_h(t/\tau)$ so that in the nondimensional eq. 4, the characteristic
transition "relaxation" time between dominance by the high frequency (differential) and
the low frequency ($U_h$ term) is $t = 1$. Although we give results for the full range $0 < h < 2$
- i.e. both the "relaxation" and "oscillation" ranges – for simplicity, we refer to the solution
$U_h(t)$ as "fractional Relaxation noise" (fRn) and to $Q_h(t)$ as "fractional Relaxation motion"
(fRm).  Note that fRn is only strictly a noise when $h \leq 1/2$.
In dealing with fRn and fRm, we must be careful of various small and large $t$
divergences.  For example, eqs. 1 and 4 are the fractional Langevin equations
corresponding to generalizations of integer ordered stochastic diffusion equations: the
classical $h = 1$ case is the Ohrenstein-Uhlenbeck process.  Since $\gamma(t)$ is a "generalized
function" - a "noise" - it does not converge at a mathematical instant in time, it is only
strictly meaningful under an integral sign.  Therefore, a standard form of eq. 4 is obtained
by integrating both sides by order $h$ (i.e. by differentiating by $-h$ and assuming that
differentiation and integration of order $h$ commute):
$$U_h(t) = -{}_{-\infty}D_t^{-h}U_h + {}_{-\infty}D_t^{-h}\gamma = -\frac{1}{\Gamma(h)}\int_{-\infty}^t (t-v)^{h-1}U_h(v)dv + \frac{1}{\Gamma(h)}\int_{-\infty}^t (t-v)^{h-1}\gamma(v)dv ,$$

(5)

(see e.g. [*Karczewska and Lizama*, 2009]).  The white noise forcing in the above is
statistically stationary; the solution for $U_h(t)$ is also statistically stationary.  It is tempting
to obtain an equation for the motion $Q_h(t)$ by integrating eq. 4 from $-\infty$ to $t$ to obtain the
fractional Langevin equation: $_{-\infty}D_t^h Q_h + Q_h = W$ where $W$ is Wiener process (a standard
Brownian motion) satisfying $dW = \gamma(t)dt$.  Unfortunately the Wiener process integrated
$-\infty$ to $t$ almost surely diverges, hence we relate $Q_h$ to $U_h$ by an integral from 0 to $t$.
In the high frequency limit, the derivative dominates and we obtain the simpler
fractional Langevin equation:
$$_{-\infty}D_t^h F_h = \gamma; \qquad B_h(t) = \int_0^t F_h(v)dv \qquad\qquad (6)$$

Whose solution $F_h$ is the fractional Gaussian noise process (fGn, not to be confused with
the forcing), and whose integral $B_h$ is fractional Brownian motion (fBm).  We thus
anticipate that $F_h$ and $B_h$ are the high frequency limits of fRn, fRm.
**2.2 Green's functions**
Although it will turn out that Fourier techniques are very convenient for calculating
the statistics, there are also advantages to classical (real space) approaches and in any case
they are needed for studying the predictability properties (section 4).  We therefore start
with a discussion of Green's functions that are the classical tools for solving
inhomogeneous linear differential equations:
$$F_h(t) = \int_{-\infty}^t G_{0,h}^{(fGn)}(t-v)\gamma(v)dv$$
$$U_h(t) = \int_{-\infty}^t G_{0,h}^{(fRn)}(t-v)\gamma(v)dv$$
, (7)

where $G_{0,h}^{(fGn)}$ and $G_{0,h}^{(fRn)}$ are Green's functions for the differential operators corresponding
respectively to $_{-\infty}D_t^h$ and $_{-\infty}D_t^h + 1$.  Note that due to causality, all the Green's functions
used in this paper vanish for $t<0$.
$G_{0,h}^{(fGn)}$ and $G_{0,h}^{(fRn)}$ are the usual "impulse" (Dirac) response Green's functions (hence
the subscript "0").  For the differential operator $\Xi$ they satisfy:
$\Xi G_{0,h}(t) = \delta(t)$. (8)
Integrating this equation we find an equation for their integrals $G_{1,h}$ which are thus
"step" (Heaviside, subscript "1") response Green's functions satisfying:
$$\Xi G_{1,h}(t) = \Theta(t); \quad \Theta(t) = \int_{-\infty}^t \delta(v)dv \ ; \qquad \frac{dG_{1,h}}{dt} = G_{0,h}, \qquad (9)$$

where $\Theta$ is the Heaviside (step) function ($= 0$ for $t<0$, $= 1$ for $t \geq 0$).  The inhomogeneous
equation:

$\Xi f(t) = F(t)$ (10)

has a solution in terms of either an impulse or a step Green's function:
$$f(t)=\int_{-\infty}^{t}G_{0,h}(t-v)F(v)dv=\int_{-\infty}^{t}G_{1,h}(t-v)F'(v)dv; \quad F'(v)=\frac{dF}{dv} , \qquad (11)$$

the equivalence being established by integration by parts with the conditions $F(-\infty)=0$
and $G_{1,h}(0)=0$. The use of the step rather than impulse response is standard in the Energy
Balance Equation literature since it gives direct information on energy balance and the
approach to equilibrium (see e.g. [*Lovejoy et al.*, 2021]). The step response for the noise
is also the basic impulse response function for the motion.
For fGn, the Green's functions are simply the kernels of the fractional integrals:
$$F_h(t)=\frac{1}{\Gamma(h)}\int_{-\infty}^{t}(t-v)^{h-1}\gamma(v)dv , \qquad (12)$$

obtained by integrating both sides of eq. 6 by order $h$. We conclude:
$$G_{0,h}^{(fGn)}=\frac{t^{h-1}}{\Gamma(h)}; \quad G_{1,h}^{(fGn)}=\frac{t^{h}}{\Gamma(h+1)}; \quad -\frac{1}{2}\le h<\frac{1}{2} . \qquad (13)$$

For fRn, we now recall some classical results useful in geophysical applications.
First, these Green's functions are often equivalently written in terms of Mittag-Leffler
functions ("generalized exponentials"), $E_{\alpha,\beta}$:
$$G_{0,h}(t)=t^{h-1}E_{h,h}(-t^{h}); \qquad E_{\alpha,\beta}(z)=\sum_{n=0}^{\infty}\frac{z^{n}}{\Gamma(\alpha n+\beta)} \qquad (14)$$

$$G_{0,h}(t)=\sum_{n=1}^{\infty}(-1)^{n+1}\frac{t^{nh-1}}{\Gamma(nh)}; \quad 0<h\le 2$$

(to lighten the notation in eq. 14 and in the following, we suppress the superscripts for fRn,
fRm proceesses). A convenient feature of Mittag-Leffler functions is that they can be
easily integrated by any positive order $\alpha$:
$$G_{\alpha,h}(t)={}_0D_t^{-\alpha}(G_{0,h}(t))= \begin{array}{ll} t^{h-1+\alpha}E_{h,h+\alpha}(-t^{h})=t^{\alpha-1}\sum_{n=1}^{\infty}(-1)^{n+1}\frac{t^{nh}}{\Gamma(\alpha+nh)}; & t\ge 0 \\ \\ 0; & t<0 \end{array}$$

$$\alpha\ge 0; \quad 0\le h\le 2 \qquad (15)$$

([*Podlubny*, 1999]). As mentionned, the constraint $t>0$ is due to causality, physical Green's
functions vanish for negative arguments. In the following this will simply be assumed.
With $\alpha=1$, we obtain the useful formula:
$$G_{1,h}(t)=t^{h}E_{h,h+1}(-t^{h}); \quad G_{1,h}(t)=\sum_{n=1}^{\infty}(-1)^{n+1}\frac{t^{nh}}{\Gamma(1+nh)} \qquad (16)$$

With this, we see that $G_{0,h}^{(fGn)}$ and $G_{1,h}^{(fGn)}$ are simply the first terms in the power series
expansions of the corresponding fRn, fRm Green's functions. The solution to eq. 4 with
the white noise forcing $\gamma(t)$ is therefore:
$$U_{0,h}(t) = \int_{-\infty}^{t} G_{0,h}(t-v)\gamma(v)dv \qquad (17)$$

Where for this "pure" fRn process, we have added the subscript "0" for reasons
discussed below. We note that at the origin, for $0 < h < 1$, $G_{0,h}$ is singular whereas $G_{1,h}$ is
regular so that it is may be advantageous to use the latter (step) response function (for
example in the numerical simulations in section 4). These Green's function responses are
shown in figure 1. When $0 < h \le 1$, the step response is monotonic; in an energy balance
model, this would correspond to relaxation to equilibrium. When $1 < h < 2$, we see that
there is overshoot and oscillations around the long term value; it is therefore (presumably)
outside the physical range of an equilibrium process.
In order to understand the relaxation process – i.e. the approach to the asymptotic
value 1 in fig. 1 for the step response $G_{1,h}$ - we need the asymptotic expansion:
$$G_{\alpha,h}(t) = \sum_{n=0}^{\infty} \frac{(-1)^n}{\Gamma(\alpha - nh)} t^{\alpha-1-nh}; \quad t \gg 1 \quad , \qquad (18)$$

For $\alpha = 0, 1$ we obtain the special cases corresponding to impulse and step responses:
$$G_{0,h}(t) = \sum_{n=0}^{\infty} (-1)^n \frac{t^{-1-nh}}{\Gamma(-nh)}; \quad G_{1,h}(t) = \sum_{n=0}^{\infty} (-1)^n \frac{t^{-nh}}{\Gamma(1-nh)}; \quad t \gg 1 \qquad (19)$$

($0 < h < 1$, $1 < h < 2$; note that the $n = 0$ terms are 0, 1 for $G_{0,h}$, $G_{1,h}$ respectively) [*Podlubny*,
1999], i.e. the asymptotic expansions are power laws in $t^{-h}$ rather than $t^h$. According to this,
the asymptotic approach to the step function response (bottom row in fig. 1) is a slow,
power law process. In the FEBE, this implies for example that the classical $CO_2$ doubling
experiment would yield a power law rather than exponential approach to a new
thermodynamic equilibrium. Comparing this to the EBE, i.e. the special case $h = 1$, we
have:
$$G_{0,1}(t) = e^{-t}; \quad G_{1,1}(t) = 1 - e^{-t} \quad , \qquad (20)$$

so that when $h = 1$, the asymptotic step response is instead approached exponentially fast.
We see that when $h = 1$ the process is a classical Ornstein-Uhlenbeck process so that fRn
can be considered a generalization of the latter. There are also analytic formulae for fRn
when $h = 1/2$ (the HEBE) discussed in appendix B notably involving logarithmic
corrections.

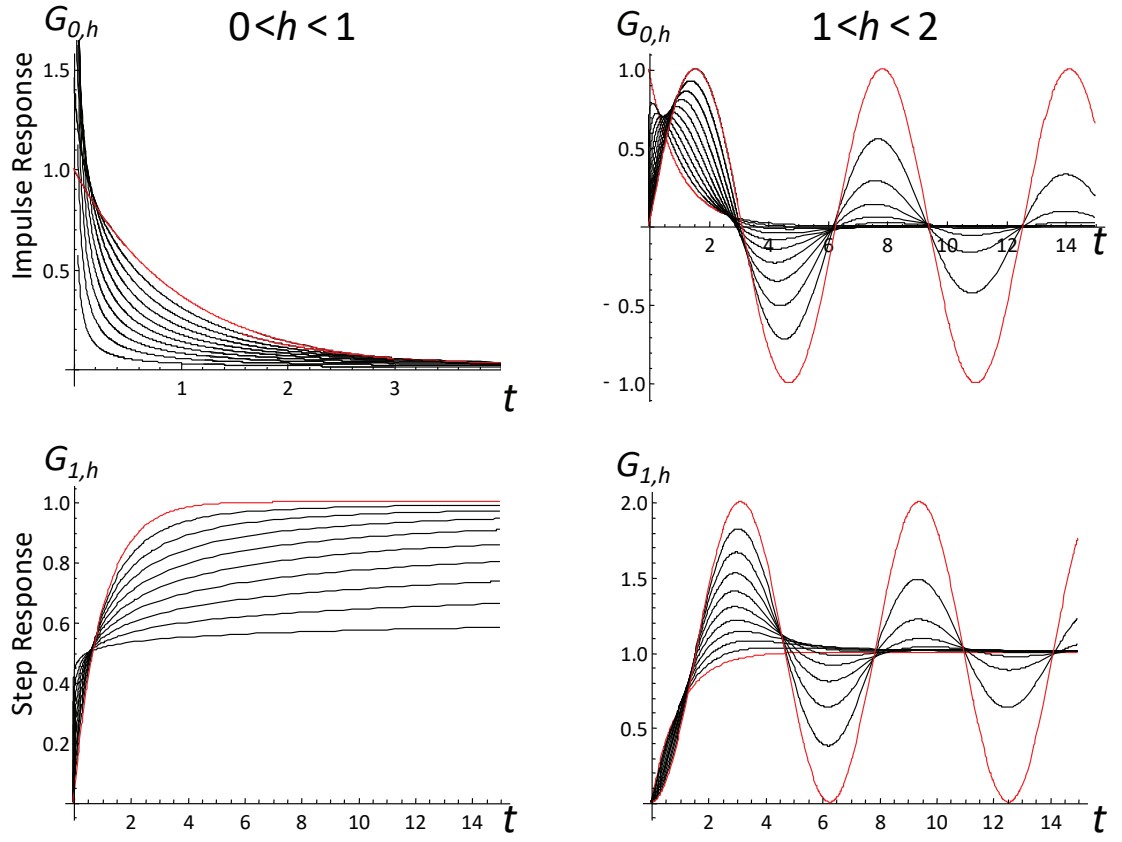

Fig. 1a: The impulse (top) and step response functions (bottom) for the fractional relaxation
range ($0 < h < 1$, left, red is $h = 1$, the exponential), the black curves, bottom to top are for $h = 1/10$,
2/10, ..9/10) and the fractional oscillation range ($1 < h < 2$, red are the integer values $h = 1$, bottom,
the exponential, and top, $h = 2$, the sine function, the black curves, bottom to top are for $h = 11/10$,
12/10, ..19/10.

**2.3 The $\alpha$ order fractionally integrated fRn, fRm processes:**
Before proceeding to discuss the statistics of fRn, fRm processes, it is useful to
make a generalization to the fractionally integrated processes:
$$U_{\alpha,h} = {}_{-\infty}D_t^{-\alpha}U_{0,h} \qquad (21)$$
$U_{\alpha,h}$ is the "$\alpha$ order integrated, fractional $h$ relaxation noise". Combined with the Green's
function relation $G_{\alpha,h} = {}_{-\infty}D_t^{-\alpha}G_{0,h}$ (eq. 15; recall that $G_{0,h}(t) = 0$ for $t<0$), we find that $U_{\alpha,h}$,
$G_{\alpha,h}$ are respectively the fractionally integrated relaxation noises and Green's functions of
the fractionally integrated fractional relaxation equation:
$$\left({}_{-\infty}D_t^{\alpha+h} + {}_{-\infty}D_t^{\alpha}\right)U_{\alpha,h} = \gamma; \quad \left({}_{-\infty}D_t^{\alpha+h} + {}_{-\infty}D_t^{\alpha}\right)G_{\alpha,h} = \delta(t) \qquad (22)$$
If the highest order derivative is constrained to be an integer (i.e. $\alpha+h = 1$ or $2$), then the
equation is a standard fractional Langevin equation, for example $U$ could for the velocity
of a particle with fractional damping and white noise forcing, although even here, the initial
conditions are usually taken to be at $t = 0$ not $t = -\infty$. Equivalently, $U_{\alpha,h}$, is the solution of
the relaxation equation but with an fGn forcing:
$$\left(_{-\infty}D_t^h+1\right)U_{\alpha,h} = {}_{-\infty}D_t^{-\alpha}\gamma = F_\alpha\left(t\right); \quad 0\le\alpha<1/2 \tag{23}$$
(the Weyl fractional derivatives commute). $F_\alpha$ is the $\alpha$ order fGn process, and the
restriction $\alpha<1/2$ is needed to ensure low frequency convergence (see below).
In the Earth's radiative balance, such fractionally integrated fRn processes arise in
two physically interesting situations. The first is where the forcing itself has a long
memory – e.g. it is an fGn process. Whereas the memory in a pure fRn process is purely
from the high frequency storage term, in this case, the forcing (the overall radiative
imbalance) also contributes to the memory and this has important consequences for the
predictability (section 4). Although the solutions $U_{\alpha,h}$ are mathematically the same whether
from the fractional relaxation equation with fGn forcing (eq. 23) or the fractionally
integrated fractional relaxation equation with white noise forcing (eq. 22), only the former
is directly relevant for the Earth energy balance. This is because the energy balance
involves the response from both stochastic (internal) *and* deterministic (external) forcing.
For the latter, it is important that following a step function forcing, at long times, the system
will approach a new state of thermodynamic equilibrium. This implies that the term in the
equation that dominates at low frequencies – the lowest order term - be of order zero so
that if $F$ in eq. 1 is a step function, that the new equilibrium temperature (anomaly) is $T =$
$sF$.
The second situation where fractionally integrated fRn processes arise is for the
energy storage (even in the purely white noise forcing case). The storage process is the
difference between the forcing and the response:
$$S_{\alpha,h} = F_\alpha - U_{\alpha,h} \tag{24}$$
so that:
$$S_{\alpha,h} = {}_{-\infty}D_t^h U_{\alpha,h} = U_{h-\alpha,h} \tag{25}$$
Even when the forcing is pure white noise ($\alpha = 0$), the storage is an $h$ ordered fractionally
integrated process: $S_{0,h} = U_{h,h}$; this corresponds to the storage following an impulse forcing.
The storage following a step forcing is obtained by integration order 1: $U_{1+h,h}$. Similarly,
the Green's function for the fRn storage following an impulse forcing is $G_{h,h}$ and following
a step forcing, $G_{1+h,h}$ (fig. 1b). Since it turns out that most of the pure fRn ($\alpha = 0$) results
are readily generalized to $0<\alpha<1/2$, many fractionally integrated results are given below.

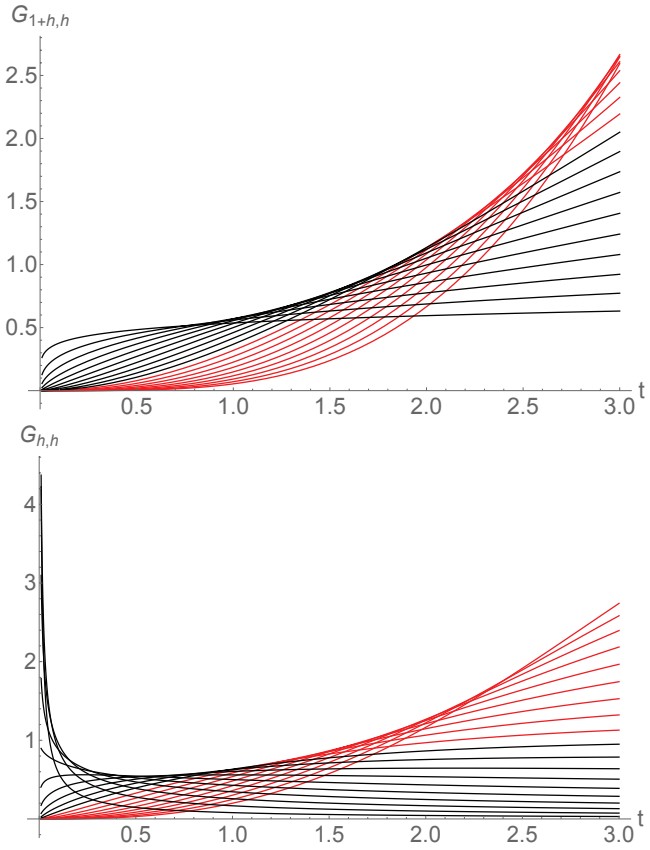

Fig. 1b:  The storage Green's functions for the fractional relaxation equation ($\alpha = 0$): top
impulse response ($G_{h,h}$), bottom, step response ($G_{1+h,h}$).  Black is for $h = 1/10, 2/10,..10/10$,
red for $11/10, 12/10, …19/10$ (to identify the curves, use the fact that at large $t$, they are in
order of increasing $h$ (bottom to top).  For small $t$, $G_{h,h} \propto t^{2h-1}$ (eq. 15) so that for $h \leq 1/2$,
the impulse response is singular at the origin.  For large $t$, $G_{h,h} \propto t^{h-1}$ (eq. 18) so that for
$h<1$, the total impulse response storage decreases following the impulse, for $h = 1$ (the
EBE), it tends to unity and for $h>1$, it diverges.
**2.4 Statistics**
In the above, we discussed fGn, fRn and their order one integrals fBm, fRm as well
as fractional generalizations, presenting a classical (real space) approach stressing the links
with fGn, fBm, we now turn to their statistics.  $U_{\alpha,h}(t)$ is a mean zero stationary Gaussian
process (i.e. $<U_{\alpha,h}(t)> = 0$ where "$<.>$" indicates ensemble or statistical averaging),
therefore its statistics are determined completely by it's autocorrelation function $R_{\alpha,h}(t)$
which is only a function of the lag $t$:
$$R_{\alpha,h}(t) = \left\langle U_{\alpha,h}(t+v) U_{\alpha,h}(v) \right\rangle = \int_0^\infty G_{\alpha,h}(t+v) G_{\alpha,h}(v) dv \qquad (26)$$

The far right equality follows from $U_{\alpha,h} = G_{\alpha,h} * \gamma$ and $\langle \gamma(t)\gamma(t') \rangle = \delta(t-t')$ ("*"
indicates "convolution"). The process can only be normalized by $R_{\alpha,h}(0)$ when there is
no small scale divergence i.e. when:
$$R_{\alpha,h}(0) = \langle U_{\alpha,h}^2 \rangle = \int_0^\infty G_{\alpha,h}(v)^2 \, dv < \infty; \quad \alpha + h > 1/2 \qquad (27)$$

When $\alpha+h<1/2$, this diverges; in order to be normalized, the process must be averaged at
a finite resolution (below).
Although it is possible to follow [*Mandelbrot and Van Ness*, 1968] and derive many
statistical properties in real space, a Fourier approach is not only more streamlined, but is
more powerful.   The reason for the simplicity of the Fourier approach is that the Fourier
Transform (*FT*, indicated by the tilda) of the Weyl fractional derivative is symbolically:
$$(i\omega)^h \overset{FT}{\leftrightarrow} {}_{-\infty}D_t^h \qquad (28)$$

(e.g. [*Podlubny*, 1999], this is simply the extension of the usual rule for the *FT* of integer-
ordered derivatives).   Therefore since $U_{\alpha,h}$, $G_{\alpha,h}$ are respectively solutions and Green's
functions of the fractionally integrated fractional relaxation equation (eq. 22) we have:
$$\left( (i\omega)^{\alpha+h} + (i\omega)^\alpha \right)\tilde{U}_{\alpha,h} = \tilde{\gamma} \overset{FT}{\leftrightarrow} \left( {}_{-\infty}D_t^{\alpha+h} + {}_{-\infty}D_t^\alpha \right)U_{\alpha,h} = \gamma, \qquad (29)$$

$$\left( (i\omega)^{\alpha+h} + (i\omega)^\alpha \right)\tilde{G}_{\alpha,h} = 1 \overset{FT}{\leftrightarrow} \left( {}_{-\infty}D_t^{\alpha+h} + {}_{-\infty}D_t^\alpha \right)G_{\alpha,h} = \delta$$

So that:
$$\tilde{U}_{\alpha,h}(\omega) = \frac{\tilde{\gamma}}{(i\omega)^\alpha \left(1+(i\omega)^h\right)}; \quad \tilde{G}_{\alpha,h}(\omega) = \frac{1}{(i\omega)^\alpha \left(1+(i\omega)^h\right)}; \quad 0<\alpha<1; \quad 0<h<2$$

$$(30)$$

We see that in the limit $h \to 0$, $U_{\alpha,0}$ is an $\alpha$ order fGn process (see e.g. eq. 23).

Now we can use the fact that the white noise $\gamma$ has a flat spectrum:

$$\langle \tilde{\gamma}(\omega)\tilde{\gamma}(\omega') \rangle = \delta(\omega+\omega')\langle |\tilde{\gamma}(\omega)|^2 \rangle = 2\pi\delta(\omega+\omega') \overset{FT}{\leftrightarrow} \langle \gamma(t)\gamma(t') \rangle = \delta(t-t')$$

$$(31)$$

The modulus (vertical bars) intervene since for any real function $f(t)$ we have
$\tilde{f}(\omega) = \tilde{f}^*(-\omega)$, where the superscript "*" indicates complex conjugate.

Application of eq. 31 leads to:

$$R_{\alpha,h}(t) = \frac{1}{2\pi}\int_{-\infty}^\infty e^{i\omega t} E_U(\omega)\, d\omega; \quad E_U(\omega) = \langle |\tilde{U}_{\alpha,h}(\omega)|^2 \rangle = \frac{1}{|\omega|^{2\alpha}\left(1+(-i\omega)^h\right)\left(1+(i\omega)^h\right)}$$

$$(32)$$

i.e. the spectrum $E_U$ is the *FT* of the correlation function $R_{\alpha,h}(t)$ (the Wiener-Khintchin
theorem).  Applying this to $U_{\alpha,h}$, we obtain:

$$R_{\alpha,h}(t) = \frac{1}{2\pi} \int\limits_{-\infty}^{\infty} \frac{\cos(\omega t)d\omega}{|\omega|^{2\alpha}\left(1+(i\omega)^h\right)\left(1+(-i\omega)^h\right)} \tag{33}$$

This shows that $R_{\alpha,h}(t) = R_{\alpha,h}(-t)$ so that below, we only consider $t \geq 0$.
Since, $R_{\alpha,h}(0)$ diverges for $\alpha+h<1/2$, we consider the integral $Q_{\alpha,h}$ of the process
(the "motion") from which we can easily compute the average.  The corresponding variance
$V_{\alpha,h}$ is:

$$V_{\alpha,h}(t) = \left\langle Q_{\alpha,h}(t)^2 \right\rangle; \quad Q_{\alpha,h}(t) = \int\limits_0^t U_{\alpha,h}(v)dv \tag{34}$$

In terms of $\widetilde{U}_{\alpha,h}(\omega)$:

$$V_{\alpha,h}(t) = \frac{1}{\pi} \int\limits_{-\infty}^{\infty} \frac{(1-\cos\omega t)}{\omega^2}\left\langle \left|\widetilde{U}_{\alpha,h}(\omega)\right|^2 \right\rangle d\omega = \frac{1}{\pi} \int\limits_{-\infty}^{\infty} \frac{(1-\cos\omega t)}{|\omega|^{2+2\alpha}} \frac{d\omega}{\left(1+(i\omega)^h\right)\left(1+(-i\omega)^h\right)}$$

$\alpha<1/2,$ $0<h<2$.  (35)
We see that at low frequencies, when $\alpha \geq 1/2$ the integral diverges for all $t$.  Also note that
a series expansion for $V_{\alpha,h}(t)$ in $t$ will have only even ordered integer power terms.
Comparing eqs. 33, 35 we see that *R, V* are linked by the simple relation:

$$R_{\alpha,h}(t) = \frac{1}{2}\frac{d^2 V_{\alpha,h}(t)}{dt^2} \tag{36}$$

Therefore by integrating eq. 26 (twice), we can express $V_{\alpha,h}$ in terms of $G_{\alpha,h}$:

$$V_{\alpha,h}(t) = \int\limits_0^\infty \left(G_{\alpha+1,h}(t+v) - G_{\alpha+1,h}(v)\right)^2 dv + \int\limits_0^t G_{\alpha+1,h}(v)^2 dv \tag{37}$$

This can be verified by differentiation and using $\dfrac{dG_{\alpha+1,h}}{dt} = G_{\alpha,h}$.
The basic behaviour can be understood in the Fourier domain.  First, putting $t = 0$ in
eq. 32 (i.e. "Parseval's theorem") we have:

$$R_{\alpha,h}(0) = \frac{1}{2\pi}\int\limits_{-\infty}^{\infty} E_U(\omega)d\omega = \frac{1}{2\pi}\int\limits_{-\infty}^{\infty} \frac{d\omega}{|\omega|^{2\alpha}\left(1+(i\omega)^h\right)\left(1+(-i\omega)^h\right)} \tag{38}$$

So that when $\alpha+h <1/2$, *R* diverges at high frequencies (small *t*), hence to represent a
physical process (here, the Earth's temperature), the process must be averaged over a finite
resolution $\tau$.  When $\alpha+h>1/2$, $R(0)$ is finite and can therefore be used to obtain a normalized
autocorrelation function (eq. 27).
From eq. 32, we may also easily obtain the asymptotic high and low frequency
behaviours of the energy spectrum:

$$E_U(\omega) \propto \begin{cases} \omega^{-2(\alpha+h)} + O\left(\omega^{-2\alpha-3h}\right); & \omega \gg 1 \\ \omega^{-2\alpha} - 2\cos\left(\dfrac{\pi h}{2}\right)\omega^{h-2\alpha} + O\left(\omega^{2h-2\alpha}\right) & \omega \ll 1 \end{cases}.$$

(39)

### 2.5 Finite resolution processes


When $\alpha+h<1/2$ the process doesn't converge at any instant $t$, it is a noise, a
generalized function. To represent the Earth's temperature it must therefore be averaged
at a finite resolution $\tau$:
$$U_{\alpha,h,\tau}(t) = \frac{Q_{\alpha,h}(t) - Q_{\alpha,h}(t-\tau)}{\tau}.$$
(40)

Applying eq. 34, 40, we obtain the "resolution $\tau$" autocorrelation:
$$R_{\alpha,h,\tau}(\Delta t) = \left\langle U_{\alpha,h,\tau}(t)U_{\alpha,h,\tau}(t-\Delta t)\right\rangle = \tau^{-2}\left\langle \left(Q_{\alpha,h}(t) - Q_{\alpha,h}(t-\tau)\right)\left(Q_{\alpha,h}(t-\Delta t) - Q_{\alpha,h}(t-\Delta t-\tau)\right)\right\rangle$$
$$= \tau^{-2}\frac{1}{2}\left(V_{\alpha,h}(\Delta t - \tau) + V_{\alpha,h}(\Delta t + \tau) - 2V_{\alpha,h}(\Delta t)\right)$$
$\Delta t \geq \tau$

$$R_{\alpha,h,\tau}(0) = \tau^{-2}V_{\alpha,h}(\tau),$$
(41)

Alternatively, measuring time in units of the resolution $\lambda = \Delta t/\tau$:
$$R_{\alpha,h,\tau}(\lambda\tau) = \left\langle U_{\alpha,h,\tau}(t)U_{\alpha,h,\tau}(t-\lambda\tau)\right\rangle = \tau^{-2}\frac{1}{2}\left(V_{\alpha,h}\left((\lambda-1)\tau\right) + V_{\alpha,h}\left((\lambda+1)\tau\right) - 2V_{\alpha,h}(\lambda\tau)\right); \quad \lambda \geq 1$$

(42)

$R_{a,h,\tau}$ can be conveniently written in terms of centred finite differences:
$$R_{\alpha,h,\tau}(\lambda\tau) = \frac{1}{2}\Delta_\tau^2 V_{\alpha,h}(\lambda\tau) \approx \frac{1}{2}V_{\alpha,h}{''}(\Delta t); \quad \Delta_\tau f(t) = \frac{f(t+\tau/2) - f(t-\tau/2)}{\tau}.$$

(43)

The finite difference formula is valid for $\Delta t \geq \tau$. For finite $\tau$, it allows us to obtain the
correlation behaviour by replacing the second difference by a second derivative, an
approximation that is very good except when $\Delta t$ is close to $\tau$. Taking the limit $\tau \to 0$ in
eq. 43 we obtain the second derivative formula eq. 36.

## 3 Application to fBm, fGn, fRm, fRn


### 3.1 fBm, fGn


The above derivations were for noises and motions derived from differential
operators whose impulse and step Green's functions had convergent $V_{\alpha,h}(t)$. Before
applying them to fRn, fRm, we illustrate this by applying them first to fBm and fGn.
The fBm results are obtained by using the fGn step Green's function (eq. 13) in eq.
35 with $h = 0$ to obtain:
$$V_h^{(fBm)}(t) = 4V_{\alpha=h,0}(t) = \left(\frac{2\sin(\pi h)\Gamma(-1-2h)}{\pi}\right)t^{2h+1}; \quad -\frac{1}{2} \le h < \frac{1}{2} \quad.$$

(44)

The standard normalization and parametrisation is:

$$N_h = K_h = \left(\frac{\pi}{2\sin(\pi h)\Gamma(-1-2h)}\right)^{1/2}$$
$$= \left(-\frac{\pi}{2\cos(\pi H)\Gamma(-2H)}\right)^{1/2}; \qquad H = h + \frac{1}{2}; \quad 0 \le H < 1$$

. (45)

This normalization turns out to be convenient not only for fBm but also for fRm so that for
the normalized process:
$$V_H^{(fBm)}(t) = t^{2h+1} = t^{2H}; \quad 0 \le H < 1 \quad, \tag{46}$$

Where we have introduced the standard fBm parameter $H = h+1/2$ so that:
$$\left\langle \Delta B_H(\Delta t)^2 \right\rangle^{1/2} = \Delta t^H; \quad \Delta B_H(\Delta t) = B_H(t) - B_H(t - \Delta t) \quad, \tag{47}$$

hence $H$ is the fluctuation exponent for fBm. Note that fBm is usually *defined* as the
Gaussian process with $V_H$ given by eq. 46 i.e. with this normalization (e.g. [*Biagini et al.*,
2008]).
We can now calculate the correlation function relevant for the fGn statistics. With
the above normalization:

$$R_{h,\tau}^{(fGn)}(\lambda\tau) = \frac{1}{2}\tau^{2h-1}\left((\lambda+1)^{2h+1} + (\lambda-1)^{2h+1} - 2\lambda^{2h+1}\right); \quad \lambda \ge 1; \quad -\frac{1}{2} < h < \frac{1}{2}$$

$$R_{h,\tau}^{(fGn)}(0) = \tau^{2h-1}$$


$$R_{H,\tau}^{(fGn)}(\lambda\tau) \approx h(2h+1)(\lambda\tau)^{2h-1} = H(2H-1)(\lambda\tau)^{2(H-1)}; \quad \lambda \gg 1 \quad, \tag{48}$$

the bottom approximations are valid for large scale ratios $\lambda$. We note the difference in sign
for $H > \frac{1}{2}$ ("persistence"), and for $H < \frac{1}{2}$ ("antipersistence"). When $H = \frac{1}{2}$, the noise
corresponds to standard Brownian motion, it is uncorrelated.
**3.2 fRm, fRn**
3.2.1 $R_{\alpha,h}(t)$
Since fRm, fRn are Gaussian, their properties are determined by their second order
statistics, by $V_{\alpha,h}(t)$, $R_{\alpha,h}(t)$. These statistics are second order in $G_{\alpha,h}(t)$ and can most easily
be determined using the Fourier representation of $G_{\alpha,h}(t)$, (section 2.4, appendix A, B). The
development is challenging because unlike the $G_{\alpha,h}(t)$ functions that are entirely expressed
in series of fractional powers of $t$, $V_{\alpha,h}(t)$ and $R_{\alpha,h}(t)$ involve mixed fractional and integer
power expansions, the details are given in the appendices, here we summarize the main
results.
First, for the noises, we have:
$$R_{\alpha,h}(t) = \sum_{n=2}^{\infty} D_n \Gamma(1-hn-2\alpha)t^{-1+hn+2\alpha} + \sum_{j=1,odd}^{\infty} F_j \frac{t^{j-1}}{\Gamma(j)};$$

$$F_j = -\frac{\cos\pi\left(\frac{h}{2}+\alpha\right)}{h\sin\left(\frac{\pi h}{2}\right)\sin\left(\frac{\pi}{h}(j-2\alpha)\right)}; \qquad D_n = (-1)^n \frac{\sin\left(\frac{n\pi h}{2}+\alpha\pi\right)\sin\left(\frac{(n-1)\pi h}{2}\right)}{\pi\sin\left(\frac{\pi h}{2}\right)}$$

                                                                                            (49)

At small $t$, the lowest order terms dominate, the normalized autocorrelations are thus:
$$R_{\alpha,h}^{(norm)}(t) = (h+\alpha)(1+2(h+\alpha))t^{-1+2(h+\alpha)} + O(t^{-1+3h+2\alpha}); \quad \tau \ll t \ll 1; \quad 0 < (h+\alpha) < 1/2$$

$$R_{\alpha,h}^{(norm)}(t) = 1 - \frac{\left|\Gamma(1-2(h+\alpha))\right|\sin\left(\pi(h+2\alpha)\right)}{\pi F_1}t^{-1+2(h+\alpha)} + O(t^{-1+3h+2\alpha}); \quad \begin{array}{c} t \ll 1; \\ 1/2 < (h+\alpha) < 3/2 \end{array}$$

$$R_{\alpha,h}^{(norm)}(t) = 1 + \frac{t^2}{2F_1}F_3 + O(t^{-1+2(h+\alpha)})...; \quad t \ll 1; \quad 3/2 < (h+\alpha) < 2$$

(50)

(note $F_3 < 0$ for $3/2 < h+\alpha < 2$, see appendix A).  We see that at small $t$, the behaviour of the
normalized autocorrelations depend essentially on the sum $h+\alpha$, in particular, when
$h+\alpha < 1/2$, the process is effectively an fGn process with effective fluctuation exponent $H =$
$-\frac{1}{2} + (h+\alpha)$.  This is to be expected since $\alpha+h$ is the highest order term in the fractionally
integrated fractional relaxation equation (eq. 22).

3.2.2 $V_{\alpha,h}(t)$

Integrating twice $V_{\alpha,h}(t) = 2\int_0^t\left(\int_0^v R_{\alpha,h}(u)du\right)dv$, we obtain:

$$V_{\alpha,h}(t) = 2\sum_{n=2}^{\infty} D_n\Gamma(-1-hn-2\alpha)t^{1+hn+2\alpha} + 2\sum_{j=1,odd}^{\infty} F_j\frac{t^{j+1}}{\Gamma(j+2)}; \quad 0 < h < 2; \quad 0 \le \alpha < 1/2$$

(51)

When $0 < \alpha+h < 1/2$, the leading ($n = 2$) term for $V_{\alpha,h}$ is $t^{1+2(h+\alpha)}$, ($\propto V_{\alpha+h}^{(fBm)}$) so that the fBm
coefficient can be used for normalization using $R_{\alpha,h,\tau}(0) = \tau^{-2}V_{\alpha,h}(\tau)$.  When $h+\alpha > 1/2$, this
normalization becomes negative, so that it cannot be used, however in this case, $R_{\alpha,h}(0) =$
$F_1$ and may be used for normalization instead. For an analytic expression, convergence
properties including numerical results and modified expansions that converge more rapidly,
see appendix A, for the special case $h = 1/2$, appendix B.

For convenience, the leading terms of the normalized $V_{\alpha,h}$ are:

$$V_{\alpha,h}^{(norm)}(t) = t^{1+2(h+\alpha)} + O(t^{1+3h+2\alpha}) + O(t^2); \quad 0 < (h+\alpha) < 1/2 \qquad (52)$$

$\quad V_{\alpha,h}^{(norm)}(t) = t^2 - \dfrac{2\Gamma(-1-2(h+\alpha))\sin(\pi(h+2\alpha))}{\pi F_1}t^{1+2(h+\alpha)} + O(t^{1+3h+2\alpha}); \quad 1/2 < (h+\alpha) < 3/2$
$\quad V_{\alpha,h}^{(norm)}(t) = t^2 + \dfrac{F_3}{12F_1}t^4 + O(t^{2(h+\alpha)+1}); \quad 3/2 < (h+\alpha) < 2$

### 596 3.2.3 Asymptotic expansions

For multidecadal global climate projections, the relaxation time has been estimated
at $\approx$ 5 years ([*Procyk et al.*, 2020; 2022]), so that we are interested in the long time
behaviour (exploited for example in [*Hébert et al.*, 2021]). For this, asymptotic expansions
are needed, in appendix A we show:
$\quad R_{\alpha,h}(t) = -\displaystyle\sum_{n=0}^{\infty} D_{-n}\Gamma(1+nh-2\alpha)t^{2\alpha-(1+nh)} + P_{\alpha,h,+}(t); \quad t \gg 1$       (53)
Where the $P_{\alpha,h,+}(t) = 0$ for $h<1$ while for $1<h<2$ it has exponentially damped oscillations
(see fig. 2 lower right and appendix A).

For pure fRn processes a useful formula is:

$$R_{0,h}(t) = \sum_{n=1}^{\infty}(-1)^n \frac{1+\cot\left(\dfrac{\pi h}{2}\right)\tan\left(\dfrac{n\pi h}{2}\right)}{2\Gamma(-nh)}t^{-(1+nh)} + P_{0,h,+}(t); \qquad t \gg 1$$


(54)

Or more generally:
$\quad R_{\alpha,h}(t) = \dfrac{\Gamma(1-2\alpha)\sin(\pi\alpha)}{\pi}t^{2\alpha-1} - \dfrac{\cos\left(\dfrac{\pi h}{2}\right)}{\cos\left(\dfrac{\pi h}{2}-\pi\alpha\right)\Gamma(2\alpha-h)}t^{2\alpha-(1+h)} + ...$

$t \gg 1; \qquad 0 \le h < 2; \qquad 0 \le \alpha < 1/2$   (55)
We see that when $\alpha \neq 0$, $D_0 > 0$ so that as expected, the leading behaviour has no $h$
dependence, it is only due to the long range correlations in the forcing; we obtain the fGn
result: $\approx t^{2\alpha-1}$. For pure fRn processes this reduces to $R_{0,h}(t) = -\dfrac{1}{\Gamma(-h)}t^{-1-h}$ (note that
$\Gamma(-h)<0$ for $0<h<1$).

Integrating $R_{\alpha,h}$ twice and doubling, we obtain

$\quad V_{\alpha,h}(t) = \dfrac{2\Gamma(-1-2\alpha)\sin(\pi\alpha)}{\pi}t^{1+2\alpha} + a_{\alpha,h}t + b_{\alpha,h} - \dfrac{1+\cos(\pi h)-\sin(\pi h)\cot(\pi(h-2\alpha))}{\Gamma(2-(h-2\alpha))}t^{1+2\alpha-h} + ...; \quad t \gg 1$

(56)
(the full expansion is given in appendix A, see fig. 3 for plots). The constants of integration
$a_{\alpha,h}$, $b_{\alpha,h}$ are not determined since the expansion is not valid at $t = 0$; they can be determined
numerically if needed. However, in the limit $\alpha \to 0$ (the pure fRn case), the leading term is
exactly $t$ (corresponding to ordinary Brownian motion) so that an extra $a_{0,h}$ is not needed
(appendix A).    When $\alpha > 0$, the far left (fGn) term from the forcing dominates, at large
enough $t$, $V_{\alpha,h}(t) \propto t^{2H}$ with $H = \alpha + 1/2$, the corresponding motion is an fBm.

Using the above results we see that there are three limiting fRn/fRm cases that yield
fGn/fBm processes:

$$R_{\alpha,0}(t) = \frac{1}{4} R_{\alpha}^{(fGn)}(t); \quad\quad 0 < \alpha < 1/2; \quad\quad h = 0$$

$$R_{\alpha,h}(t) = R_{\alpha}^{(fGn)}(t); \quad\quad 0 < \alpha < 1/2; \quad\quad t \gg 1 \quad\quad\quad\quad (57)$$

$$R_{\alpha,h}(t) = R_{\alpha+h}^{(fGn)}(t); \quad\quad 0 < \alpha + h < 1/2; \quad t \approx 0$$

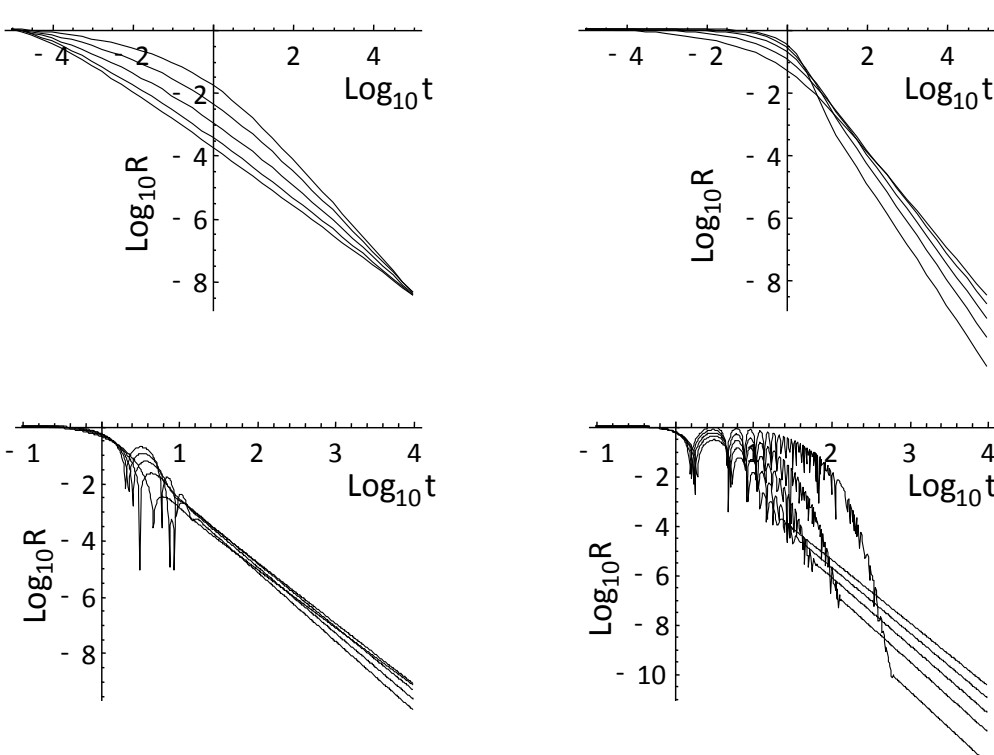

Fig. 2: The normalized correlation functions $R_{0,h}$ for fRn corresponding to the $V_{0,h}$ function
in fig. 2: $0 < h < 1/2$ (upper left) $1/2 < h < 1$ (upper right), $1 < h < 3/2$) lower left, $3/2 < h < 2$ lower
right.    In each plot, the curves correspond to $h$ increasing from bottom to top in units of $1/10$
starting from $1/20$ (upper left) to $39/20$ (bottom right). For $h < 1/2$, the resolution is important since
$R_{0,h,\tau}$ diverges at small $\tau$.   In the upper left figure, $R_{0,h,\tau}$ is shown with $\tau = 10^{-5}$; they were
normalized to the value at resolution $\tau = 10^{-5}$, for $h > 1/2$, the curves are normalized with $F_3^{-1/2}$.   In
all cases, the large $t$ slope is $-1-h$.

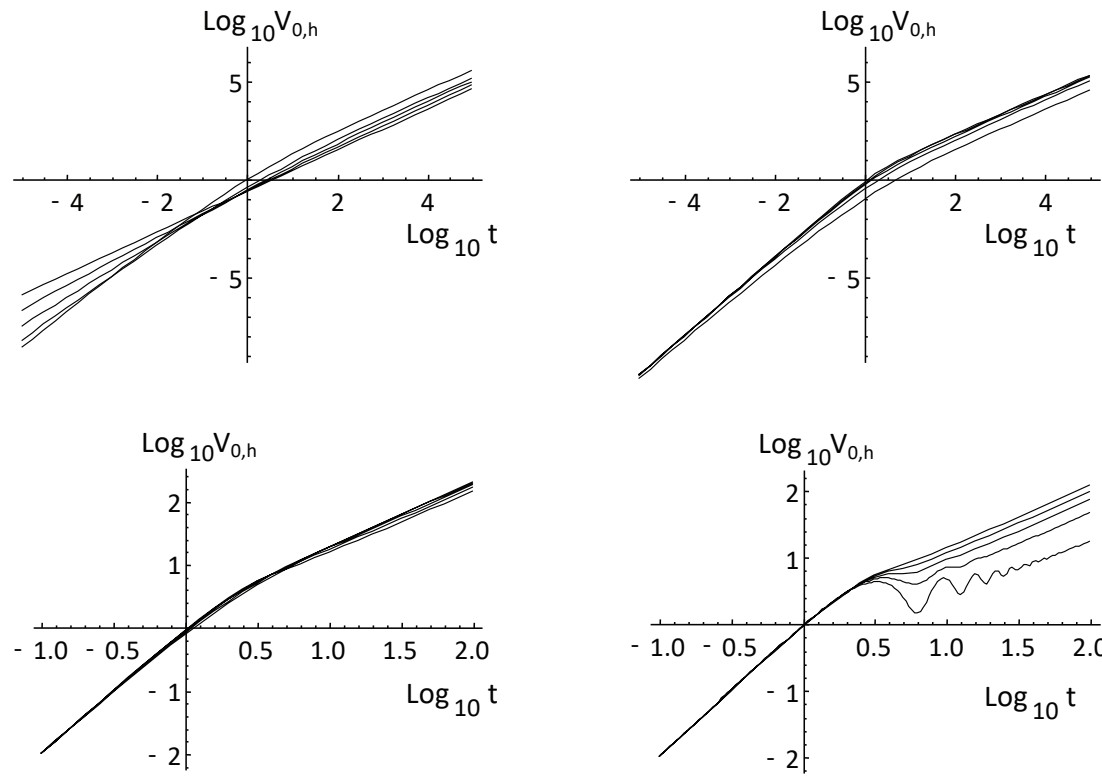

Fig. 3: The normalized $V_{0,h}$ functions for the various ranges of $h$ for fRm. The plots from
left to right, top to bottom are for the ranges $0 < h < 1/2$, $1/2 < h < 1$, $1 < h < 3/2$, $3/2 < h < 2$. Within
each plot, the lines are for $h$ increasing in units of 1/10 starting at a value 1/20 above the plot
minimum; overall, $h$ increases in units of 1/10 starting at a value 1/20, upper left to 39/20,
bottom right (ex. for the upper left, the lines are for $h$ = 1/20, 3/10, 5/20, 7/20, 9/20). For all $h$'s
the large $t$ behaviour is linear (slope = 1, although note the oscillations for the lower right hand plot
for $3/2 < h < 2$). For small $t$, the slopes are $1+2h$ ($0 < h \leq 1/2$) and 2 ($1/2 \leq h < 2$).

## 3.3 Haar fluctuations

A useful statistical characterization of the processes is by the statistics of their Haar
fluctuations over an interval $\Delta t$. For an interval $\Delta t$, Haar fluctuations (based on Haar
wavelets) are the differences between the averages of the first and second halves of an
interval. For a process $U$, the Haar fluctuation is:
$$\Delta U\left(\Delta t\right)_{Haar} = \frac{2}{\Delta t} \int_{t-\Delta t/2}^{t} U\left(v\right)dv - \frac{2}{\Delta t} \int_{t-\Delta t}^{t-\Delta t/2} U\left(v\right)dv.$$
(58)

In terms of the process at resolution $\Delta t/2$, (i.e. averaged at this scale) $U_{\Delta t/2}\left(t\right)$:
$$\Delta U\left(\Delta t\right)_{Haar} = \frac{2}{\Delta t}\left(U_{\Delta t/2}\left(t\right) - U_{\Delta t/2}\left(t - \Delta t/2\right)\right).$$
(59)

Therefore:
$$\left\langle \Delta U\left(\Delta t\right)_{Haar}^{2} \right\rangle = \left(\frac{2}{\Delta t}\right)^{2}\left(4V\left(\Delta t/2\right)-V\left(\Delta t\right)\right). \tag{60}$$

Where $V(t)$ is the variance of the integral of $U$ over an interval $t$ (eq. 34).
Using eq. 60 we can determine the behaviour of the RMS Haar fluctuations; terms
like $V_{\alpha,h}\left(t\right)\propto t^{\xi}$ contribute $\propto t^{\xi/2-1}$ to the RMS Haar fluctuation $\left\langle \Delta U_{\alpha,h}\left(\Delta t\right)_{Haar}^{2} \right\rangle^{1/2}$ (the
exception is when $\xi = 2$ which contributes nothing). Applying this equation to fGn
parameter $h$ we obtain $\left\langle \Delta F_{h}\left(\Delta t\right)_{Haar}^{2} \right\rangle^{1/2} \propto \Delta t^{H}$ with $H = h - \frac{1}{2}$.
Using the results above for $V_{\alpha,h}$ we therefore obtain the leading exponents:

$$
\begin{aligned}
H &= h+\alpha-1/2; \quad 0<h+\alpha<3/2 \\
H &= 1; \qquad\qquad 3/2<h+\alpha<2
\end{aligned} \;;\quad \Delta t\ll 1
$$

$$H = \alpha - \frac{1}{2}; \quad \Delta t \gg 1$$

(61)

Fig. 4 shows that the theory agrees well with the numerics.
For the range of $\alpha$, $h$ discussed here ($0\leq\alpha<1/2$, $0\leq h\leq 2$), $H$ spans the range -1/2 (white
noise) to 1. In comparison, fGn processes have $H$ covering the range $-1 <H <0$ and fBm
processes have $0<H<1$, therefore, depending on whether the process is observed at time
scales below or above the relaxation time scale ($\Delta t = 1$), fractionally integrated fRn
processes can mimick fGn or fBm processes. If we consider the integrals - the motions -
the value of $H$ is increased by 1 (although for Haar fluctuations, it cannot exceed $H = 1$).
Overall, from an empirical viewpoint, if over some range of scales (that may only be a
factor of 100 or less), it may be quite hard to distinguish the various models, especially
since the transition from low to high frequency scaling may be very slow (see especially
appendix B for the $h = 1/2$ case). Recent work shows that the maximum likelihood method
may be the optimum parameter estimation technique [*Procyk*, 2021].

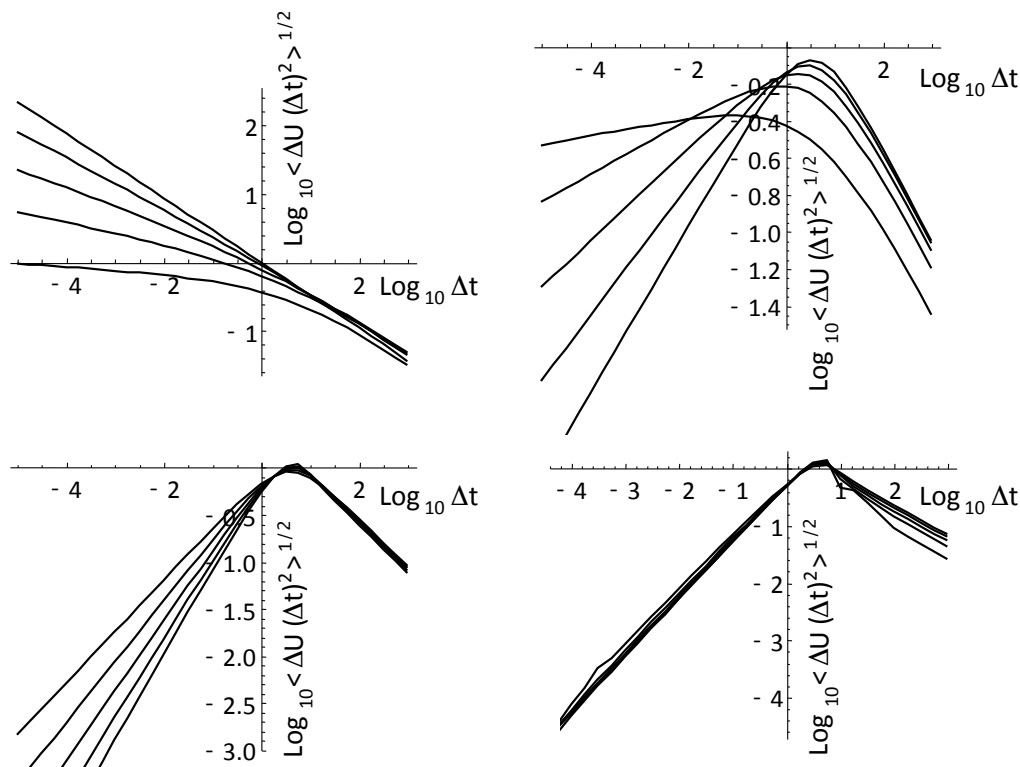

Fig. 4: The RMS Haar fluctuation plots for the pure ($\alpha = 0$) fRn process for $0 < h < 1/2$
(upper left), $1/2 < h < 1$ (upper right), $1 < h < 3/2$ (lower left), $3/2 < h < 2$ (lower right). The
individual curves correspond to those of fig. 2, 3. The small $\Delta t$ slopes follow the theoretical
values $h - 1/2$ up to $h = 3/2$ (slope= 1); for larger $h$, the small $t$ slopes all = 1. Also, at large $t$
due to dominant $V \approx t$ terms, in all cases we obtain slopes $t^{-1/2}$.
**3.4 Sample processes**
It is instructive to view some samples of fRn, fRm processes, (here we consider only
$\alpha = 0$). For simulations, both the small and large scale divergences must be considered.
Starting with the approximate methods developed by [*Mandelbrot and Wallis*, 1969], it
took some time for exact fBm, and fGn simulation techniques to be developed [*Hipel and*
*McLeod*, 1994], [*Palma*, 2007]. Fortunately, for fRm, fRn, the low frequency situation is
easier since the long time memory is much smaller than for fBm, fGn. Therefore, as long
as we are careful to always simulate series a few times longer than the relaxation time and
then to throw away the earliest 2/3 or 3/4 of the simulation, the remainder will have accurate
statistics. With this procedure to take care of low frequency issues, we can therefore use
the solution for fRn in the form of a convolution, and use standard numerical convolution
algorithms.
We must nevertheless be careful about the high frequencies since the impulse
response Green's functions $G_{0,h}$ are singular for $h<1$. In order to avoid singularities,
simulations of fRn are best made by first simulating the motions $Q_{0,h}$ using $Q_{0,h} \propto G_{1,h} * \gamma$
and obtain the resolution $\tau$ fRn, using $U_{0,h,\tau}(t) = \left(Q_{0,h}(t+\tau) - Q_{0,h}(t)\right)/\tau$. Numerically,
this allows us to use the smoother (nonsingular) $G_{1,h}$ in the convolution rather than the
singular $G_{0,h}$. The simulations shown in figs. 5, 6 follow this procedure and the Haar
fluctuation statistics were analyzed verifying the statistical accuracy of the simulations.
In order to clearly display the behaviours, recall that when $t>>1$, we showed that all
the fRn converge to Gaussian white noises and the fRm to Brownian motions (albeit in a
slow power law manner). At the other extreme, for $t << 1$, we obtain the fGn and fBm
limits (when $0 < h < 1/2$) and their generalizations for $1/2 < h < 2$.
Fig. 5a shows three simulations, each of length $2^{19}$, pixels, with each pixel
corresponding to a temporal resolution of $\tau = 2^{-10}$ so that the unit (relaxation) scale is $2^{10}$
elementary pixels. Each simulation uses the same random seed but they have $h$'s increasing
from $h = 1/10$ (top set) to $h = 5/10$ (bottom set). The fRm at the right is from the running
sum of the fRn at the left. Each series has been rescaled so that the range (maximum -
minimum) is the same for each. Starting at the top line of each group, we show $2^{10}$ points
of the original series degraded by a factor $2^9$. The second line shows a blow-up by a factor
of 8 of the part of the upper line to the right of the dashed vertical line. The line below is
a further blown up by factor of 8, until the bottom line shows 1/512 part of the full
simulation, but at full resolution. The unit scale indicating the transition from small to
large is shown by the horizontal red line in the middle right figure. At the top (degraded
by a factor $2^9$), the unit (relaxation) scale is 2 pixels so that the top line degraded view of
the simulation is nearly a white noise (left), (ordinary) Brownian motion (right). In contrast,
the bottom series is exactly of length unity so that it is close to the fGn limit with the
standard exponent $H = h+1/2$. Moving from bottom to top in fig. 5a, one effectively
transitions from fGn to fRn (left column) and fBm to fRm (right).
If we take the empirical relaxation scale for the global temperature to be $2^7$ months
($\approx$10 years, [*Lovejoy et al.*, 2017]) and we use monthly resolution temperature anomaly
data, then the nondimensional resolution is $2^{-7}$ corresponding to the second series from the
top (which is thus $2^{10}$ months $\approx$ 80 years long). Since $h \approx 0.38\pm0.03$ [*Procyk et al.*, 2022],
the second series from the top in the bottom set is the most realistic, we can make out the
low frequency ondulutions that are mostly present at scales 1/8 of the series (or less).
Fig. 5b shows realizations constructed from the same random seed but for the
extended range $1/2 < h < 2$ (i.e. beyond fGn). Over this range, the top (large scale,
degraded resolution) series are close to white noises (left) and Brownian motions (right).
For the bottom series, there is no equivalent fGn or fBm process, the curves become
smoother although the rescaling may hide this somewhat (see for example the $h = 13/20$
set, the blow-up of the far right 1/8 of the second series from the top shown in the third line.
For $1 < h < 2$, also note the oscillations with frequency $2\pi / \sin(\pi / h)$ (eq. 53, A.3), this is
the fractional oscillation range.
Fig. 6a shows simulations similar to fig. 5a (fRn on the left, fRm on the right) except
that instead of making a large simulation and then degrading and zooming, all the
simulations were of equal length ($2^{10}$ points), but the relaxation scale was changed from
$2^{15}$ pixels (bottom) to $2^{10}$, $2^5$ and 1 pixel (top). Again the top is white noise (left), Brownian
motion (right), and the bottom is (nearly) fGn (left) and fBm (right), fig. 6b shows the
extensions to $1/2 < h < 2$.

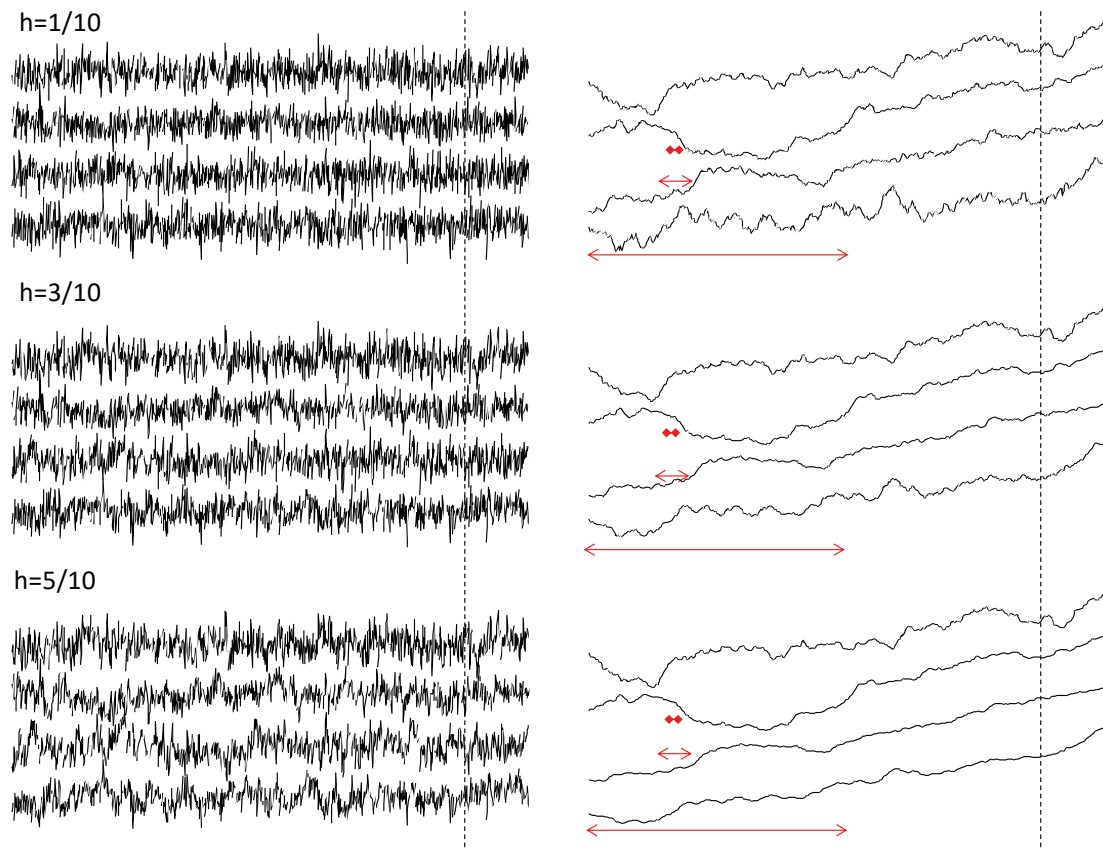

Fig. 5a: fRn and fRm simulations (left and right columns respectively) for $h$ = 1/10, 3/10,
5/10 (top to bottom sets, all with $\alpha$ = 0) i.e. the exponent range that overlaps with fGn and fBm.
There are three simulations, each of length $2^{19}$ pixels, each use the same random seed with the unit
scale equal to $2^{10}$ pixels (i.e. a resolution of $\tau = 2^{-10}$). The entire simulation therefore covers the
range of scale 1/1024 to 512 units. The fRm at the right is from the running sum of the fRn at the
left.
Starting at the top line of each set, we show $2^{10}$ points of the original series degraded in
resolution by a factor $2^9$. Since the length is $t = 2^9$ units long, each pixel has resolution $\tau = 1/2$).
The second line of each set takes the segment of the upper line lying to the right of the dashed
vertical line, 1/8 of its length. It therefore spans t=0 to $t = 2^9/8 = 2^6$ but resolution was taken as $\tau =$
$2^{-4}$, hence it is still $2^{10}$ pixels long. Since each pixel has a resolution of $2^{-4}$, the unit scale is $2^4$ pixels
long, this is shown in red in the second series from the top (middle set). The process of taking 1/8
and blowing up by a factor of 8 continues to the third line (length $t = 2^3$, resolution $\tau = 2^{-7}$), unit
scale =$2^7$ pixels (shown by the red arrows in the third series) until the bottom series which spans
the range $t = 0$ to $t = 1$ and a resolution $\tau = 2^{-10}$ with unit scale $2^{10}$ pixels (the whole series displayed).
Each series was rescaled in the vertical so that its range between maximum and minimum was the
same.
The unit relaxation scales indicated by the red arrows mark the transition from small to large
scale. Since the top series in each set has a unit scale of 2 (degraded) it is nearly a white noise (left),
or (ordinary) Brownian motion (right). In contrast, the bottom series is exactly of length $t = 1$ so
that it is close to the fGn and fBm limits (left and right) with the standard exponent $H = h$ +1/2. As
indicated in the text, the second series from the top in the bottom set is most realistic for monthly
temperature anomalies.


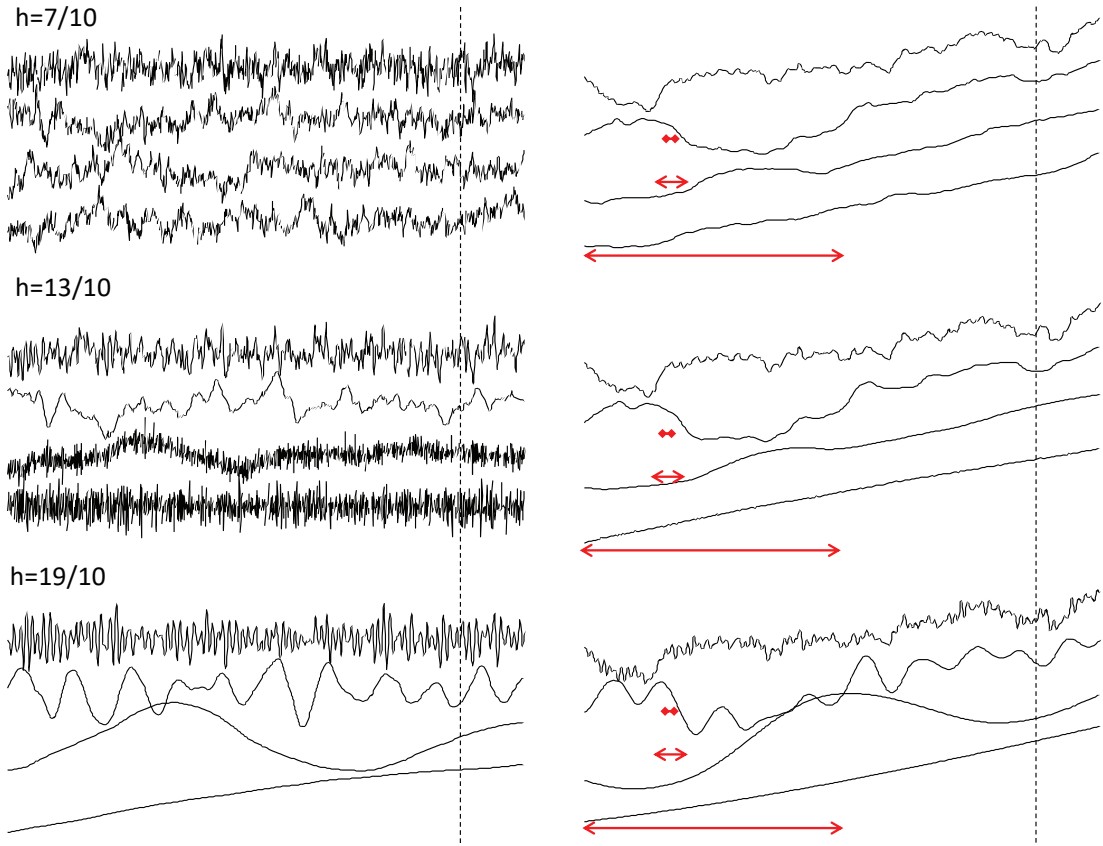

Fig. 5b:  The same as fig. 5a but for $h$ = 7/10, 13/10 and 19/10 (top to bottom).   Over this
range, the top (large scale, degraded resolution) series is close to a white noise (left) and Brownian
motion (right).  For the bottom series, there is no equivalent fGn or fBm process, the curves become
smoother although the rescaling may hide this somewhat (see for example the middle $h$ = 13/20 set,
the blow-up of the far right 1/8 of the second series from the top shown in the third line).  Also note
for the bottom two sets with $1 < h < 2$, the oscillations that have frequency $2\pi / \sin(\pi / h)$, this is
the fractional oscillation range.

h=1/10

h=3/10

h=5/10

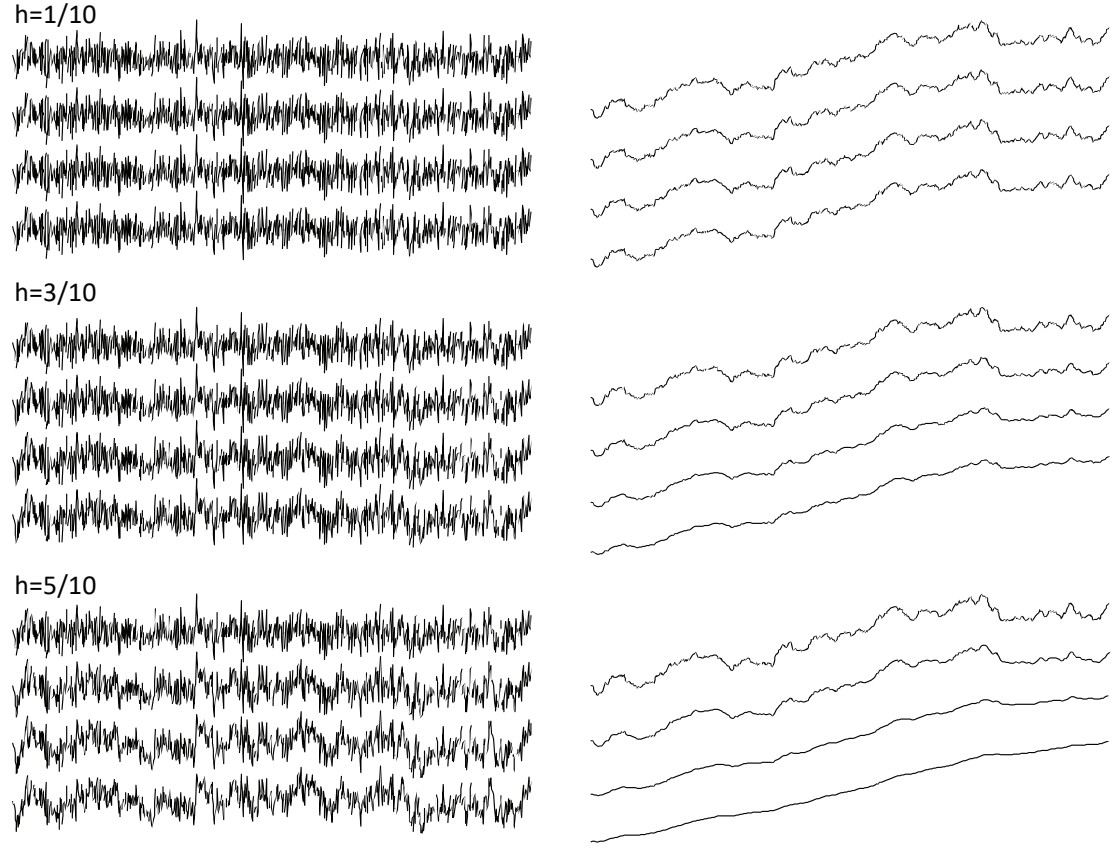

Fig. 6a: This set of simulations is similar to fig. 5a (fRn on the left, fRm on the right) except
that instead of making a large simulation and then degrading and zooming, all the simulations were
of equal length ($2^{10}$ points), but resolutions $\tau = 2^{-15}$, $2^{-10}$, $2^{-5}$, 1 (bottom to top). The simulations
therefore spanned the ranges of scale $2^{-15}$ to $2^{-5}$; $2^{-10}$ to 1; $2^{-5}$ to $2^{5}$; 1 to $2^{10}$ and the same random
seed was used in each so that we can see how the structures slowly change when the relaxation
scale changes.    The bottom fRn, $h=$ 5/10 set is the closest to that observed for the Earth's
temperature, and since the relaxation scale is of the order of a few years, the second series from the
top of this set (with one pixel = one month) is close to that of monthly global temperature anomaly
series. In that case the relaxation scale would be 32 months and the entire series would be $2^{10}/12 \approx$
85 years long.
The top series (of total length $2^{10}$ relaxation times) is (nearly) a white noise (left), and
Brownian motion (right), and the bottom is (nearly) an fGn (left) and fBm (right). The total range
of scales covered here ($2^{10}x2^{15}$) is larger than in fig. 5a and allows one to more clearly distinguish
the high and low frequency regimes.

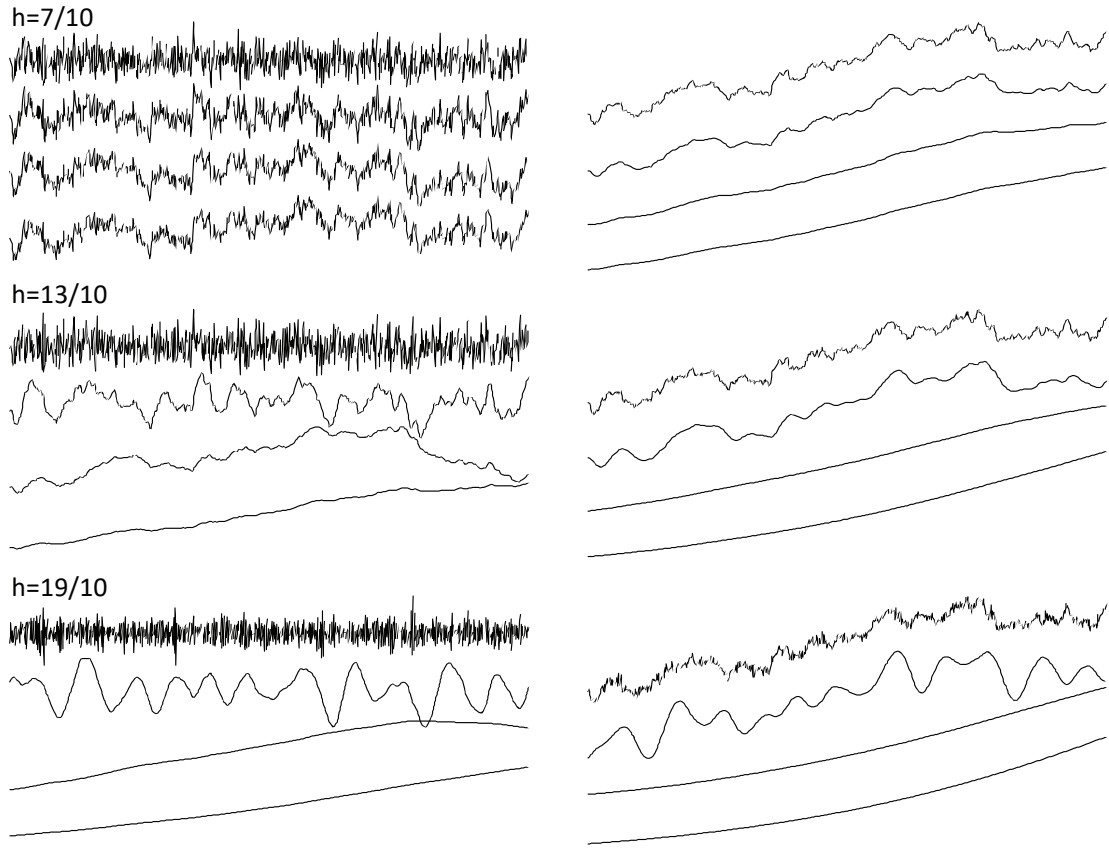

h=7/10

h=13/10

h=19/10

Fig. 6b: The same fig. 6a but for larger *h* values; see also fig. 5b.

## 4. Prediction


The initial value for Weyl fractional differential equations is effectively at $t = -\infty$,
so that for fRn, it is not directly relevant at finite times (although the ensemble mean is
assumed $= 0$; for fRm, the initial condition $Q_{\alpha,h}(0) = 0$ is important). The prediction
problem is thus to use past data (say, for $t < 0$) in order to make the most skillful prediction
for $t > 0$. We are therefore dealing with a *past value* rather than a usual *initial value*
problem. The emphasis on past values is particularly appropriate since in the fGn limit,
the memory is so large that values of the series in the distant past are important. Indeed,
prediction of fGn with a finite length of past data involves placing strong (mathematically
singular) weights on the most ancient data available (see [*Gripenberg and Norros*, 1996],
[*Del Rio Amador and Lovejoy*, 2019], [*Del Rio Amador and Lovejoy*, 2021a], [*Del Rio
Amador and Lovejoy*, 2021b]). This is quite different from standard stochastic predictions
that are based on short memory (exponential) auto-regressive or moving average type
processes that are not much different from initial value problems.
To deal with the small scale divergences when $0 < h+\alpha \leq 1/2$ it is necessary to
predict the finite resolution fRn: $U_{\alpha,h,\tau}(t)$. Using eq. 40 for $U_{\alpha,h,\tau}(t)$, we have:

$$U_{\alpha,h,\tau}(t) = \frac{1}{\tau}\left[\int_{-\infty}^{t} G_{1+\alpha,h}(t-v)\gamma(v)dv - \int_{-\infty}^{0} G_{1+\alpha,h}(-v)\gamma(v)dv\right] -$$

$$\frac{1}{\tau}\left[\int_{-\infty}^{t-\tau} G_{1+\alpha,h}(t-\tau-v)\gamma(v)dv - \int_{-\infty}^{0} G_{1+\alpha,h}(-v)\gamma(v)dv\right] \quad . \qquad (62)$$

$$= \frac{1}{\tau}\left[\int_{-\infty}^{t} G_{1+\alpha,h}(t-v)\gamma(v)dv - \int_{-\infty}^{t-\tau} G_{1+\alpha,h}(t-\tau-v)\gamma(v)dv\right]$$

Now define the predictor for $t \geq 0$ (indicated by a circonflex):
$$\widehat{U_{\alpha,h,\tau}}(t) = \frac{1}{\tau}\left[\int_{-\infty}^{0} G_{1+\alpha,h}(t-v)\gamma(v)dv - \int_{-\infty}^{0} G_{1+\alpha,h}(t-\tau-v)\gamma(v)dv\right]. \qquad (63)$$

To show that it is indeed the optimal predictor, consider the predictor error $E_\tau(t)$:

$$E_\tau(t) = U_{\alpha,h,\tau}(t) - \widehat{U_{\alpha,h,\tau}}(t) = \tau^{-1}\left[\int_{-\infty}^{t} G_{1+\alpha,h}(t-v)\gamma(v)dv - \int_{-\infty}^{t-\tau} G_{1+\alpha,h}(t-\tau-v)\gamma(v)dv\right]$$

$$-\tau^{-1}\left[\int_{-\infty}^{0} G_{1+\alpha,h}(t-v)\gamma(v)dv - \int_{-\infty}^{0} G_{1+\alpha,h}(t-\tau-v)\gamma(v)dv\right] \quad .$$

$$= \tau^{-1}\left[\int_{0}^{t} G_{1+\alpha,h}(t-v)\gamma(v)dv - \int_{0}^{t-\tau} G_{1+\alpha,h}(t-\tau-v)\gamma(v)dv\right]$$

(64)

Eq. 64 shows that the error depends only on $\gamma(v)$ for $v>0$ whereas the predictor (eq. 63)
only depends on $\gamma(v)$ for $v<0$, hence they are orthogonal:
$$\left\langle E_\tau(t)\widehat{U_{\alpha,h,\tau}}(t)\right\rangle = 0, \qquad (65)$$

this is a sufficient condition for $\widehat{U_{\alpha,h,\tau}}(t)$ to be the minimum square predictor which is the
optimal predictor for stationary Gaussian processes, (e.g. [*Papoulis*, 1965]). The prediction
error variance is:
$$\left\langle E_\tau(t)^2\right\rangle = \tau^{-2}\left[\int_{0}^{t-\tau}\left(G_{1+\alpha,h}(t-v) - G_{1+\alpha,h}(t-\tau-v)\right)^2 dv + \int_{t-\tau}^{t} G_{1+\alpha,h}(t-v)^2 dv\right],$$

(66)
or with a change of variables:
$$\left\langle E_\tau(t)^2\right\rangle = \tau^{-2}V_{\alpha,h}(\tau) - \tau^{-2}\left[\int_{t-\tau}^{\infty}\left(G_{1+\alpha,h}(v+\tau) - G_{1+\alpha,h}(v)\right)^2 dv\right], \qquad (67)$$

where we have used $\left\langle U_{\alpha,h,\tau}^2\right\rangle = \tau^{-2}V_{\alpha,h}(\tau)$ (the unconditional variance).
There are numerous skill indicators but the most popular and easy to interpret
definition of forecast skill is the "Minimum Square Skill Score" or "MSSS" (see [*Del Rio*
*Amador and Lovejoy*, 2021a] for discussion of this and other indicators). For this, we
obtain:

$$S_{k,\tau}(t) = 1 - \frac{\left\langle E_\tau(t)^2 \right\rangle}{\left\langle E_\tau(\infty)^2 \right\rangle} = \frac{\int\limits_{t-\tau}^{\infty} \left(G_{1+\alpha,h}(u+\tau) - G_{1+\alpha,h}(u)\right)^2 du}{V_{\alpha,h}(\tau)}$$

$$= \frac{\int\limits_{t-\tau}^{\infty} \left(G_{1+\alpha,h}(v+\tau) - G_{1+\alpha,h}(v)\right)^2 dv}{\int\limits_{0}^{\infty} \left(G_{1+\alpha,h}(v+\tau) - G_{1+\alpha,h}(v)\right)^2 dv + \int\limits_{0}^{\tau} G_{1+\alpha,h}(v)^2 dv}$$

(68)

When $h < 1/2$ and $G_{1,h}(t) = G_{1,h}^{(fGn)}(t) = \dfrac{t^h}{\Gamma(1+h)}$, we obtain the fGn result:

$$S_k = \frac{\xi_h(\infty) - \xi_h(\lambda)}{\xi_h(\infty) + \dfrac{1}{2h+1}} \qquad\qquad \xi_h(\lambda) = \int\limits_{0}^{\lambda-1} \left((v+1)^h - v^h\right)^2 dv \qquad (69)$$

[*Lovejoy et al.*, 2015]. Where $\lambda$ is the forecast horizon (lead time) measured in the number of time steps in the future (due to the fGn scaling, it is independent of the resolution $\tau$). The MSSS gives the fraction of the variance explained by the optimum predictor, when the skill = 1, the forecast is perfect.

To survey the implications, let's start by showing the $\tau$ independent results for fGn, shown in fig. 7 which is a variant on a plot published in [*Lovejoy et al.*, 2015]. We see that when $h \approx 1/2$ ($H \approx 1$) that the skill is very high, indeed, in the limit $h \to 1/2$, we have perfect skill for fGn forecasts (this would of course require an infinite amount of past data to attain).

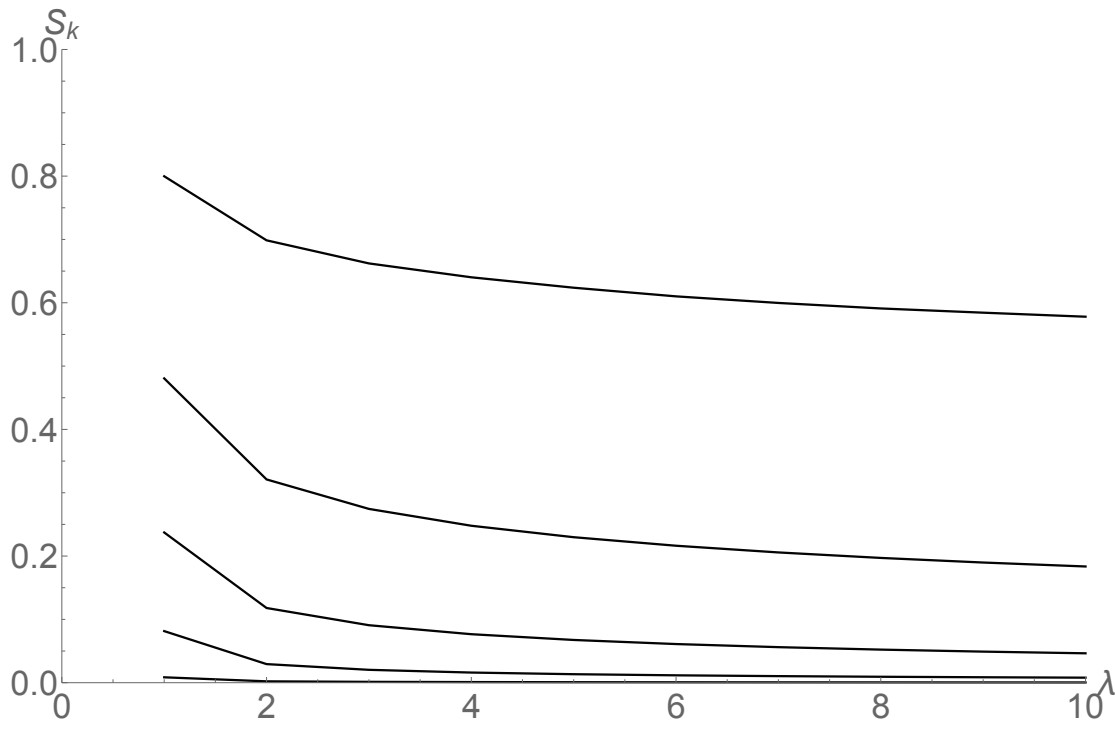

Fig. 7: The prediction skill ($S_k$) for pure fGn processes for forecast horizons up to $\lambda = 10$
steps (ten times the resolution).  This plot is non-dimensional, it is valid for time steps of any
duration.  From bottom to top, the curves correspond to $h = 1/20,\ 3/10,\ \ldots 9/20$ (red, top, close to
the empirical $h$).

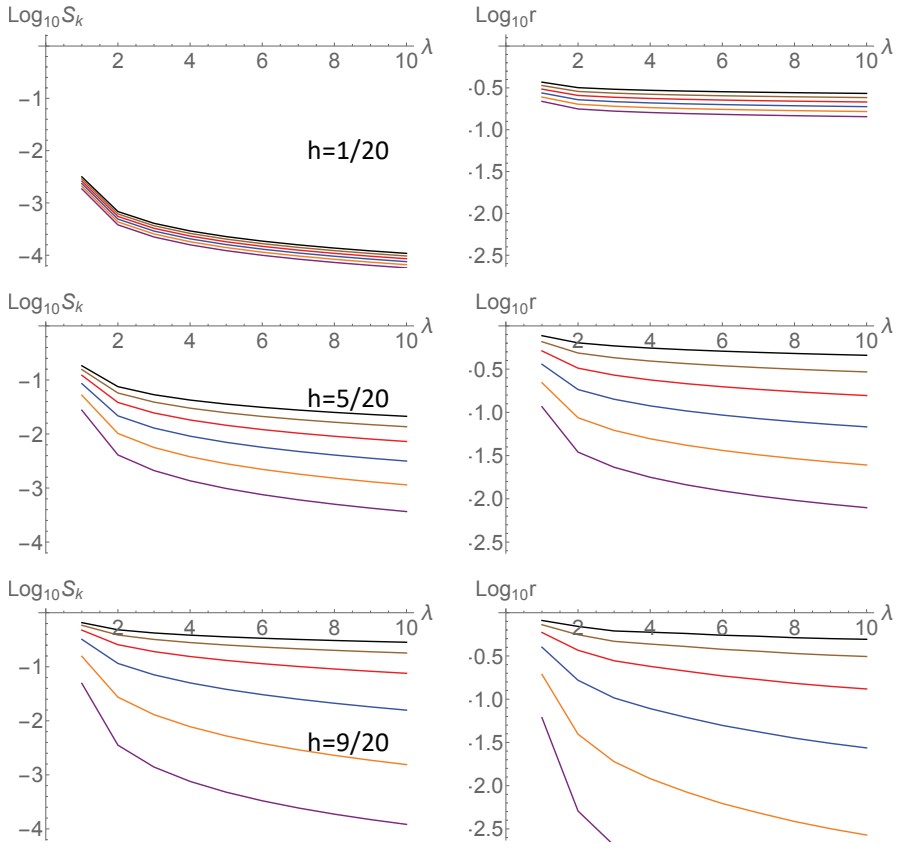

Fig. 8: The left column shows the skill ($S_k$) of pure ($\alpha = 0$) fRn forecasts (as in fig. 7 for fGn) for fRn skill with $h = 1/20, 5/20, 9/20$ (top to bottom set); $\lambda$ is the forecast horizon, the number of steps of resolution $\tau$ forecast into the future. The right hand column shows the ratio ($r$) of the fRn to corresponding fGn skill.

Here the result depends on $\tau$; each curve is for different values increasing from $10^{-4}$ (top, black) to 10 (bottom, purple) increasing by factors of 10 (the red set in the bottom plots with $\tau = 10^{-2}$, $h = 9/20$ are closest to the empirical values).

Now consider the fRn skill, we'll start by considering the pure ($\alpha = 0$) fRn case where the memory comes completely from the (high frequency) storage, anticipating that the fGn forced case ($\alpha \neq 0$) obtains its memory and skill from both storage and the forcing. In comparison with fGn, fRn has an extra parameter, the resolution of the data, $\tau$. Figure 8 shows curves corresponding to fig. 7 for fRn with forecast horizons integer multiples ($\lambda$) of $\tau$ i.e. for times $t = \lambda\tau$ in the future, but with separate curves, one for each of five $\tau$ values increasing from $10^{-4}$ to 10 by factors of ten. When $\tau$ is small, the results should be close to those of fGn, i.e. with potentially high skill, and in all cases, the skill is expected to vanish quite rapidly for $\tau > 1$ since in this limit, fRn becomes an (unpredictable) white noise (although there are scaling corrections to this).

To better understand the fGn limit, it is helpful to plot the ratio of the fRn to fGn skill (fig. 8, right column). We see that even with quite small values $\tau = 10^{-4}$ (top, black curves), that some skill has already been lost. Fig. 9 shows this more clearly, it shows one time step and ten time step skill ratios. To put this in perspective, it is helpful to compare this using

some of the parameters relevant to macroweather forecasting. According to [*Lovejoy et al.*,
2015] and [*Del Rio Amador and Lovejoy*, 2019], the relevant empirical Haar exponent is ≈
-0.1 for the global temperature so that $h = 1/2 - 0.1 ≈ 0.4$. Although direct empirical
estimates of the relaxation time, are difficult since the responses to anthropogenic forcing
begin to dominate over the internal variability after ≈10 years [*Procyk et al.*, 2022] have
used the deterministic response to estimate a global relaxation time of ≈ 5 years (work in
progress using maximum likelihood estimates shows that a scales of hundreds of kilometers,
it is quite variable ranging from months to decades [*Procyk*, 2021]). For monthly resolution
forecasts, the non-dimensional resolution is $τ ≈ 1/100$. With these values, we see (red
curves) that we may have lost ≈ 30% of the fGn skill for one month forecasts and ≈ 85%
for ten month forecasts. Comparing this with fig. 7 we see that this implies about 60% and
10% skill (see also the red curve in fig. 8, bottom set).
Going beyond the $0 < h < 1/2$ region that overlaps fGn, fig. 9, 10 clearly shows that
the skill continues to increase with $h$. We already saw (fig. 4) that the range $1/2 < h < 3/2$
has RMS Haar fluctuations that for $Δt < 0$ mimic fBm and these do indeed have higher skill,
approaching unity for $h$ near 1 corresponding to a Haar exponent ≈ 1/2, i.e. close to an fBm
with $H = 1/2$, i.e. a regular Brownian motion. Recall that for Brownian motion, the
increments are unpredictable, but the process itself is predictable (persistence). In figure
9, we show the skill for various $h$'s as a function of resolution $τ$. Fig. 11a shows that for $h$
$< 3/2$, the skill decreases rapidly for $τ > 1$. Fig. 11b in the fractional oscillation equation
regime shows that the skill oscillates.
We may now consider the skill of the fGn forced process ($α ≠ 0$), fig. 12. For small
$τ$, short lags, $λ$ (the upper left), the contours are fairly linear along lines of constant $h+α$,
so that as expected, the predictability is essentially that of an fGn process but with effective
exponent $h+α$. At the opposite extreme (large $τ$, $h$, the lines are fairly horizontal, indicating
that the skill from the storage (i.e. from $h$) is negligible, and that all the memory (and hence
skill) comes from the forcing fGn, exponent $α$. The in-between resolutions and lags
generally have in-between slopes. As expected, the skill from the storage drops off quickly
for resolutions $≈> τ$. For $h≥1$, there is some waviness in the contours due to the oscillatory
nature of the Green's functions.

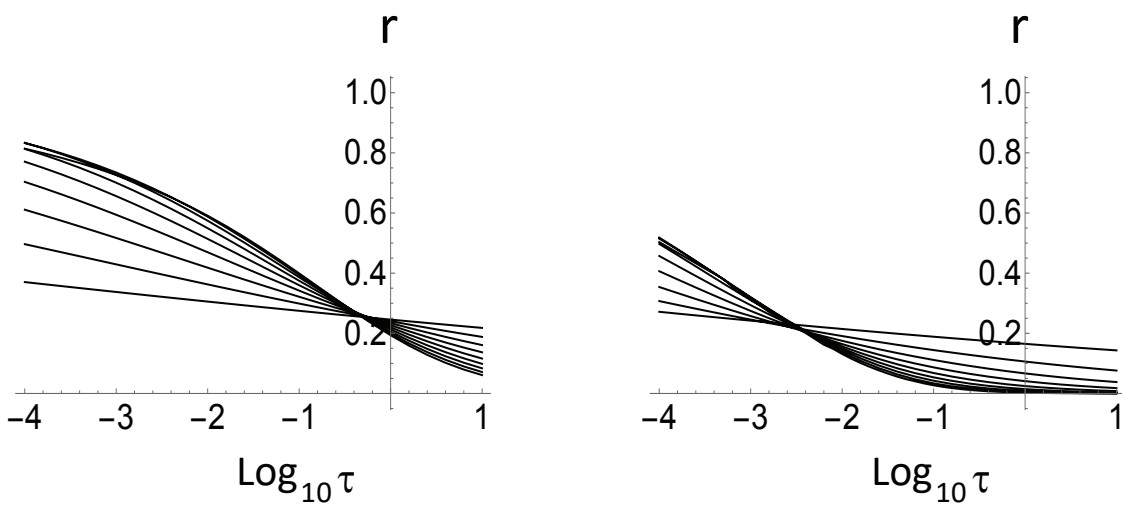

Fig. 9:  The ratio of (α = 0) fRn skill to fGn skill (left: one step horizon, right: ten step
forecast horizon) as a function of resolution τ for *h* increasing from (at left) bottom to top (*h* = 1/20,
2/20, 3/20…9/20); the *h* = 9/20 curves (close to the empirical value) is the curve that starts at the
left of each plot.

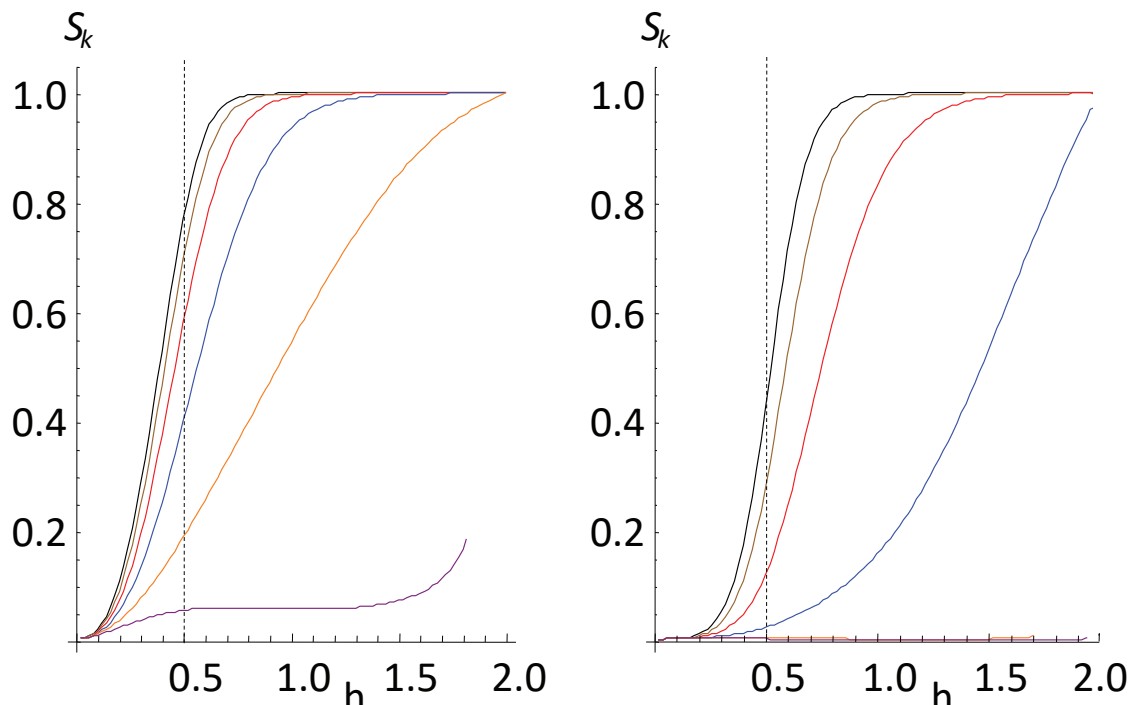

Fig. 10: The one step (left) and ten step (right) pure ($\alpha = 0$) fRn forecast skill as a function
of $h$ for various resolutions ($\tau$) ranging from $\tau = 10^{-4}$ (black, left of each set) through $\tau = 10^{-3}$
(brown) $10^{-2}$ (red), 0.1 (blue), 1 (orange), 10 (purple). In the right set $\tau = 1$ (orange), 10 (purple)
lines are nearly on top of the $S_k = 0$ line. Again red ($\tau = 10^{-2}$) is the more empirical relevant value
for monthly data. Recall that the regime $h < 1/2$ (to the left of the vertical dashed lines) corresponds
to the overlap with fGn.

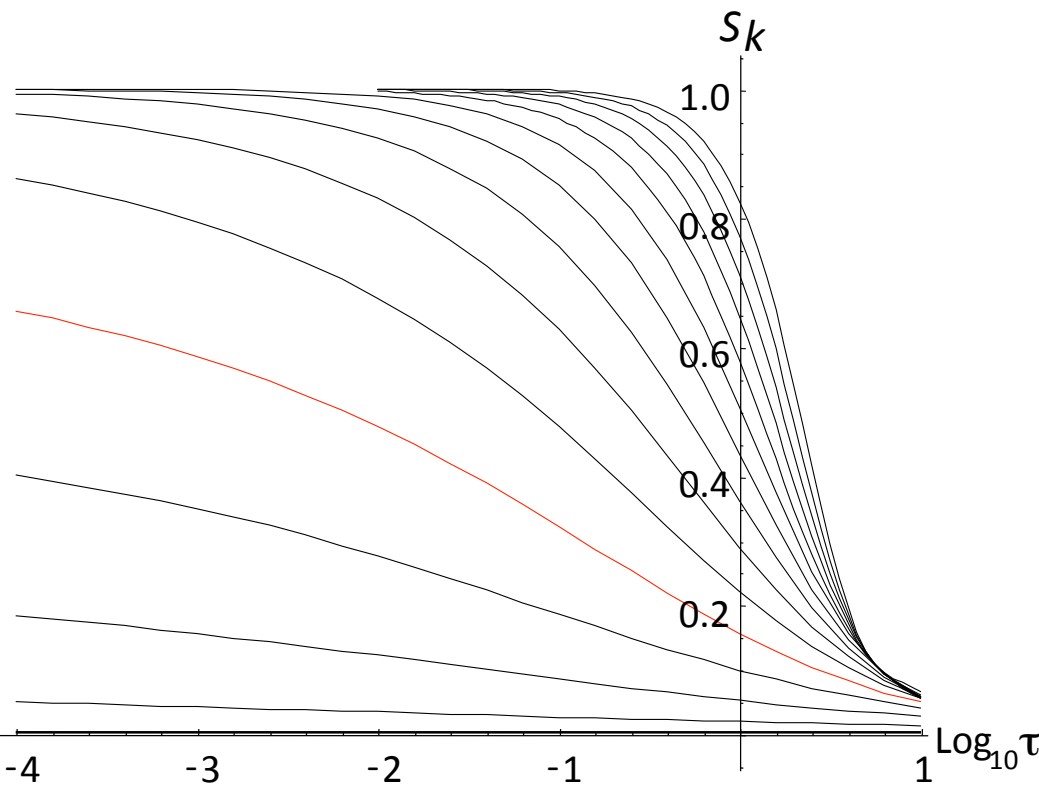

Fig. 11a:  One step pure ($\alpha = 0$)  fRn prediction skills as a function of resolution for $h$'s
increasing from 1/20 (bottom) to 29/20 (top), every 1/10.  Note the rapid transition to low skill,
(white noise) for $\tau > 1$.  The curve for $h = 9/20$ is shown in red.

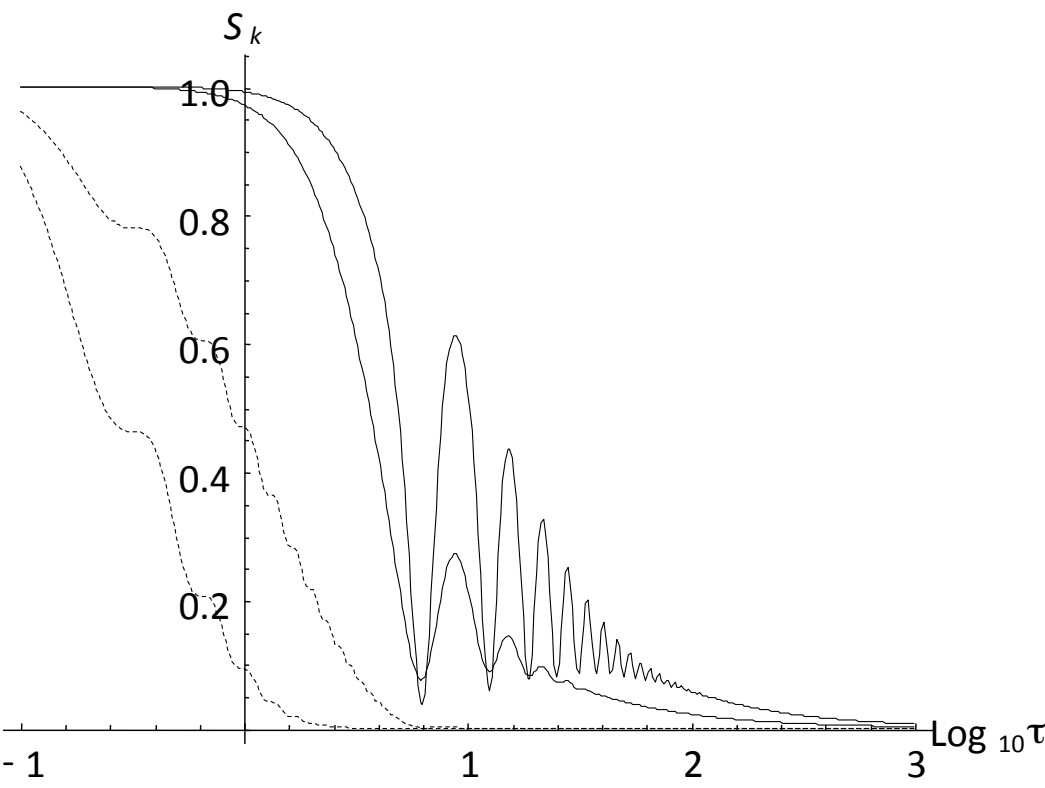

Fig. 11b: Same as fig. 11a except for $h = 37/20$, $39/20$ showing the one step skill (black),
and the ten step skill (dashed).  The right hand dashed and right hand solid lines, are for $h = 39/20$,
they clearly show that the skill oscillates in this fractional oscillation equation regime.   The
corresponding left lines are for $h = 37/20$.

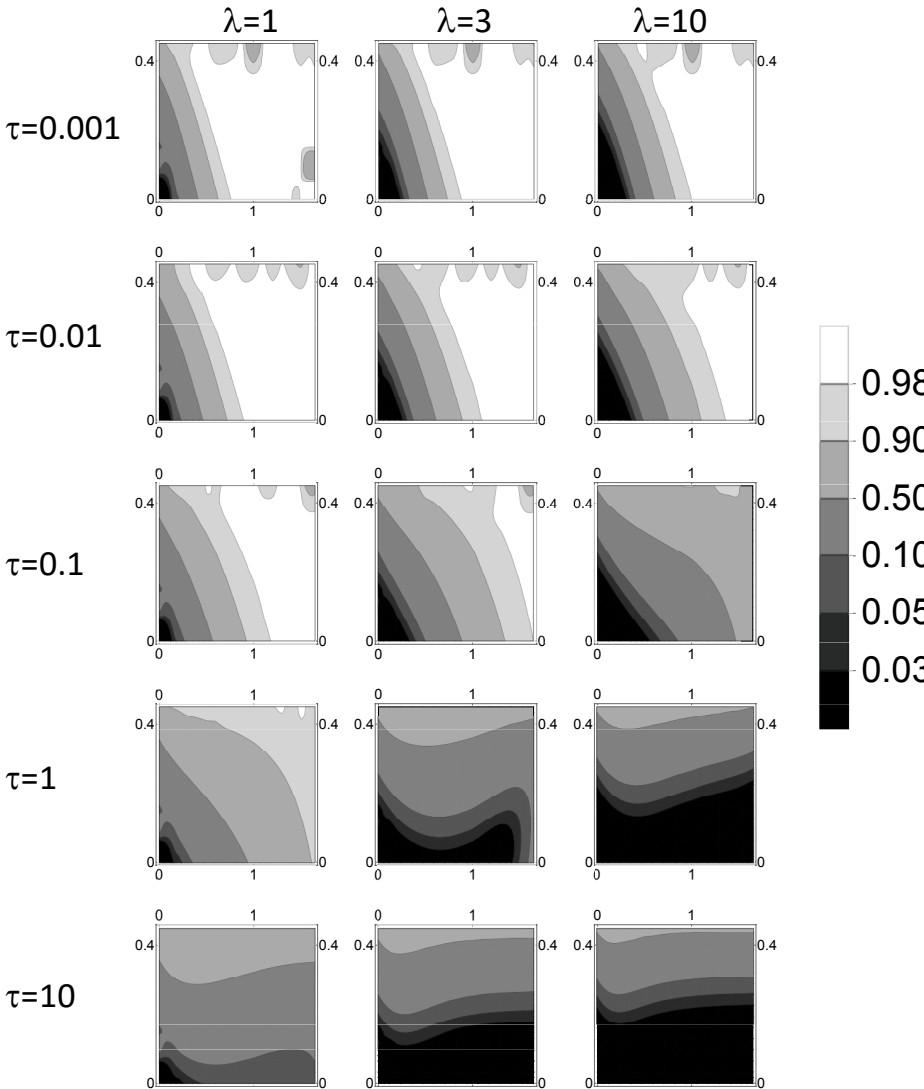

Fig. 12: Contour plots of the forecast skill, with *h* along the horizontal and α along the vertical axis.
The plots are for increasing nondimensional resolutions: τ = 0.001, 0.01, 0.1, 1, 10 (top to bottom), with
forecasts for lags λ = 1, 3, 10 (left to right) and with contour levels (legend) varying from nearly no skill
(0.03), to nearly full skill (0.98).
**4. Conclusions:**
Ever since [*Budyko*, 1969] and [*Sellers*, 1969], the energy balance between the earth
and outer space has been modelled by the Energy Balance Equation (EBE), based on the
continuum heat equation, see [*North and Kim*, 2017] for a recent review and see [*Ziegler*
*and Rehfeld*, 2020] for a recent regional application).  It is most commonly used as a model
for the globally averaged temperature where it is usually derived by applying Newton's
law of cooling applied to a uniform slab of material, a "box".  The resulting EBE is a first
order relaxation equation describing the exponential relaxation of the temperature to a new
equilibrium after it has been perturbed by an external forcing.  Its first order (*h* = 1)
derivative term  accounts for energy storage.
The resulting model relaxes to equilibrium much too quickly so that to increase
realism, it is usual to introduce a few interacting slabs (representing for example the
atmosphere and ocean mixed layer; the Intergovernmental Panel on Climate Change
recommends two such components [*IPCC*, 2013]).   However, it turns out that these $h = 1$
box models do not use the correct surface radiative-conductive boundary conditions.  If
one assumes heat transport by the classical heat equation and radiative-conductive
boundary conditions are used instead, one instead obtains the Half-order EBE, the HEBE
with $h = 1/2$ [*Lovejoy*, 2021a; b] which is already close to the global empirical value ($h =$
0.38±0.03, [*Procyk et al.*, 2022], [*Del Rio Amador and Lovejoy*, 2019], see also [*Lovejoy*
*et al.*, 2015]).  However this model is only valid in the macroweather regime - for time
scales of weeks and longer and due to the spatial scaling in the atmosphere, the fractional
heat equation (FHE) may be more a more appropriate model than the classical one.  The
use of the FHE can be justified by recognizing that a realistic energy transport model
involves a continuous hierarchy of mechanisms. The extension to the FHE leads directly
to a fractional relaxation equation that generalizes the EBE: the Fractional Energy Balance
Equation [*Lovejoy*, 2021a; b] (FEBE).  The FEBE can also be derived phenomenologically
by assuming that energy storage processes are scaling, [*Lovejoy*, 2019a;  2019b; *Lovejoy*
*et al.*, 2021]).
When forced by a Gaussian white noise, the FEBE is also a generalization of
fractional Gaussian noise (fGn) and its integral (fractional Relaxation motion, fRm),
generalizes fractional Brownian motion (fBm).  More classically, it generalizes the
Orenstein-Uhlenbeck process that corresponds to the $h =1$ special case (i.e. the standard
EBE with white noise forcing).  Over the parameter range $0 < h < 1/2$, the high frequency
FEBE limit (fGn) has been used as the basis of monthly and seasonal temperature forecasts
[*Lovejoy et al.*, 2015], [*Del Rio Amador and Lovejoy*, 2019; *Del Rio Amador and Lovejoy*,
2021a; *Del Rio Amador and Lovejoy*, 2021b]; at one month lead times, these macroweather
forecasts are similar in skill to conventional numerical models whereas for bimonthly,
seasonal and annual forecasts they are more skillful [*Del Rio Amador and Lovejoy*, 2021a].
For multidecadal time scales the low frequency limit has been used as the basis of climate
projections through to the year 2100 [*Hebert*, 2017], [*Lovejoy et al.*, 2017], [*Hébert et al.*,
2021], and more recently, the full FEBE has been used directly [*Procyk et al.*, 2020],
[*Procyk*, 2021], [*Procyk et al.*, 2022].
It was the success of predictions and projections with different exponents but
theoretically derived the same empirical underlying FEBE $h \approx 0.4$, that over the last years,
motivated the development of the FEBE (announced in [*Lovejoy*, 2019a]) and the work
reported here.  The statistical characterizations – correlations, structure functions, Haar
fluctuations and spectra as well as the predictability properties are important for these and
other FEBE applications and are derived in this paper.
While the deterministic fractional relaxation equation is classical, various technical
difficulties arise when it is generalized to the stochastic case: in the physics literature, it is
a Fractional Langevin Equation (FLE) that has almost exclusively been considered as a
model of diffusion of particles starting at an origin.  This requires  $t = 0$ initial conditions
that imply that the solutions are strongly nonstationary. In comparison, the Earth's
temperature fluctuations that are associated with its internal variability are statistically
stationary.  This can easily be modelled with initial conditions at $t = -\infty$ i.e. by using Weyl
fractional derivatives.  In addition, in the usual FLE, the highest order derivative is an
integer so that sample processes are RMS differentiable order at least one ([*Watkins et al.*,
2020] have called the FEBE a "Fractionally Integrated FLE") . In the FEBE and the
fractionally integrated extensions, the highest order derivative is readily of order <1/2 so
that sample processes are generalized functions ("noises") and must be smoothed/averaged
for physical applications.

Although EBE's were originally developed to understand the deterministic
temperature response to external forcing, the temperature also responds to stochastic
"internal" forcing. While the Earth system variability is generally highly nonGaussian
(multifractal, [*Lovejoy*, 2018]), the temporal macroweather regime modelled here is the
quasi-Gaussian exception. This paper therefore explores the statistics of the temperature
response when it is stochastically forced by Gaussian processes: both by white noise ($\alpha =$
0) and by a (long memory) fractional Gaussian noise (fGn) processes. The white noise
special case –"pure fRn, fRm" - is the $\alpha = 0$ special case, fGn forced case extends the
parameter range to $0 \leq \alpha < 1/2$. According to work in progress using satellite and reanalysis
radiances, both cases appear to be empirically relevant for modelling the Earth's energy
balance.

A key novelty is therefore to consider the fractional relaxation - equation (a
Fractional Langevin Equation, FLE) forced by white and scaling noises starting from
$t = -\infty$: equivalent to Weyl "fractionally integrated fractional relaxation equation"). In
addition, the highest order terms in standard FLE's are integer ordered, the fractional terms
represent damping and are of lower order, guaranteeing that solutions are regular functions.
However, the FEBE's highest order term is fractional and over the main empirically
significant parameter range ($\alpha+h<1/2$) the processes are noises (generalized functions): in
order to represent physical processes, they must be averaged. This is conveniently handled
by introducing their integrals or "motions". We proceeded to derive their fundamental
statistical properties including series expansions about the origin and infinity. These
expansions are nontrivial since they mix fractional and integer ordered terms (Appendix
A). Since the FEBE is used as the basis for macroweather predictions, the theoretical
predictability skill is important in applications and was also derived.

With these stationary Gaussian forcings, the solutions are a new stationary process
– fractional Relaxation noise (fRn, $\alpha =0$) and their extensions to fractionally integrated fRn
processes ($\alpha>0$). Over the range $0 < \alpha + h < 1/2$, we show that the small scale limit is a
fractional Gaussian noise (fGn) – and its integral - fractional Relaxation motion (fRm) -
has stationary increments and which generalizes fractional Brownian motion (fBm).
Although at long enough times, the fRn ($\alpha = 0$) tends to a Gaussian white noise, and fRm
to a standard Brownian motion, this long time convergence is typically very slow (when
$\alpha>0$, the long time behaviours are fGn and fBm processes, parameter $\alpha$).

Much of the effort was to deduce the asymptotic small and large scale behaviours
of the autocorrelation functions that determine the statistics and in verifying these with
extensive numerical simulations. An interesting exception was the $h = 1/2$ special case
which for fGn corresponds to an exactly 1/f noise. Here, we give the exact mathematical
expressions for the full correlation functions, showing that they had logarithmic
dependencies at both small and large scales. The resulting Half order EBE (HEBE) has an
exceptionally slow transition from small to large scales (a factor of a million or more is
needed) and empirically, it is quite close to the global temperature series over scales of
months, decades and possibly longer.
Beyond improved monthly, seasonal temperature forecasts and multidecadal
projections, the stochastic FEBE opens up several paths for future research. One of the
more promising is to apply these techniques to the spatial FEBE and generalize it in various
directions. This is a follow up on the special value $h = 1/2$ that is very close to that found
empirically and that can be analytically deduced from the classical Budyko-Sellers energy
transport equation by improving the mathematical treatment of the radiative boundary
conditions [*Lovejoy*, 2021a; b]. In the latter case, one obtains a partial fractional
differential equation for the horizontal space-time variability of temperature anomalies
over the Earth's surface, allowing regional forecasts and projections. This has already
allowed improved regional projections ([*Procyk*, 2021]) and promises better monthly,
seasonal forecasts.
While the FEBE has already demonstrated its ability to project future climates,
these improvements will allow for the modelling of the nonlinear albedo-temperature
feedbacks needed for modelling of transitions between different past climates. Finally,
FEBE based projections have shown that in spite of improved computer power and
algorithms, that conventional GCM approaches may be suffering from diminishing returns:
the GCMs in the latest IPCC assessment (AR6, 2021) are even more uncertain: a range 2 -
$5.5K/CO_2$ doubling (90% confidence) as those in the previous assessment (AR5, 2013, 1.5
- 4.5K per doubling) while also being somewhat warmer. The FEBE had the somewhat
lower but much less uncertain range $1.6 – 2.4K/CO_2$ doubling (90% confidence).
Conventional GCM approaches attempt to explicitly model as many degrees of freedom as
possible and by the year 2030, they are expected to have kilometric scale ("cloud
resolving") resolutions that will model structures that live for only 15 minutes and then–
average them over decades. The FEBE (with regional and other fututre extensions), is in
contrast, a high level stochastic model that accounts for the collective interactions of huge
numbers of degrees of freedom [*Lovejoy*, 2019a], it is thus a promising candidate for a new
generations of climate models.
**Acknowledgements**
I thank L. Del Rio Amador, R. Procyk, R. Hébert, C. Penland, N. Watkins for
discussions. I also acknowledge an exchange with K. Rypdal. We thank anonymous
referees for suggestions including the fifth referee for encouraging comments on the
Fourier approach. This work was unfunded, there were no conflicts of interest.

## Appendix A: The small and large scale fRn, fRm statistics:

**A.1 $R_{\alpha,h}(t)$ as a Laplace transform**

In section 2.4, we derived general statistical formulae for the auto-correlation functions of motions and noises defined in terms of Green's functions of fractional operators. Since the processes are Gaussian, autocorrelations fully determine the statistics. While the autocorrelations of fBm and fGn are well known those for fRm and fRn are new and are not so easy to deal with since they involve quadratic integrals of Mittag-Leffler functions. In this appendix, we derive the basic power law expansions as well as large $t$ (asymptotic) expansions, and we numerically investigate their accuracy.

It is simplest to start with the Fourier expression for the autocorrelation function for the unit white noise forcing (eq. 33). First convert the inverse Fourier transform (eq. 66) into a Laplace transform. For this, consider the integral over the contour $C$ in the complex plane:

$$I_C(t) = \frac{1}{2\pi} \int_C \frac{e^{zt}}{z^\alpha (-z)^\alpha \left(1+z^h\right)\left(1+(-z)^h\right)} dz \tag{A.1}$$

Take $C$ to be the closed contour obtained by integrating along the imaginary axis (this part gives $R_{\alpha,h}(t)$, eq. 33), and closing the contour along an (infinite) semicircle over the second and third quadrants. When $0<h<1$, there are no poles in these quadrants, but we must integrate around a branch cut on the negative real axis. When $1<h<2$, we must take into account two new branch cuts and two new poles in the negative real half plane. In a polar representation $z = re^{i\theta}$, the additional branch cuts are along the rays $z = re^{\pm i\pi/h}$; $r>1$, circling around the poles at $z = e^{\pm i\pi/h}$. The additional branch cuts give no net contribution, but the residues of the poles do make a contribution ($P_{\alpha,h} \neq 0$ below). We can express both cases with the formula:

$$R_{\alpha,h}(t) = -\frac{1}{\pi} \text{Im} \int_0^\infty \frac{e^{-xt} dx}{x^{2\alpha} e^{i\alpha\pi} \left(1+x^h\right)\left(1+x^h e^{i\pi h}\right)} + P_{\alpha,h,+}(t); \quad t>0$$

$$\tag{A.2}$$

"Im" indicates the imaginary part and:

$$P_{\alpha,h,\pm}(t) = 0; \qquad\qquad 0 < h < 1$$

$$P_{\alpha,h,\pm}(t) = -e^{t\cos\left(\frac{\pi}{h}\right)} \frac{\sin\left(\pm\frac{\pi}{h}(1-\alpha) + \frac{h\pi}{2} + t\sin\left(\frac{\pi}{h}\right)\right)}{h\sin\left(\frac{\pi h}{2}\right)}; \quad 1 < h < 2 \qquad 0 \leq \alpha < 1/2$$

$$\tag{A.3}$$

While the integral term is monotonic, the $P_{\alpha,h}$ term oscillates with frequency $\omega = 2\pi / \sin(\pi / h)$. $P_{\alpha,h}$ accounts for the oscillations visible in figs. 2, 3, 5b although since

when $1 < h < 2$, $\cos(\pi/h) < 1$, they decay exponentially. When $h > 1$, this pole contribution
dominates $R_{\alpha,h}(t)$ for a wide range of $t$ values around $t = 1$, although as we see below,
eventually at large $t$, power law terms come to the fore.
Comments:
a) When $\alpha = 0$, $h = 1$, we obtain the classical Ornstein-Uhlenbeck autocorrelation:
$R_{0,1}(t) = \frac{1}{2} e^{-|t|}$.
b) In the case $h = 0$, the process reduces to an fGn process:
$R_{\alpha,0}(t) = t^{-1+2\alpha} \Gamma(1-2\alpha) \sin(\pi\alpha)/(4\pi)$. There is an extra factor of 4 that comes from the
small $h$ limit $_{-\infty}D_t^h + 1 \to 2$.

## A.2 Asymptotic expansions:

An advantage of writing $R_{\alpha,h}(t)$ as a Laplace transform is that we can use Watson's
lemma to obtain an asymptotic expansion (e.g. [*Bender and Orszag*, 1978]). The idea is
that an expansion of eq. A.2 around $x = 0$ can be Laplace transformed term by term to yield
an asymptotic expansion for large $t$.
The expansion of the integrand around $x = 0$ can be obtained from a binomial
expansion (see also A.10):

$$\frac{1}{x^{2\alpha} e^{i\pi\alpha}\left(1+x^h\right)\left(1+x^h e^{i\pi h}\right)} = \frac{e^{-i\pi\alpha}}{e^{i\pi h}-1} \sum_{n=0}^{\infty} (-1)^n \left(e^{i(n+1)\pi h} - 1\right) x^{-2\alpha+nh}; \quad x < 1$$

(A.4)
this leads to:
$$-\frac{1}{\pi} \operatorname{Im} \frac{1}{x^{2\alpha} e^{i\alpha\pi}\left(1+x^h\right)\left(1+x^h e^{hi\pi}\right)} = -\sum_{n=0}^{\infty} D_{-n} x^{nh-2\alpha} \tag{A.5}$$

$$D_n = (-1)^{n+1} \frac{\cos\left(\left(n-\frac{1}{2}\right)\pi h + \alpha\pi\right) - \cos\left(\frac{\pi h}{2} + \alpha\pi\right)}{2\pi \sin\left(\frac{\pi h}{2}\right)} = (-1)^n \frac{\sin\left(\frac{n\pi h}{2} + \alpha\pi\right)\sin\left(\frac{(n-1)\pi h}{2}\right)}{\pi \sin\left(\frac{\pi h}{2}\right)}$$

(note $D_{-n}$ is used in the expansion here; $D_n$ is used below).
Therefore, taking the term by term Laplace transform and using Watson's lemma:
$$R_{\alpha,h}(t) = -\sum_{n=0}^{\infty} D_{-n} \Gamma(1+nh-2\alpha) t^{2\alpha-(1+nh)} + P_{\alpha,h,+}(t); \qquad t \gg 1$$

$(0 < \alpha < 1/2)$.     (A.6)
Where we have included the exponentially decaying residue $P_{\alpha,h,+}$ that contributes when
$1 < h < 2$. Note that although $\Gamma$ diverges for all negative integer arguments, using the identity
$\Gamma(1+hn-2\alpha) \sin\left((nh-2\alpha)\pi\right) = -\pi/\Gamma(2\alpha-nh)$     we     see     that     the     product
$\sin\left((nh-2\alpha)\pi\right)\Gamma(2\alpha-nh)$ is finite.
The first terms are explicitly:
$$R_{\alpha,h}(t) = \frac{\Gamma(1-2\alpha)\sin(\pi\alpha)}{\pi}t^{2\alpha-1} - \frac{\cos\left(\frac{\pi h}{2}\right)}{\cos\left(\frac{\pi h}{2}-\pi\alpha\right)\Gamma(2\alpha-h)}t^{2\alpha-(1+h)} + ...$$

$t \gg 1$          (A.7)

We see that when $\alpha \neq 0$, $D_0 > 0$ so that as expected, the leading behaviour has no $h$
dependence, it is only due to the long range correlations in the forcing; we obtain the fGn
result: $t^{2\alpha-1}$. However for the pure fRn case, $\alpha = 0$ and $D_0 = 0$ so that we obtain:
$$R_{0,h}(t) = \sum_{n=1}^{\infty}(-1)^n \frac{1+\cot\left(\frac{\pi h}{2}\right)\tan\left(\frac{n\pi h}{2}\right)}{2\Gamma(-nh)}t^{-(1+nh)} + P_{0,h,+}(t); \quad t \gg 1 \qquad (A.8)$$

i.e. the leading behaviour is $t^{-(1+h)}$. Note that the leading $n = 1$ coefficient reduces to
$-1/\Gamma(-h)$ and that for $0 < h < 1$, $\Gamma(-h) < 0$.
For the motions (fRm), we need the expansion of $V_{\alpha,h}(t)$, it can be obtained by
integrating $R_{\alpha,h}$ twice (using eq. 36):
$$V_{\alpha,h}(t) = a_{\alpha,h}t + b_{\alpha,h} - 2\sum_{n=0}^{\infty}D_{-n}\Gamma(-1+nh-2\alpha)t^{2\alpha+1-nh} + 2P_{\alpha,h,-}(t); \quad t \gg 1 \qquad 0 \leq \alpha < 1/2$$

(A.9)
Where $P_{a,h-}$ is from the poles when $1 < h < 2$. Since the asymptotic expansion is not valid for
$t = 0$, we used the indefinite integrals of $R_{\alpha,h}$ hence there is a linear $a_{\alpha,h}t + b_{\alpha,h}$ term from
the constants of integration. However, when $\alpha > 0$, the leading term is the $t^{2\alpha+1}$ term from
the fGn forcing and in the pure fRn case ($\alpha = 0$), we can take $\lim_{\alpha \to 0}\left(-2D_0\Gamma(-1-2\alpha)t^{2\alpha+1}\right) = t$
so that the leading term $n = 0$ already gives the correct fRm behaviour: $V_{\alpha,h}(t) \approx t$ so that
$a_{0,h} = 0$ ($b_{0,h}$ can be determined numerically).

## A.3 Power series expansions about the origin:

For many applications one is interested in the behavior of $R_{\alpha,h}(t)$ for scales of
months which is typically less than the relaxation time, i.e. $t < 1$. It is therefore important
to understand the small $t$ behaviour. We again consider the Laplace integral for the $0 < h < 1$
case. In this case, we can divide the range of integration in eq. A2 into two parts for $0 < x < 1$
and $x > 1$. For the former, we use the expansion in eq. A4 and for the latter:
$$\frac{1}{x^{2\alpha}e^{i\pi\alpha}(1+x^h)(1+x^h e^{i\pi h})} = \frac{e^{-i\pi\alpha}}{e^{i\pi h}-1}\sum_{n=1}^{\infty}(-1)^{n+1}\left(e^{-i(n-1)\pi h}-1\right)x^{-2\alpha-nh}; \quad x > 1 \quad (A.10)$$

We can now integrate each term seperately using:

$$\int_0^1 e^{-xt}x^{nh-2\alpha}\,dx = \sum_{j=1}^{\infty}\frac{(-1)^{j-1}}{(hn-2\alpha+j)\Gamma(j)}t^{j-1}$$


$$\int_1^{\infty} e^{-xt}x^{-(nh+2\alpha)}\,dx = E_{nh+2\alpha}(t) = \pi\frac{t^{-1+hn+2\alpha}}{\sin(\pi nh+2\pi\alpha)\Gamma(hn+2\alpha)}+\sum_{j=1}^{\infty}\frac{(-1)^{j-1}}{(hn+2\alpha-j)\Gamma(j)}t^{j-1}$$

(A.11)
where $E_{\beta}(t)=\int_1^{\infty}e^{-xt}x^{-\beta}\,dx$ is the exponential integral. Adding the two integrals and
summing over $n$, we obtain:
$$R_{\alpha,h}(t) = \sum_{n=2}^{\infty}D_n\Gamma(1-hn-2\alpha)t^{-1+hn+2\alpha}+\sum_{j=1}^{\infty}F_j\frac{t^{j-1}}{\Gamma(j)} \qquad (A.12)$$

$$F_j = \frac{1}{\pi h}\text{Im}\left[\frac{e^{-i\alpha\pi}}{e^{i\pi h}-1}\left(e^{i\pi h}\sum_{n=-\infty}^{\infty}(-1)^n\frac{e^{i\pi nh}}{(n+a)}-\sum_{n=-\infty}^{\infty}(-1)^n\frac{1}{(n+a)}\right)\right]; \quad a=\frac{j-2\alpha}{h}$$

(we have interchanged the order of summations and used $D_n$ from eq. A5 with $n>0$).
The series for the coefficient $F_j$ can now be summed analytically. Although the
sum is a special case of the Lipchitz summation and Poisson summation formulae, the
easiest method is to use the Sommerfeld-Watson transformation (e.g. [*Mathews and*
*Walker*, 1973]) that converts an infinite sum into a contour integral that is then deformed.
The Sommerfeld-Watson transformation states that for a an analytic function $f(z)$ that goes
to zero at least as fast as $|z|^{-1}$, that:
$$\sum_{n=-\infty}^{\infty}(-1)^n f(n) = -\pi\sum_k\frac{R_k}{\sin\pi z_k} \qquad (A.13)$$

Where $z_k$ is the location of the poles of $f(z)$ and $R_k$ is the residue of the corresponding pole.
In the above, take:

$$f(z)=\frac{e^{iz\pi h}}{(z+a)}$$

(A.14)
There is a single pole at $z_1 = -a$ and the residue is $R_1 = e^{-ia\pi h}$, therefore:

$$e^{i\pi h}\sum_{n=-\infty}^{\infty}\frac{(-1)^n e^{in\pi h}}{(n+a)}=\pi\frac{e^{i\pi h(1-a)}}{\sin\pi a}$$

(A.15)
The second sum needed in $F_j$ can be obtained using $h = 0$ in the above so that
overall:

$$F_j = \frac{1}{h\pi}\text{Im}\left[\frac{e^{-i\alpha\pi}}{e^{i\pi h}-1}\left(\pi\frac{e^{i\pi h(1-a)}-1}{\sin\pi a}\right)\right]=\frac{1}{h\sin(\pi(j-2\alpha)/h)}\text{Im}\left[\frac{e^{-i\pi j}e^{i\pi(h/2+\alpha)}-e^{-i\pi(h/2+\alpha)}}{e^{i\pi h/2}-e^{-i\pi h/2}}\right]$$

(A.16)
If $j$ is even, then the term in the square bracket is pure real hence $F_j$ vanishes.
Otherwise:
$$F_j = -\frac{\cos\pi\left(\dfrac{h}{2}+\alpha\right)}{h\sin\left(\dfrac{\pi h}{2}\right)\sin\left(\dfrac{\pi}{h}(j-2\alpha)\right)}; \quad j = odd \tag{A.17}$$

Note that $F_1>0$ for $h + \alpha >1/2$ (with $0\leq\alpha<1/2$, $0\leq h<2$), whereas for $h + \alpha <1/2$ it is quite
complicated (see below).

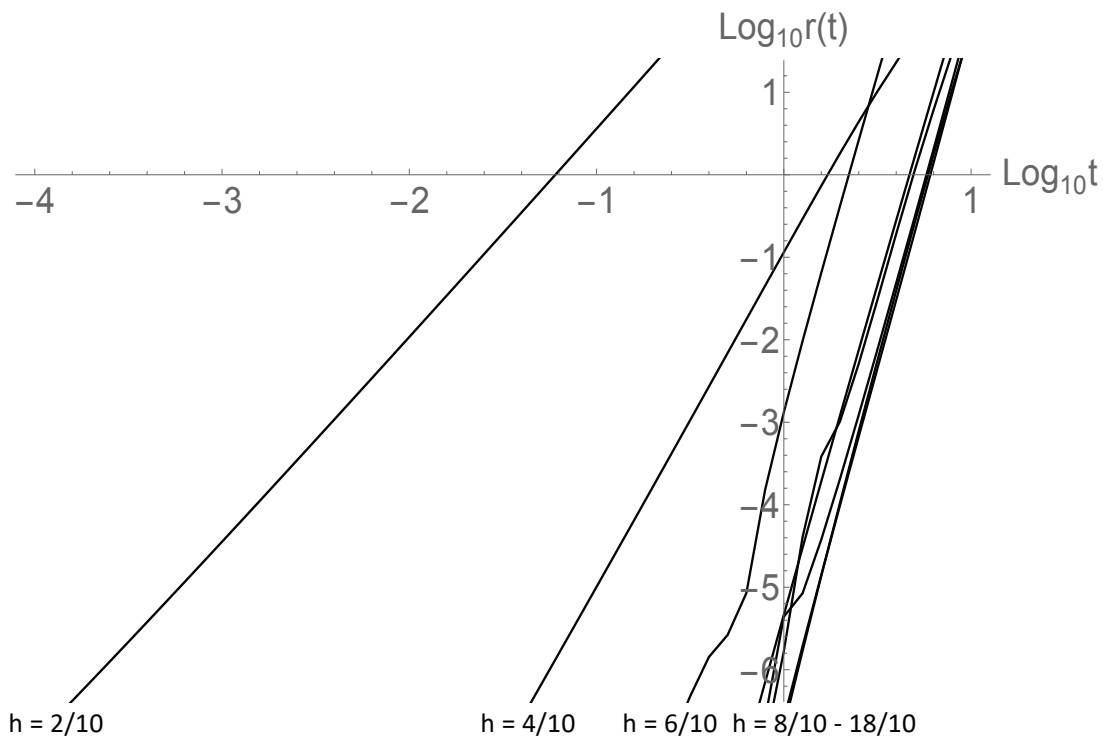

Fig. A1: This shows the logarithm of the relative error in the $R_{0,h}^{(10,10)}(t)$ approximation (i.e.
with 10 fractional terms and 10 integer order terms) with respect to the deviation from the fGn
$R_{0,h}(t)$   $r = \log_{10}\left|1-\left(R_h^{fGn}(t)-R_{0,h}^{(10,10)}(t)\right)/\left(R_h^{fGn}(t)-R_{0,h}(t)\right)\right|$.   The lines are for $h = 2/10$,
$4/10,\ldots,16/10$, $18/10$ (excluding the exponential case $h = 1$), from left to right (note convergence
is only for irrational $h$, therefore an extra $10^{-4}$ was added to each $h$). For the low $h$ values the
convergence is particularly slow.
Comments:
1) These and the following formulae are for $t>0$; in addition, only the even integer
ordered terms are non zero (the sum over odd $j$).

2) Each integer term of the expansion $F_j$ is itself obtained as an infinite sum, so that
the overall result for $R_{\alpha,h}(t)$ is effectively a doubly infinite sum. This procedure swaps the
order of the summation and apparently explains the fact that while the expansions were
derived for the case $0<h<1$, the final expansion is valid for $0\leq\alpha<1/2$ and the full range
$0<h<2$: numerically, it accurately reproduces the oscillations when $h>1$.
3) The fGn correlation function is given by the single $n=2$ term:

$$R_h^{(fGn)}(t)=D_2\Gamma(1-2h)t^{-1+2h}=\frac{\sin(h\pi)}{\pi}\Gamma(1-2h)t^{-1+2h}\qquad\text{(A.18)}$$
It is also proportional to the correlation function of the fGn forced $h=0$, fRn process:
$$R_h^{(fGn)}(t)=4R_{\alpha=h,0}(t).$$

4) When $0<\alpha+h<1/2$, $R$ is divergent at the origin; this leading term
$\Gamma(-1-2(h+\alpha))\sin(\pi(h+\alpha))t^{-1+2(h+\alpha)}/\pi$ is only dependent on $h+\alpha$ corresponding to an
fGn with parameter $h+\alpha$. When $\frac{1}{2}<h+\alpha<2$, it is still the leading fractional term, but the
constant $F_1$ dominates at small $t$.
5) The $F_j$ terms diverge when $(j-2\alpha)/h$ is an integer. For example, if $\alpha=0$, the
overall sum over all $j$ thus diverges for all rational $h$. For irrational $h$, the convergence
properties are not easy to establish, although due to the $\Gamma$ functions, these series apparently
converge for all $t\geq0$, but the convergence is rather slow.
Fig. A1 shows some numerical results for $\alpha=0$ showing the convergence of the
$10^{th}$ order fractional $10^{th}$ order integer power approximation ($n_{max}=j_{max}=10$). Since the
leading (fGn) term diverges for small $t$, when $h\leq1/2$ it is more useful to consider the
convergence of the difference with respect to the fGn term i.e. $R_h^{(fGn)}(t)-R_{0,h,a}(t)$ where
the approximation $R_{0,h,a}(t)$ is from the sum from $n=3$ to $10$ and odd $j\leq9$. Fig. A1 shows
the logarithm of the ratio of the approximation with respect to the true value:
$r=\log_{10}\left|1-\left(R_h^{(fGn)}(t)-R_{0,h,a}(t)\right)/\left(R_h^{(fGn)}(t)-R_{0,h}(t)\right)\right|$ (to avoid exact rationals, $10^{-4}$ was
added to the $h$ values). From the figure we sees that the approximation is satisfactory
except for small $h$. In the next section we return to this.
6) For $\alpha+h>1/2$, when $t=0$, the only nonzero term is from the constant $F_1$: $R_{\alpha,h}(0)$
$=F_1$, this gives the normalization constant. Comparing with eq. 27, we therefore have:
$$R_{\alpha,h}(0)=\int_0^\infty G_{\alpha,h}(u)^2\,du=F_1=-\frac{\cos\pi\left(\dfrac{h}{2}+\alpha\right)}{h\sin\left(\dfrac{\pi h}{2}\right)\sin\left(\dfrac{\pi}{h}(1-2\alpha)\right)};\quad\alpha+h>1/2;\quad\begin{array}{l}0\leq\alpha<1/2\\1/2<h<2\end{array}$$

(A.19)
Similarly, when $\alpha+h>3/2$, for the quadratic the squared integral of $G'_{\alpha,h}$ is finite and it
gives the coefficient of the $t^2$ term so that:

$$\int_0^\infty G'_{\alpha,h}(s)^2 \, ds = -\frac{F_3}{\Gamma(3)} = \frac{\cos\left(\frac{\pi}{2}(h+2\alpha)\right)}{2h\sin\left(\frac{\pi h}{2}\right)\sin\left(\frac{\pi}{h}(3-2\alpha)\right)}; \quad h+\alpha > \frac{3}{2}$$

(A.20)

7) The expression for $V_{\alpha,h}(t)$ can be obtained by integrating twice (eq. 36).

8) In the special cases $h = 1/m$, with $m$ a positive integer, $F_j$ is independent of $j$ and
the integer powered series can be summed yielding a result proportional to $\cosh t$. However,
this large $t$ divergence is cancelled out by the fractional term and the result is finite (this
partial cancellation is discussed in the next subsection). The special important case $h = 1/2$
is dealt with in appendix B.
**A.4 A Convenient approximation**

The expansion for $R_{\alpha,h}$ is the sum of a fractional and an integer ordered series.
Partial sums appear to converge (fig. A1), albeit slowly. For simplicity, we consider the
case of primary interest, a pure fRn process ($\alpha = 0$). Examination of partial sums shows
that the integer ordered and fractional ordered terms tend to cancel, the difficulty due to
the coefficient of the integer ordered terms $j \approx hn + 2\alpha$ that comes from the exponential
integral and can be large when $j \approx hn + 2\alpha$. This suggests an alternative way of expressing
the series:
$$R_{0,h}(t) = \sum_{n=2}^\infty D_n E_{nh}(t) + \sum_{j=1}^\infty C_j \frac{(-1)^{j-1}}{\Gamma(j)} t^{j-1}; \quad C_j = \sum_{n=2}^\infty \frac{D_n}{(hn+j)}$$
(A.21)


Where $D_n$ is given by eq. A.5 and the $n$ sums start at $n = 2$ since $D_1 = 0$. $C_j$ can be expressed
as:
$$C_j = -\frac{ie^{-ih\pi}}{2\pi h\left(e^{ih\pi}-1\right)}\left(-\left(e^{ih\pi}+e^{2ih\pi}\right)\Phi\left(-1,1,1+\frac{j}{h}\right)+\Phi\left(e^{ih\pi},1,1+\frac{j}{h}\right)+e^{3ih\pi}\Phi\left(e^{-ih\pi},1,1+\frac{j}{h}\right)\right)$$

(A.22)
where $\Phi$ is the Hurwitz-Lerch phi function $\Phi(z,s,a) = \sum_{n=0}^\infty z^n (n+a)^{-s}$.
We can also expand the exponential integral:
$$E_{nh}(t) = \pi\frac{t^{-1+hn}}{\sin(\pi nh)\Gamma(hn)} + \sum_{j=1}^\infty \frac{(-1)^{j-1}}{(hn-j)\Gamma(j)} t^{j-1}$$
(A.23)

For the $j_{max}$ and $n_{max}$ partial sums, we have:
$$R_{0,h}^{(n_{max},j_{max})}(t) = \sum_{n=2}^{n_{max}} D_n \Gamma(1-nh) t^{-1+hn} + \sum_{j=1}^{j_{max}} F_{j,n_{max}} \frac{(-1)^{j-1}}{\Gamma(j)} t^{j-1}; \quad F_{j,n_{max}} = C_j + \sum_{n=2}^{n_{max}} \frac{D_n}{hn-j}$$

(A.24)
Now define the ($j_{max}$, $n_{max}$) approximation by:
$$R_{0,h,n_{max},j_{max}}(t) = \frac{R_{0,h}^{(n_{max}+1,j_{max})}(t) + R_{0,h}^{(n_{max},j_{max})}(t)}{2}$$
(A.25)

This has the effect of adding in half the next higher $n$ term and is more accurate; overall,
$j_{max}$ and $n_{max}$ may now be taken to be much smaller than in the previous approximation. For
example putting $n_{max} = 2$, $j_{max} = 1$, we get with the partial sum:
$$R_{0,h,2,1}(t) = R_h^{(fGn)}(t) + \frac{D_3}{2}\Gamma(1-3h)t^{-1+3h} + F_1$$
(A.26)

Where:
$$F_1 = C_1 + \frac{D_2}{2h-1} + \frac{D_3}{2(3h-1)}$$

$$D_2 = \frac{\sin(\pi h)}{\pi}; \quad D_3 = -\frac{\sin(\pi h)(1+2\cos(\pi h))}{\pi}$$

(A.27)
To understand the behaviour, fig. A2 shows the behaviour of coefficient of the
$t^{-1+3h}$ term $\frac{D_3}{2}\Gamma(1-3h)$, the constant term $F_1$ and the coefficient of the next integer (linear
in $t$) term $F_2 = C_2 + \frac{D_2}{2h-2} + \frac{D_3}{2(3h-2)}$. Up until the end of the fGn region ($h = 1/2$), the
$t^{-1+3h}$ and $F_1$ terms have opposite signs and tend to cancel. In addition, we see that for $t$
$\approx<1$ and $h<1$, they dominate over the (omitted) linear term. Fig. A3 shows that the $R_{0,h,2,1}$
approximation is surprisingly good for $h<1$ and is still not so bad for $1< h <2$. This
approximation is thus useful for monthly resolution macroweather temperature fields that
have relaxation times of years or longer and where $h$ is mostly over the range $0< h <1/2$,
but over some tropical ocean regions can increase to as much as $h \approx 1.2$ ([*Del Rio Amador*
*and Lovejoy*, 2021a]). Fig. A3 shows that the (2,1) approximation is reasonably accurate
for $t \approx<1$, especially for $h<1$.

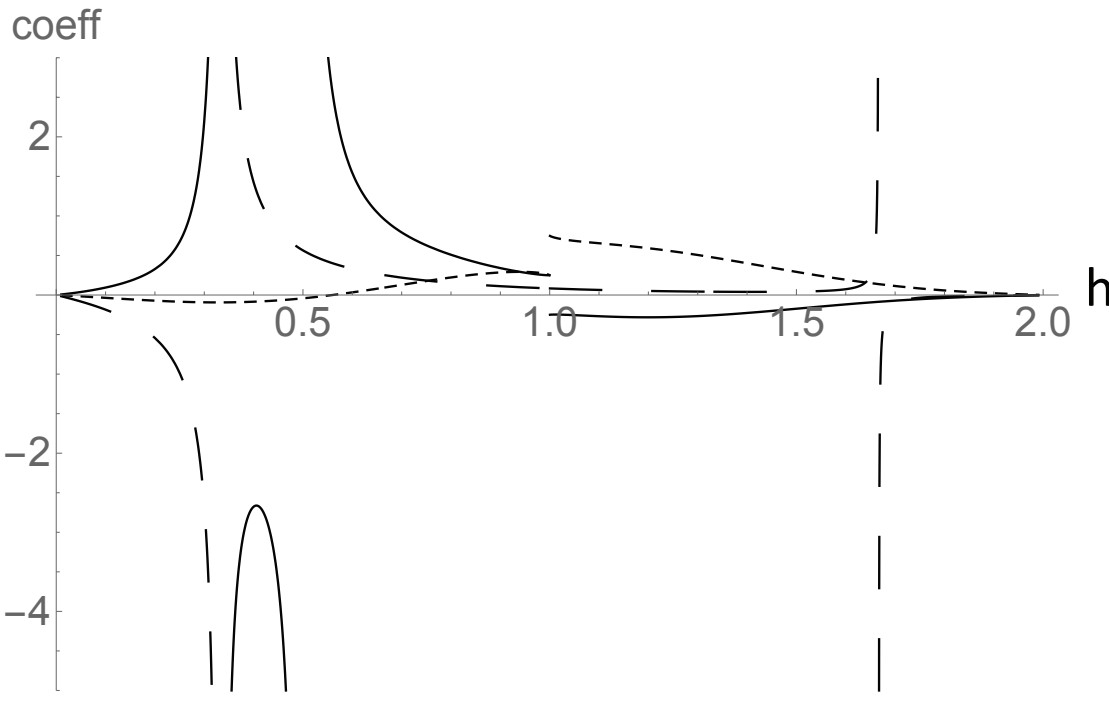

Fig. A2: The solid line is the constant term $F_1$, the long dashes are the coefficients
$\frac{D_3}{2}\Gamma(1-3h)$ of the fractional power, the short dashes are the coefficients of the linear term:
$F_2 = C_2 + \frac{D_2}{2h-2} + \frac{D_3}{2(3h-2)}$.  We can see that the contribution of the linear term (used in the $R_{0,h,2,2}(t)$
approximation) for $h<1$ and $t<1$ is fairly small; whereas for $1<h<2$, it is larger and the $R_{0,h,2,2}(t)$
approximation is significantly better than the $R_{0,h,2,1}(t)$ approximation (see fig. A3).

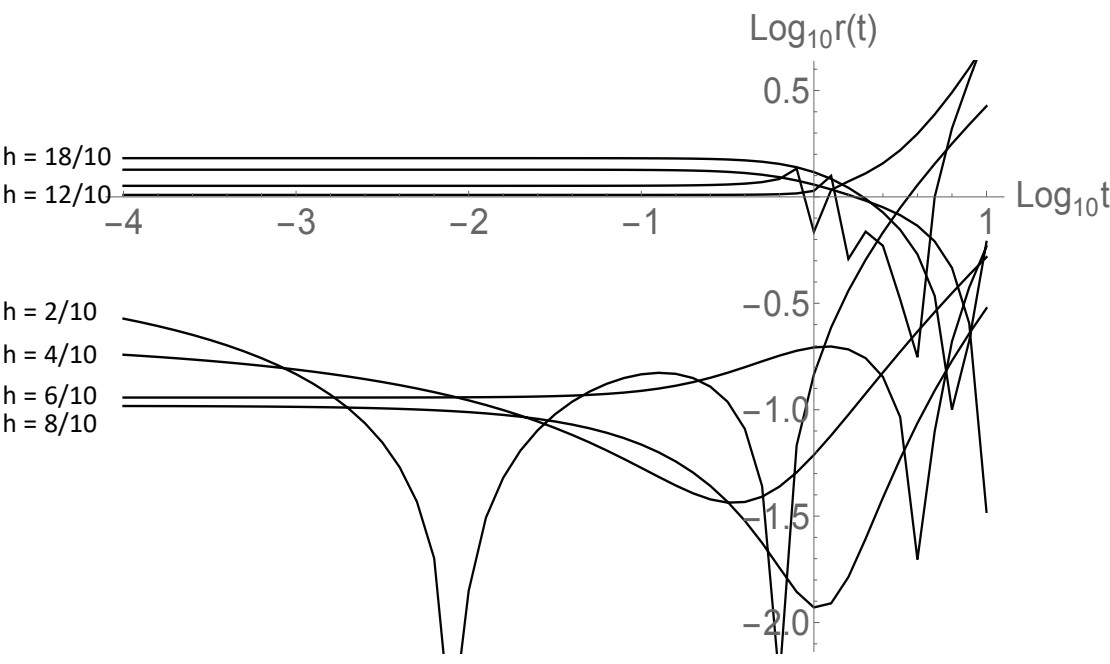

Fig. A3: This shows the logarithm of the relative error in the (2,1) approximation with
respect       to       the       deviation       from       the       fGn       $R_h(t)$
($r = \log_{10}\left|1 - \left(R_h^{fGn}(t) - R_{0,h,2,1}(t)\right) / \left(R_h^{fGn}(t) - R_{0,h}(t)\right)\right|$).   For $h<1$, $t<0$ it is of the order $\approx 30\%$
whereas for $h>1$, it of the order 100%.  The $h = 1$ (exponential) curve is not shown although when
$t<0$ the error is of order 60%.

## Appendix B: The h=1/2 special case

When $\alpha = 0$, $h = 1/2$, the high frequency fGn limit is an exact "1/f noise", (spectrum
$\omega^{-1}$) it has both high and low frequency divergences. The high frequency divergence can
be tamed by averaging, but not the low frequency divergence so that fGn is only defined
for $h<1/2$. However, for fRn, the low frequencies are convergent over the whole range 0
$< h < 2$, and for $h = 1/2$ we find that the correlation function has a logarithmic dependence
at both small and large scales. This is associated with particularly slow transitions from
high to low frequency behaviours. The critical value $h = 1/2$ corresponds to the HEBE
[*Lovejoy*, 2021a; b] where it was shown that the value $h = 1/2$ could be derived analytically
from the classical Budyko-Sellers energy balance equation. Therefore, $R_{\alpha,1/2}(t)$, $V_{\alpha,1/2}(t)$,
characterize the statistics of the temperature response of the classical heat equation
response to an fGn order $\alpha$ forcing.
It is possible to obtain exact analytic expressions for $R_{\alpha,1/2}(t)$, $V_{\alpha,1/2}(t)$ and the Haar
fluctuations; we develop these in this appendix, for some early results, see [*Mainardi and*
*Pironi*, 1996].
The starting point is the Laplace expression A2 with $h = 1/2$:

$$R_{\alpha,h}(t) = -\frac{1}{\pi}\operatorname{Im}e^{-i\alpha\pi}\int_0^\infty \frac{e^{-xt}dx}{x^{2\alpha}\left(1+x^{1/2}\right)\left(1+ix^{1/2}\right)} = -\frac{1}{\pi\sqrt{2}}\operatorname{Im}e^{-i\pi\alpha}\int_0^\infty x^{-2\alpha}\left(\frac{e^{i\pi/4}}{1+x^{1/2}}+\frac{e^{-i\pi/4}}{1+x}-\frac{e^{i\pi/4}x^{1/2}}{1+x}\right)e^{-xt}dx$$

(B1)
We require the following Laplace transforms:

$$L_1(t) = \int_0^\infty \frac{e^{-xt}}{x^{2\alpha}\left(1+x^{1/2}\right)}dt = e^{-t-2i\pi\alpha}\left(\Gamma(1-2\alpha)\Gamma(2\alpha,-t)-i\Gamma\left(\frac{3}{2}-2\alpha\right)\Gamma\left(2\alpha-\frac{1}{2},-t\right)\right)$$

$$L_2(t) = \int_0^\infty \frac{e^{-xt}}{x^{2\alpha}\left(1+x\right)}dt = e^t\Gamma(1-2\alpha)\Gamma(2\alpha,t)$$

$$L_3(t) = \int_0^\infty \frac{e^{-xt}x^{1/2}}{x^{2\alpha}\left(1+x\right)}dt = e^t\Gamma\left(\frac{3}{2}-2\alpha\right)\Gamma\left(2\alpha-\frac{1}{2},t\right)$$

(B.2)
Where we have introduced the incomplete gamma function: $\Gamma(a,z) = \int_z^\infty u^{a-1}e^{-u}\,du$ (with a
branch cut in the complex plane from $-\infty$ to 0). The general result is thus:
$$R_{\alpha,1/2}(t) = \frac{1}{2\pi}\left(\sin\pi\alpha\left(L_1(t)+L_2(t)-L_3(t)\right)+\cos\pi\alpha\left(-L_1(t)+L_2(t)+L_3(t)\right)\right)$$
(B.3)
Fig. B1 shows plots $R_{\alpha,1/2}(t)$ over 8 orders of magnitude in $t$, indicating the generally
very slow converge to the asymptotic behaviour (shown as straight lines at the right).
Fig. B1 also shows the singular small $t$ behaviour of the pure fRn case ($\alpha = 0$).  In
this limit both $L_1$, and $L_2$, are  singular - they both yield logarithmic small scale divergences.
Pure fRn is of special interest, and yields the somewhat simpler result:
$$R_{0,1/2}(t) = \frac{1}{2}\left(e^{-t}\text{erfi}\sqrt{t} - e^{t}\text{erfc}\sqrt{t}\right) - \frac{1}{2\pi}\left(e^{t}\,Ei(-t) + e^{-t}\,Ei(t)\right);$$

$$Ei(z) = -\int_{-z}^{\infty}e^{-u}\,\frac{du}{u}$$
;

$$\text{erfi}(z) = -i\left(\text{erf}(iz)\right); \quad \text{erf}(z) = \frac{2}{\sqrt{\pi}}\int_{0}^{z}e^{-s^2}\,ds$$

(B.4)
We can use these results to obtain small and large $t$ expansions:
$$R_{0,1/2}(t) = -\left(\frac{2\gamma_E + \pi + 2\log t}{2\pi}\right) + \frac{2\sqrt{t}}{\sqrt{\pi}} - \frac{t}{2} - \left(\frac{3 + 2\gamma_E + \pi + 2\log t}{4\pi}\right)t^2 + O\left(t^{3/2}\right); \quad t \ll 1$$

(B.5)

$$R_{0,1/2}(t) = \frac{1}{2\sqrt{\pi}}t^{-3/2} - \frac{1}{\pi}t^{-2} + \frac{15}{8\sqrt{\pi}}t^{-7/2} + O\left(t^{-4}\right); \quad t \gg 1 \ ,$$

where $\gamma_E$ is Euler's constant $= 0.57...$ (the asymptotic formula can be obtained as a special
case of eq. in appendix A, but note the logarithmic small scale divergence).
To obtain the corresponding results for $V_{0,1/2}$ use: $V_{0,1/2}(t) = 2\int_{0}^{t}\left(\int_{0}^{v}R_{0,1/2}(u)\,du\right)dv$.
The exact $V_{0,1/2}$ is:
$$V_{0,1/2}(t) = G_{3,4}^{2,2}\left[t\left|\begin{array}{cccc} 2, & 2, & 5/2 & \\ 2, & 2, & 0, & 5/2 \end{array}\right.\right] + \frac{e^{t}}{\pi}\left(Shi(t) - Chi(t)\right) + \left(e^{-t}\text{erfi}\left(\sqrt{t}\right) - e^{t}\text{erf}\left(\sqrt{t}\right)\right)$$
$$+ t\left(1 + \frac{\gamma_E - 1}{\pi}\right) - 4\sqrt{\frac{t}{\pi}} + \frac{(1+t)\log t}{\pi} + 1 + \frac{\gamma_E}{\pi}$$

(B.6)
where $G_{3,4}^{2,2}$ is the MeijrG function, Chi is the CoshIntegral function and Shi is the
SinhIntegral function.  The expansions are:
$$V_{0,1/2}(t) = -\frac{t^2\log t}{\pi} + \frac{191 - 156\gamma_E - 78\pi}{144\pi} + \frac{16}{15\sqrt{\pi}}t^{5/2} - \frac{t^3}{6} - \frac{t^4\log t}{12\pi} + O\left(t^{3/2}\right); \quad t \ll 1$$

(B.7)

$$V_{0,1/2}(t) = t + \frac{\pi + 2\gamma_E}{\pi} + \frac{2\log t}{\pi} - \frac{4}{\sqrt{\pi}}t^{1/2} + \frac{1}{\sqrt{\pi}}t^{-1/2} - \frac{2}{\pi}t^{-2} + \frac{15}{4\sqrt{\pi}}t^{-3/2} + O\left(t^{-4}\right); \quad t \gg 1 \ .$$

We can also work out the variance of the Haar fluctuations:
$$\left\langle \Delta U_{0,1/2}^2 \left( \Delta t \right)_{Haar} \right\rangle = \frac{\Delta t^2 \log \Delta t}{4\pi} + \frac{6\pi + 12\gamma_E - \log 16 + 960 \log 2}{240\pi} + \frac{512 \left( \sqrt{2} - 2 \right)}{240\sqrt{\pi}} \Delta t^{1/2} + \frac{\Delta t}{3} + O \left( \Delta t^{3/2} \right); \quad \Delta t \ll 1$$
(B.8)
$$\left\langle \Delta U_{0,1/2}^2 \left( \Delta t \right)_{Haar} \right\rangle = 4\Delta t^{-1} - \frac{32\sqrt{2}}{\sqrt{\pi}} \Delta t^{-3/2} + \frac{3t^{-2} \log \Delta t}{\pi} + O \left( \Delta t^{-2} \right); \quad \Delta t \gg 1 \ .$$

Figure B2 shows numerical results for $\alpha = 0$, $h = \frac{1}{2}$, the transition between small and
large $t$ behaviour is extremely slow; the 9 orders of magnitude depicted in the figure are
barely enough.   The extreme low $(R_{1/2})^{1/2}$ (dashed) asymptotes at the left to a slope zero
(a square root  logarithmic limit, eq. B8), and to a -3/4 slope at the right.   The RMS Haar
fluctuation (black) changes slope from $H = 0$ to -1/2 (left to right).  Fig. B2 also shows the
logarithmic derivative of the RMS Haar (black) compared to a regression estimate over
two orders of magnitude in scale (dashed; a factor 10 smaller and 10 larger than the
indicated scale was used, this represents a possible empirically accessible range).   This
figure underlines the gradualness of the transition from $H = 0$ to $H = -1/2$.   If empirical
data were available only over a factor of 100 in scale, depending on where this scale was
with respect to the relaxation time scale (unity in the plot), the RMS Haar fluctuations could
have any slope in the range 0 to -1/2 with only small deviations.

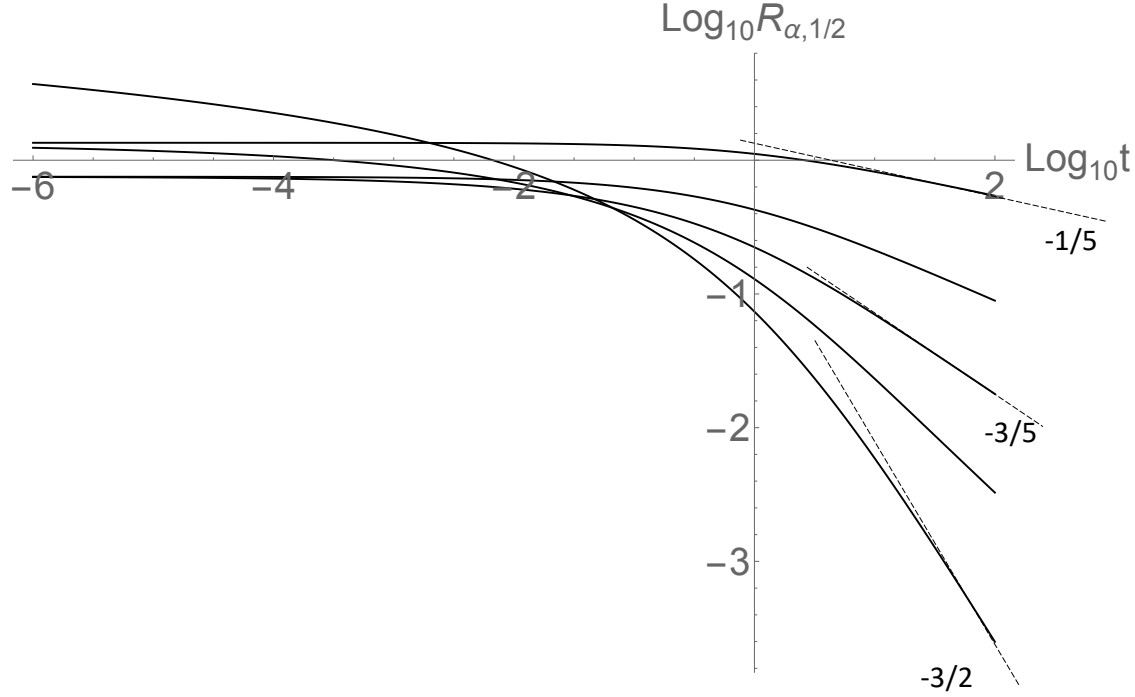

Fig. B1: $R_{\alpha,1/2}$ for $\alpha$ increasing from 0 (pure fRn) to 8/10 in steps of 1/10 (at right: bottom
to top). The $\alpha = 0$ curve has a logarithmic divergence at small $t$ (the far left).  Recall from section
that at large $t$, $R_{0,1/2} \approx t^{-3/2}$ and for $\alpha > 0$: $R_{\alpha,1/2} \approx t^{2\alpha-1}$, for $\alpha = 0$, 1/5, 2/5 the theoretical asymptotes of
the leading terms are indicated for reference.
.

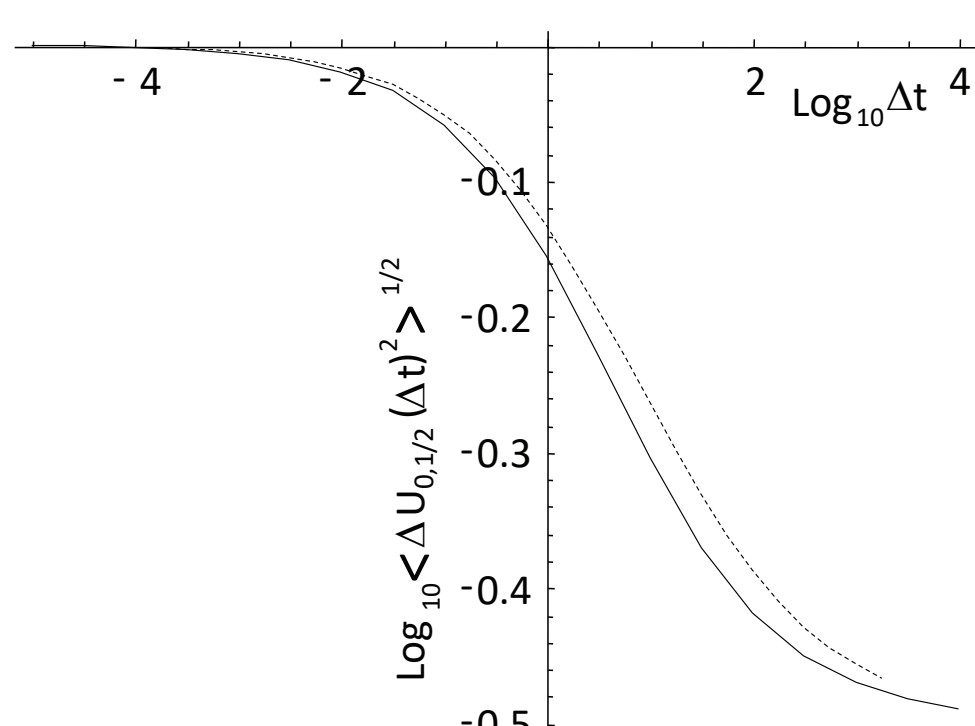

Fig. B2: The logarithmic derivative of the RMS Haar fluctuations of $U_{0,1/2}$ (solid) in fig.
B1 compared to a regression estimate over two orders of magnitude in scale (dashed; a factor 10
smaller and 10 larger than the indicated scale was used). This plot underlines the gradualness of
the transition from slopes 0 to -0.5 corresponding to *apparent* $H = 0$ to $H = -1/2$ scaling. Over
range of 100 or so in scale there is approximate scaling but with exponents that depend on the range
of scales covered by the data. If data were available only over a factor of 100 in scale, the RMS
Haar fluctuations could have any slope in the fGn range 0 to  -1/2 with only small deviations.

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
