# Peer review of "Fractional relaxation noises, motions and the fractional energy balance equation Shaun Lovejoy"

_Nonlinear Processes in Geophysics, 2019_

## Referee Comment (RC1) · Anonymous Referee #1 · 17 Nov 2019

The manuscript is devoted to a stochastic fractional model for the Earth's energy balance. The author developed a framework for handling fractional equations driven by white noise forcing, and analytically determined both the small and large scale limits for fractional relaxation motions (fRm) and fractional relaxation noises (fRn). He derived the main statistical properties of both fRm and fRn, including Green function, autocorrelation function, Haar fluctuations, spectra and sample process. These are extensions of fractional Gaussian noise and fractional Brownian motion. Furthermore, he examined the prediction of fRn, fRm with a past value problem by the minimum square skill score, which are needed for forecasting the Earth's temperature.

The main points of the manuscript is expressed concisely and the paper is overall well organized. I recommend acceptance by NPG, after a revision taking into account the

following comments and suggestions.

1. The main case studied in this manuscript is $a = -\infty$, i.e., Weyl fractional derivative. It would be nice to give more explanation on how to understand the $a = -\infty$ in the fractional derivative operator $_a D_t^H$, and what is the significance of the application in the model?

2. The caption of Fig.2 is not clear, such as " $H$ increasing in units of $1/10$ starting at a value $1/20$ above the plot minimum". Perhaps it is better to say like this " $H$ increasing in units of $1/10$ starting at a value $1/20$(upper left) to $39/20$(bottom right)" ?

3. In Fig. 5a, 5b, 6a, 6b about fRn and fRm simulations, it would be nice to add the sample path simulation of fractional Gaussian noise and fractional Brownian motion for comparison, which will help reader to better understand the fractional relaxation noise.

4. Fig. 7 and Fig.8 present the simulation for fGn and fRn forecasts. Is it possible to say or predict the Earth temperature $T$ based on fractional energy balanced equation?

5. It would be nice to give a summary of the main differences and links between fRn (fRm) and fGn (fGm) in conclusions.

Please also note the supplement to this comment:
https://www.nonlin-processes-geophys-discuss.net/npg-2019-39/npg-2019-39-RC1-supplement.pdf

---

## Referee Comment (RC2) · Anonymous Referee #2 · 20 Nov 2019

The paper introduces a novel class of stochastic processes, called "fractional relaxation noises and motions" (fRn, fRm) and discusses their particular application to a stochastic relaxation equation (Eq. 1), used to describe the Earth energy balance. The principal motivation behind the proposed processes is to synthesize the stochastic differential equation and scaling modeling approaches, each of which has been thoroughly explored (and shown efficient) in mathematics and applied literature.

The paper starts (Section 2) with introducing the fractional Langevin equation (Eq. 1) for the Earth energy balance, and uses it as a motivational example to develop the proposed theory, in parallel to what has been done for fractional Gaussian noises and Brownian motions (fGn, fBm). Section 3 discusses specific technical details, including derivations of process spectra and illustrating sample trajectories. This section also

notices a practically important aspect of not being able to distinguish between fRn, fRm, and fBm for specific ranges of parameters and for a finite range of observations. Section 4 examines the classical prediction problem, emphasizing that here one deal with the past value rather than initial value problem. Section 5 concludes.

The paper is clearly written (in particular, given its heavy math content). The materials is novel and of clear importance for the nonlinear geosciences community. I trust the paper will be of interest to the broad NPG readership and urge its publication.

I have a comment related to the paper organization. Given the discussed material and the publication venue, I expect there will be two main categories of readers (that of course overlap) – those who are more interested in the math details, and those who are mainly interested in qualitative findings and applications. I think the paper will benefit from reorganization that first will clearly list the main proven facts (process definition, statement of stationarity, correlation function, spectra, sample path, prediction), and then will present the underlying derivations. The current version is math heavy and makes it hard to clearly see the key points of the presented material. Also, it is very important to explicitly list the differences between the fRn, fRm and their counterparts fGn, fBm.

Other minor comments/typos: 1) l. 59-60: Rewrite to mention the authors (West et al.) outside of the reference. This is how the sentence was originally intended. 2) l. 68: "martingales" should not be capitalized. 3) l. 72 and everywhere: "Earth" should be capitalized. 4) Please use proper punctuation in all equations (commas, periods). 5) Eq. (1): Why not give refs to the classical EBM of Budyko and Sellers before diving into a fractional version? 6) Eq. (2): Define Gamma 7) l. 167: "standard Brownian motion" (instead of "usual Brownian motion")

---

## Referee Comment (RC3) · Anonymous Referee #3 · 11 Dec 2019

This article deals with the resolution and the mathematical properties of the solutions of a stochastic fractional relaxation equation.

The main motivation is supplied by the fact that, to model the earth's energy balance, this equation presents several advantages over previously considered equations.

Physics requirements have implications which are used in the derivation of this equation, and lead to its particular form:

- integer order derivatives in the equation lead to unrealistic Green functions so that fractional derivatives are required

- Solutions must be stationary

[Figure]

This paper certainly is an important contribution to the existing literature on fractional processes. In particular, many divergence issues are discussed and handled convincingly. A natural question which is not addressed (though it is an important issue in real-life applications) is the discussions of Gaussian vs. nonGaussian modelling of the noise part in the equations. Many references in this paper are (rightly) to Benoit Mandelbrot, who was extremely careful to discuss this matter with great care, and I would have expected this issue to be discussed (even briefly) in such a review paper (the mathematical study of the processes solutions of equations with nonGaussian noise, though it started much later than in the Gaussian case, now starts to be substantial). The nature of this paper may seem surprising: This volume will contain review papers on geophysics issues. The present paper is mainly constituted of lengthy explicit computations, which are often hard to follow, partly because the paper is not self-contained: at many key-points in the proofs, the reader is just referred to another paper.

Additionally, very little physical intuition is given to back these formal computations. When this happens (see e.g. lines 445-454), the explanations are sketched and can give light only to readers that already are well acquainted with these questions. However, here, the review part, and in particular the geophysical motivations, are barely sketched and the reader is mostly advised to consult references. A very positive point is that simulations are welcome and convincing: they clearly show that different qualitative behaviors can occur. Here too, I think that the geophysical implications of these differences would deserve to be discussed in more details. The issue of prediction only is discussed for its physics implications.

As it is, this paper is more an exploratory paper in applied mathematics (as can be found in applied math journal, such as e.g. SIAM review). In my opinion, substantial rewriting would be needed to make it a review paper in geophysics.

---

## Referee Comment (RC4) · Anonymous Referee #4 · 11 Dec 2019

The author proposes here new models, built using fractional derivatives in a stochastic framework, to model geoscience dynamics in a way compatible with scaling properties and long-range correlations.

Since there are many equations I regret that the author is not using LateX, which would be useful for a correct display of all the complex notations (see e.g. line 307, 320, 322, 333 etc. where alignment is not correct).

There are many mathematical expressions and the narrative is not clear. I suggest to explicitly indicate what is the process studied here, and what are its properties. A section on this seems really necessary: either in the beginning or at the end, as a kind of summary. Also, what is precisely the novelty, why is it necessary to have an infinite memory, what is new with respect to fBm. Also is the modeled process multifractal or

monofractal ?

Figures 5 and 6 display some simulation realizations for various parameter values. What are precisely the equations used for these simulations? It would be useful to provide the code to the community.

Line 110: the author cites Lovejoy et al 2019 as the original introduction of the idea presented here. However this paper is indicated in the references as "in preparation". Hence I suggest to remove it from the reference list, since it cannot be consulted and is not yet published.

The same applies to Hebert et al 219, which is under revision.

This is a long paper with relatively few references. I recommend to add more references. For example equation (4) for the Âń canonical Âż Weyl relaxation equation: if this equation is classical, a reference is here welcome. Another example: equation (16) seems to be a mathematical result and hence reference to relevant mathematical literature is needed.

There is a rather vast literature on fractional dynamics, or continuous time fractional random walks, which is only superficially discussed. I recommend to provide a link with this literature, at least in the introduction and discussion. For example the following might be relevant to discuss (this list is not exhaustive):

M. M. Meerschaert, A. Sikorskii, Stochastic Models for Fractional Calculus, De Gruyter Studies in Mathematics 43, 2012 R. Hilfer (Ed.), Applications of Fractional Calculus in Physics, World Scientific, 2000. J. Klafter, S. Lim, R. Metzler (Eds.), Fractional Dynamics: Recent Advances, World Scientific, 2011. D. Baleanu, K. Diethelm, E. Scalas, J. Trujillo, Fractional Calculus, Models and Numerical Methods, 2nd Edition, 2016

---

## Author Comment (AC1) · 17 Jan 2020

Overall reactions/comments

The referees were both positive and helpful, the manuscript has been considerably improved, notably by a significant rewrite of the introduction and conclusion. The number of references has nearly tripled and the introduction now better situates the paper into a large body of literature. The updated conclusions underline the specific contributions of this paper and better point to future directions. I should also mention that when the paper was written, I was on the verge of submitting another paper on the deterministic FEBE and its physical basis. Instead, the time was spent on the more urgent task of deriving the special H = 1/2 case – the Half ordered EBE (HEBE) from the classical

energy transport equations (submitted in November). Although the determinstic FEBE paper will be submitted shortly, it still has not been done and therefore – with the referee's encouragement - I have given much more discussion of the geophysics aspects of the problem.

Referee 1: Anonymous Referee #1

The manuscript is devoted to a stochastic fractional model for the Earth's energy balance. The author developed a framework for handling fractional equations driven by white noise forcing, and analytically determined both the small and large scale limits for fractional relaxation motions (fRm) and fractional relaxation noises (fRn). He derived the main statistical properties of both fRm and fRn, including Green function, autocorrelation function, Haar fluctuations, spectra and sample process. These are extensions of fractional Gaussian noise and fractional Brownian motion. Furthermore, he examined the prediction of fRn, fRm with a past value problem by the minimum square skill score, which are needed for forecasting the Earth's temperature. The main points of the manuscript is expressed concisely and the paper is overall well organized. I recommend acceptance by NPG, after a revision taking into account the following comments and suggestions.

Au: I thank the referee for the positive evaluation.

1. The main case studied in this manuscript is a $=-\infty$, i.e., Weyl fractional derivative. It would be nice to give more explanation on how to understand the a $=-\infty$ in the fractional derivative operator aDHt, and what is the significance of the application in the model?

Au: The key motivation and consequence of choice a$=-\infty$ is that it is needed to obtain a statistically stationary fRn process and an fRm process with stationary increments. This is explicitly developed in detail in appendix A. In the introduction we have added more material on this.

2. The caption of Fig.2 is not clear, such as "H increasing in units of 1/10 starting at a value 1/20 above the plot minimum". Perhaps it is better to say like this "H increasing in units of 1/10 starting at a value 1/20 (upper left) to 39/20 (bottom right)" ?

Au: Thanks, this has been done.

3. In Fig. 5a, 5b, 6a, 6b about fRn and fRm simulations, it would be nice to add the sample path simulation of fractional Gaussian noise and fractional Brownian motion for comparison, which will help reader to better understand the fractional relaxation noise.

Au: As indicated in the captions, fig. 5a, 6a already do show the fGn and fBm processes: their bottom lines. The wording in the captions has been improved to make this clear, and a reference to this has been added to the text.

4. Fig. 7 and Fig. 8 present the simulation for fGn and fRn forecasts. Is it possible to say or predict the Earth temperature T based on fractional energy balanced equation?

Au: Yes, indeed it was the success of macroweather predictions and climate projections that using the exponents from the FEBE that led to the initial proposal to use that the FEBE be a good model. Although this was mentioned in the introduction and conclusion, I have added a new paragraph detailing this.

5. It would be nice to give a summary of the main differences and links between fRn (fRm) and fGn (fGm) in conclusions.

Au: This has been done.

  Referee #2 Anonymous Referee #2

The paper introduces a novel class of stochastic processes, called "fractional relaxation noises and motions" (fRn, fRm) and discusses their particular application to a stochastic relaxation equation (Eq. 1), used to describe the Earth energy balance. The principal motivation behind the proposed processes is to synthesize the stochastic differential equation and scaling modeling approaches, each of which has been thoroughly explored (and shown efficient) in mathematics and applied literature.

The paper starts (Section 2) with introducing the fractional Langevin equation (Eq. 1) for the Earth energy balance, and uses it as a motivational example to develop the proposed theory, in parallel to what has been done for fractional Gaussian noises and Brownian motions (fGn, fBm). Section 3 discusses specific technical details, including derivations of process spectra and illustrating sample trajectories. This section notices a practically important aspect of not being able to distinguish between fRn, fRm, and fBm for specific ranges of parameters and for a finite range of observations. Section 4 examines the classical prediction problem, emphasizing that here one deals with the past value rather than initial value problem. Section 5 concludes.

The paper is clearly written (in particular, given its heavy math content). The materials is novel and of clear importance for the nonlinear geosciences community. I trust the paper will be of interest to the broad NPG readership and urge its publication.

Au: Thank you for your positive evaluation.

I have a comment related to the paper organization. Given the discussed material and the publication venue, I expect there will be two main categories of readers (that of course overlap) – those who are more interested in the math details, and those who are mainly interested in qualitative findings and applications. I think the paper will benefit from reorganization that first will clearly list the main proven facts (process definition, statement of stationarity, correlation function, spectra, sample path, prediction), and then will present the underlying derivations. The current version is math heavy and makes it hard to clearly see the key points of the presented material. Also, it is very important to explicitly list the differences between the fRn, fRm and their counterparts fGn, fBm.

Au: I apologize for the heavy math details. The organization of a paper takes into account many factors, in this case mostly the needed to systematically, logically develop the material, but remains somewhat subjective. Yet referee #1 states: "The main points

of the manuscript is expressed concisely and the paper is overall well organized." In order to answer this criticism as well as possible, I made numerous additions to the introduction including a new final paragraph detailing the main results and how they fit in the paper. In the conclusions, I have also compared and contrasted fRn, fRm and their counterparts fGn, fBm.

Other minor comments/typos: 1) l. 59-60: Rewrite to mention the authors (West et al.) outside of the reference. This is how the sentence was originally intended.

Au: OK

2) l. 68:"martingales" should not be capitalized. Au: OK

3) l. 72 and everywhere: "Earth" should be capitalized. Au: OK

4) Please use proper punctuation in all equations (commas, periods). Au: OK

5) Eq. (1): Why not give refs to the classical EBM of Budyko and Sellers before diving into a fractional version? Au: This has now been done.

6) Eq. (2): Define Gamma

Au: OK

7) l. 167: "standard Brownian motion"(instead of "usual Brownian motion"). Au: OK   Referee #3 Anonymous Referee #3

This article deals with the resolution and the mathematical properties of the solutions of a stochastic fractional relaxation equation. The main motivation is supplied by the fact that, to model the earth's energy balance, this equation presents several advantages over previously considered equations. Physics requirements have implications which are used in the derivation of this equation, and lead to its particular form: integer order derivatives in the equation lead to unrealistic Green functions so that fractional derivatives are required- Solutions must be stationary.

This paper certainly is an important contribution to the existing literature on fractional processes. In particular, many divergence issues are discussed and handled convincingly. A natural question which is not addressed (though it is an important issue in real-life applications) is the discussions of Gaussian vs. non Gaussian modelling of the noise part in the equations. Many references in this paper are (rightly) to Benoit Mandelbrot, who was extremely careful to discuss this matter with great care, and I would have expected this issue to be discussed (even briefly) in such a review paper (the mathematical study of the processes solutions of equations with non Gaussian noise, though it started much later than in the Gaussian case, now starts to be substantial). The nature of this paper may seem surprising: This volume will contain review papers on geophysics issues. The present paper is mainly constituted of lengthy explicit computations, which are often hard to follow, partly because the paper is not self-contained: at many key-points in the proofs, the reader is just referred to another paper.

Au: Thank you for your positive evaluation. I have augmented the introduction and conclusion to include brief discussion of the question of Levy and multifractal forcing that are indeed important, but outside the present scope. The main aspects of the paper that are not self-contained are the fractional relaxation Green's functions, as well as their series expansions. Adding this standard (albeit not widely known) material would lengthen the paper without improving its clarity.

Additionally, very little physical intuition is given to back these formal computations. When this happens (see e.g. lines 445-454), the explanations are sketched and can give light only to readers that already are well acquainted with these questions. However, here, the review part, and in particular the geophysical motivations, are barely sketched and the reader is mostly advised to consult references. A very positive point is that simulations are welcome and convincing: they clearly show that different qualitative behaviors can occur. Here too, I think that the geophysical implications of these differences would deserve to be discussed in more details. The issue of prediction only is discussed for its physics implications.

Au: I have added several paragraphs especially in the introduction giving more physical justification, discussion. Several papers with this physical side are either under review or are in advanced stages of preparation.

As it is, this paper is more an exploratory paper in applied mathematics (as can be found in applied math journal, such as e.g. SIAM review). In my opinion, substantial rewriting would be needed to make it a review paper in geophysics.

Au: It is a research, not a review paper.

  Referee #4 Anonymous Referee #4

The author proposes here new models, built using fractional derivatives in a stochastic framework, to model geoscience dynamics in a way compatible with scaling properties and long-range correlations. Since there are many equations I regret that the author is not using LateX, which would be useful for a correct display of all the complex notations (see e.g. line 307, 320, 322,333 etc. where alignment is not correct).

Au: I have improved these. The final version will of course be re-typeset by NPG.

There are many mathematical expressions and the narrative is not clear. I suggest to explicitly indicate what is the process studied here, and what are its properties. A section on this seems really necessary: either in the beginning or at the end, as a kind of summary. Also, what is precisely the novelty, why is it necessary to have an infinite memory, what is new with respect to fBm. Also is the modeled process multifractal or monofractal ?

Figures 5 and 6 display some simulation realizations for various parameter values. What are precisely the equations used for these simulations? It would be useful to provide the code to the community.

Au: The numerical details were stated but too briefly; I have added more details and references, at the beginning of section 3.6.

Line 110: the author cites Lovejoy et al 2019 as the original introduction of the idea presented here. However this paper is indicated in the references as "in preparation". Hence I suggest to remove it from the reference list, since it cannot be consulted and is not yet published. The same applies to Hebert et al 219, which is under revision.

Au: I removed these references. In the meantime, a new paper on the half-order FEBE (the HEBE) analytically derived from the standard energy transport equations was submitted, this delayed the deterministic FEBE paper that was referred to in the text.

This is a long paper with relatively few references. I recommend to add more references. For example equation (4) for the canonical Weyl relaxation equation: if this equation is classical, a reference is here welcome. Another example: equation (16) seems to be a mathematical result and hence reference to relevant mathematical literature is needed. There is a rather vast literature on fractional dynamics, or continuous time fractional random walks, which is only superficially discussed. I recommend to provide a link with this literature, at least in the introduction and discussion.

Au: Thank you for the suggestions. I added these and many others, there are now 55 references – nearly triple the previous number.

For example the following might be relevant to discuss (this list is not exhaustive): M. M. Meerschaert, A. Sikorskii, Stochastic Models for Fractional Calculus, De Gruyter Studies in Mathematics 43, 2012 R. Hilfer (Ed.), Applications of Fractional Calculus in Physics, World Scientific, 2000. J. Klafter, S. Lim, R. Metzler (Eds.), Fractional Dynamics: Recent Advances, World Scientific, 2011. D. Baleanu, K. Diethelm, E. Scalas, J. Trujillo, Fractional Calculus, Models and Numerical Methods, 2nd Edition, 2016

---

## Referee Report (RR1)

**A review of "Fractional relaxation noises, motions and the fractional energy balance equation" by Shaun Lovejoy.**

General comments

Despite the reference increase, this revised version has still important gaps on the state-of-art and therefore remains unclear on the truly original contributions of this manuscript. There has been indeed an abundant literature on fractional differential equations, in particular on the relaxation-oscillation equation that is the topic of this paper.

The choice of this linear equation is debatable with respect to the focus on nonlinear geophysics of this special centennial issue. In this respect, there are several rather surprising statements, such as the second part of the sentence: "The choice of a Gaussian white noise forcing was made both for theoretical simplicity but also for physical realism" (L.171-172). This is unfortunately in direct agreement with the oft-quoted, ironic Chester Kisiel's pray to the theoretical hydrologist: "Oh, Lord, please, keep the world linear and Gaussian!".

Similarly, the claim of "the paucity of mathematical literature on stochastic fractional equations (see however [*Karczewska and Lizama*, 2009])" (L.78) is in contradiction with various review papers, in particular the Physics Report of Metzler and Klafter (2000) that focuses on "various generalisations to fractional order [that] have been employed, i.e. different fractional operators [that] have been introduced to replace either the time derivative or the occurring spatial derivatives, or both" and has more than 300 references. More generally, the author's emphasis on opposing fractional vs. integer order differential equations, both linear, seems rather outdated.

The tentative argument in favour of having an "enormous memory" with the help of a lower integration bound $t_0 = -\infty$ of the so-called Weyl fractional integration/differentiation (which in fact could be traced back to Liouville (1832)) is overstated, while it basically corresponds to an over-simplification, not only with respect to the finite date of the Big Bang, but also to Earth climate. Contrary to the author's claim that "the interval between an initial time = 0 and a later time t [...] is the exclusive domain considered in Podlubny's mathematical monograph on deterministic fractional differential equations [Podlubny, 1999]" (L136-138), this monograph, as several others, does deal with the Weyl fractional integration/differentiation and a more careful reading of it might have helped to simplify and make more rigorous the present manuscript. Let us clarify that a lot of efforts had been spent for the finite $t_0$ case (e.g., works of Gorenflo, Mainardi and collaborators) due to the fact it was much more difficult than the (negative) infinite case: in fact, it required to define a new fractional derivative (Caputo, 1967) to handle the initial conditions, whereas the classical Riemann-Liouville failed to do it. All these important technicalities vanish for the (negative) infinite case, basically because the Laplace convolution reduces to a Fourier convolution.

A paradox of this paper is that it claims to be innovative by focusing on the Weyl fractional integral/ derivative ("the key novelty of this paper is therefore to consider the FEBE as a Weyl fractional Langevin equation", L.914), while being lost in many mathematical details (e.g., Mittag-Leffler functions) that are rarely necessary for this simplifying case (e.g., the composition of fractional integrals/derivatives is then commutative), as well as not taking advantage of other structural simplifications resulting from the combination of the linear and Gaussian assumptions (e.g., linear stability). On the contrary, the (potential) bringing-in of Fourier techniques is mostly limited to spectra in the short sub-section 3.5 "Spectra".

It seems that the main results are the scaling behaviours of the studied noise and motion, which the author calls fractional relaxation noise and motion, for small and large time lags. Unfortunately, it is not clear what is analytically obtained and to which level. Often information is missing on important issues, whereas there are numerous mathematical displays. Unfortunately, mathematical rigour is not always there, contrary to mathematical pitfalls that bring into question what is really

obtained (see examples in detailed comments). A clear synthesis, with a comparison with their classical counterparts of fractional Brownian noise and motion, is unfortunately missing.

The aforementioned problems are amplified in the sections on simulations and prediction and make them very difficult to evaluate before these problems will be solved. A final general comment is that the author's claim that the studied noises and motions are generalisation of the Brownian ones is not obvious. Indeed, what could be the new generality gained with their help? On the contrary, an important and generic property has been lost: scaling. This is a direct consequence of the introduction of a characteristic time (presently hidden by the non-dimensionalisation of equations) due to the presence of two time derivatives of different orders ($H$ and 0) instead of a unique one for (fractional) Brownian noises ($H$ or 0). With no surprise, the case of fractional differential polynomials of "degree" $n>2$ has been already formally investigated (e.g., Podlubny, 1999). This just illustrates that there are many ways to obtain different (approximate) scaling regimes on various frequency ranges. By the way, this also points out the physics of the problem well beyond the mathematical details that submerge the present manuscript. This is also in agreement with the fact that present numerical results on scaling are disappointingly simple compared to the heavy mathematical tools used in this paper.

Overall, the aforementioned problems, as well as the sharp contrast between this long paper (56 pages, 127 equations) with much more compact and rigorous papers (e.g., Karczewska and Lizama (2009)), invite to proceed to a thorough revision that will better build upon the present state-of-the art that could produced a terser paper with more rigorous, parsimonious mathematics. However, not only the mathematics need to be considerably cleaned up, but the main and challenging issue is to define a new "key novelty".

A sample of detailed comments
- introduction: the space-time fractional integration/differentiation for multifractals are surprisingly forgotten as well as others reviewed by Metzler and Klafter (2000), although being more general than the present factional time derivatives;
- Eq. 2 does not provide the Riemann-Liouville fractional derivative (but in fact the Caputo fractional derivative), furthermore no other equation does it;
- L.250: there are many reasons that an integration is not in general the inverse of a derivative, despite this is often considered to be true, including by the author;
- in Eq.3 and equations that follow, the Weyl derivative symbol could be simplified in $D_t^H$ since other fractional derivatives are not used (except in appendix A);
- L261-263: see general comment on this so-called generalisation;
- step-response function of the noise: in the present case (see general comments on the simplifying case $t_0 = -\infty$) it is in fact an impulse-response function of the motion, while in other cases it has much less generality and is much less generic than an impulse-response function. Therefore, it would be simpler to use only impulse-response functions. In particular, Eq.13 is immediate.
- Eqs.15-18 are classical (this should be said without any ambiguity) and are in fact a mathematical detour that is not indispensable, see below, especially comments on Appendix A;
- L.330: it is the Mittag-Leffler functions $E_{\alpha,\beta}$ that are often called "generalized exponentials", not the Green functions;
- L.332: see previous comments on step-response function and there is no difficulty to take care of possible divergences;
- L.341, equation without label: the symbol $G_{\zeta,H}$ is not defined and cannot inferred in a unique way from Eq.16), hence the origin of its r.h.s. is rather mysterious;

- Eq. 17: the leading term is missing, the summation should begin with $n=0$ , a precise reference should be given to the corresponding theorem that yields only a limited series, not an entire function as displayed. This is particularly important for $H=1$.
- L.347: poor display of $t^{-H}$, the present comment is unclear since only this kind of expansion could be expected;
- L367-369: this claims does not seem reasonable because Karczewska and Lizama (2009) worked on the more complex case of a finite $t_0$ and a complex vector-valued process, hence beyond the framework/approach of the present paper;
- sect.2.3:
  - it should rather begin with Eq. 36-c (with possible divergences) rather than from Eqs.19-20, which are furthermore written in a complicated manner to obtain a simple centralisation of the motion (Eq.21). In the latter equation, there is a one-way implication between the two equalities… which therefore should be stated in the reverse order.
  - Eq.34-b is a direct consequence of Eq.35 and $<dW(s)dW(s')>=ds$
  - there is a change of notations ($U$-> $Y$, $Q$->$Z$) that is not so helpful (only due to the introduction of the ad-hoc pre-factor $N_H$ in Eqs.19-20);
  - most developments constitute a mathematical detour, furthermore with too many small variations. In particular, the so-called Haar fluctuations of $Y$ are merely fluctuations of their integrals $Z$. It rather adds a distracting jargon than anything else and should be forgotten.
- section 3:
  - Eqs.40-41:
    - it is rather obvious that the normalisation coefficient $K_H$ is an imaginary number on the range $-1/2 < H < 1/2$ (and a real number on the range $1/2 < H < 3/2$, contrary to the claim of L.490), which is at odd with the non-negativeness of the structure function;
    - one then wonders what was really done in the following, since this expression of $K_H$ could not have been used;
    - as well as what is done outside of the range $-1/2 < H < 1/2$ and for the structure function of the noise (L.468 is rather ambiguous and/or inconsistant);
    - it seems the sign error results from an error on the argument of the sinus, which is indeed different according to another approach;
    - in any case, a lot of information is missing on how Eq.40 is obtained;
  - most developments of the section 3, particularly those around Eqs.40-41, as well as those of of Appendix B, would be greatly simplified if it would start with a developed sub-section 3.5 on Fourier space;
    - Eq.60 does not display relevant functions.
  - Appendix A
    - its goal is questionable since it aims to "use the R-L Green's functions to solve the Weyl fractional derivative equation" (L.1039), i.e., why to use a more complex approach than needed?
    - it seems mostly based on a circular reasoning: $_{-\infty}D = \lim\limits_{t_0 \to -\infty} (_{t_0}D)$ and R-L is a special case of $_{t_0}D$, but forgetting that it is in fact used for a fixed $t_0 = 0$. By the way, the Green function $G_0$ (Eq. 87 and others) is not defined;
  - Appendix B
    - to go from Eq.90 (improper integral of a series) to Eq.92 (series of improper integrals) requires conditions that are not discussed. They are a priori not satisfied. This explains the divergences of the resulting series;

- obviously, $A_{k,m}$ of Eq.94 does not correspond to $A_{n,m}$ of Eq.93: it rather corresponds to $A_{m,m}\Gamma(m+1)^2$, which is not relevant;
- there is no justification to sum in Eq.94 only the integrals that converge at infinity (L.1069)
- Eq.96 is obviously wrong: its r.h.s. should correspond to the summation of a geometric series (as foreseen from the l.h.s.), which is easy to obtain and quite different;
- it is extremely difficult to accept the statement "Since the series is divergent, the accuracy decreases if we use more than one term in the sum" (L.1080).
- the above inconsistencies bring into question all the claims that follow.

---

## Referee Report (RR2)

**A review of "Fractional relaxation noises, motions and the fractional energy balance equation" by Shaun Lovejoy, revised version.**

**General comments**

This new version of the manuscript puzzled me and it took a lot of effort to get back to it. Indeed, after having developed several key questions (state of the art, nonlinearity, Gaussianity, simplifying assumptions, spectral techniques, originality), my previous review invited the author "to proceed to a thorough revision that will better build upon the present state of the art that could produced a terser paper with more rigorous, parsimonious mathematics". Unfortunately, the new version of the manuscript is as long as the previous one, and the number of equations only reduced from 127 to 117, which remains at odds with more compact and rigorous papers on similar topics. One reason is that the present manuscript still reproduces mathematical developments that has been developed for the more complex problem of finite initial time and became classical, whereas this paper deals only with the simplified assumption $t_0 = -\infty$. Moreover, this is applied to a very particular case of (ordinary) fractional differential polynomials of "degree" n=2, whereas (partial) fractional differential polynomials of higher degrees have been already formally investigated, including with Fourier transforms (e.g., Podlubny, 1999). Although the author took into account some of these remarks, this duplication is maintained rather than fully acknowledging that the framework of fractional differential equations has been developed for decades across a wide range of disciplines, as evidenced by a number of key reviews.

Contrary to what the author claims, geophysics (especially nonlinear geophysics), cannot be seen as a world separated from physics, mechanics and mathematics and should be well aware of the general state of the art. Furthermore, the questions raised were not about the fact that "more steps and explanations than would be usual in a mathematical – or statistical physics journals", but merely not to forget that these steps were already carried out. Having a goal of application to a particular case in geophysics, does not exempt from any of the aforementioned obligations, including terse mathematical developments.

A second reason is that the use of Fourier techniques remains quite limited despite the author's claim to have taken into account the suggestion on the (potential) bringing-in of Fourier techniques. This mainly concerns the (new) appendices that are rather tentative with many inconsistencies and a cumbersome algebra. Readers still have to reach Sect.3.5 and Eq.65 to get a first information on spectra, whereas it is particularly trivial in the studied case. We are therefore still far from the authors's assertions of "the systematic use of Fourier and Laplace techniques" and "the Weyl fractional derivative […] is *naturally* handled by Fourier techniques" (emphasis added).

There are two new concerns: the reference to Budyko-Sellers type models turns out to be extremely weak and the section 4 does not fully address the prediction problem.

Overall, despite improvements including partial recognition of early results, I remain rather dissatisfied and can only suggest deepening the revision and eliminate inconsistencies.

**A new sample of detailed comments**

Sect.1 (L31-)

L 90-93:   This surprising statement seems rather irrelevant for most nonlinear geophysics, in particular it lacks empirical support.

L153: The link/relationship of the studied model with the conceptual Budyko-Sellers type Energy Balance models seems extremely weak as it completely lacks the space dimension that is central to

the interest of these models, in particular the albedo feedback that can trigger nonlinear bifurcations. This calls into question L94-96.

**Sect 3.2 (L 476-)**

Due to a lack of reorganisation of the paper, Sect 3.2 presents expansions that are primarily addressed in Sect.3.5 (and the corresponding appendix A). Their abrupt introduction in Sect.3.2 is mostly opaque, as is their discussion.

**Sect.3.5 (L654-)**

I recommended to "start with a developed sub-section 3.5 on Fourier space". This is still not exactly the case, although some steps have been taken in this direction, particularly within the new appendices A and B. However, instead of simplifying the necessary algebraic developments, most of the new ones are devoted to numerous expansions that are scarcely used, as well as a tour of classical special functions. Ironically, the popular (generalized) hypergeometric functions are not directly included, but the G Meijer function is not forgotten. The main problem is that there are inconsistencies, see detailed comments on the new appendices A and B.

**L656-657**

"…However, it is easier to determine it directly from the fractional relaxation equation…"
is typical of the orientation of the previous versions of this paper, which should not be maintained.

**L659**

The online equation depends on the precise definition of the Fourier transform, which is not given

**L668**

Here and at other places, there is a (surprising) confusion between $< |\hat{\gamma}|^2 >$ and its spectrum, contrary to Eq.3.

**L.674-677**

Eq.67 is trivial consequence of Eq. 66, and in fact a demonstration of Parseval's theorem rather than a consequence of it.

**L681**

A well-oriented paper would rather inverse the implications from Fourier to the physical space

**Sect.3.6 (L685-)**

This is a lengthy section, with numerous graphs instead of a selection of a few significant ones. This is presumably due to the fact that it is hardly modified from previous versions, therefore with limited inputs from new understandings, while remaining rooted in rather general considerations.

**Sect.4 (L796-)**

I disregarded Sect.4 in my previous review, simply because of the many requests for clarification of previous sections. Unfortunately, like Sect.3.6, this section has hardly changed. The consequences are worse for this section: having paid some attention to spectra might have reminded the author of some works on predictability, which rather show the present "prediction on past values" is not optimal in the physical sense. This is hidden by a tedious algebra (Eqs.70-73) performed on the ad-hoc predictor $\hat{Y}_\tau$ and its error $E_\tau$ with respect to the noise $Y_{H,\tau}$ (surprisingly, the sub-index $H$ is kept only for the last quantity), in particular to highlight an orthogonality between $\hat{Y}_\tau$ and $E_\tau$. This

property is wrongly assumed to ensure $\hat{Y}_\tau$ to be "optimum", whereas it only ensures $\hat{Y}_\tau$ to be an orthogonal projection of $Y_{H,\tau}$ on a given subset and therefore to be optimum only with respect to this subset. At best, this is a necessary condition, but not sufficient. The main question is to define the optimal subset.

Furthermore, a fundamental difficulty is that the chosen predictor is defined with the help of fractional integrals over the past of the noise $\gamma(t)$ ($t\leq0$), not using observables over the past (e.g., $Y_{H,\tau}(t)$ $t \leq 0$). There is no indication how the noise $\gamma(t)$ ($t\leq0$) is inferred from given observable(s). The title of the section is therefore inappropriate, as no prediction is actually made. This is in fact confirmed by the applications and figures presented in this section: they do not correspond to prediction from the past, but are limited to comparisons of prediction skill indicators of fGn and fRn (see comments on these indicators below).

L802-803
"The emphasis on past values is particularly appropriate since in the fGn limit, the memory is so large that values of the series in the distant past are important"
L841-844
"in the limit H→1/2, we have perfect skill for fGn forecasts (this would of course require an infinite amount of past data to attain)."
These are two examples of confusing sentences (in various places of the paper) on long range memory. It should be made clear whether or not there is a divergence (at $-\infty$), as the consequences are totally different, as ironically pointed out by my reference to a finite date of the Big Bang (and the Earth). This has unfortunately not been understood. Contrary to the author's assertion, it was worth calling attention to this approximation and so it remains.

L831
Because of the above questions on the "optimum", it is doubtful that $S_{k,t}(t)$ (Eq.76) is an appropriate score indicator. Note that the subindex $k$ is not explained.

Sect.5 (L911-)
Conclusions should be numbered 5, not 4. They should focus on the results of the paper, instead of having lengthy elaborations on the potential applications to geophysics, to which the present paper does not contribute at all. Because of the above remarks, the list of results need to be considerably revised. This may lead to a more appropriate title of the paper.

L 943-945
The mention of stationarity without any indication of its type/order is rather surprising and misleading, especially with respect to climate. Moreover, this there is no direct link to the fixing of the initial time at the Big Bang date or later. For instance, if the former remove transients, it means that transients exist.

L946
"Beyond the *proposal* that the FEBE is a good model for the Earth's temperature…"
As already mentioned, this paper does not contribute to that proposal.

Former appendices A and B
I brought them into question and can only positively appreciate their removal.

New appendix A

It is devoted to evaluate the correlation function with the help of the inverse Fourier transform of the spectrum. Instead of discussing what is really needed for physics and what would be the simplest mathematical techniques to use, this appendix plunges immediately into mathematical details of an inverse Laplace-Fourier transform along a given contour in the complex plane (there is twice a typo "-ve plane"). Moreover, these details are far from being consistent. For instance,

- the contour presented in L1011-1012 is not consistent with a branch cut of the power law $z^H$
- this branch cut is furthermore confused with given poles (L1016-1017)
- half of poles are forgotten, whereas they are essential to ensure an important symmetry of the solutions
- L1022 the form of Eq.A3 is hardly understandable, especially presented in a vague way as being "a contribution" of the poles
- L1034 Eq.A4, the "convenient coefficient" $D_n$ could have been introduced more conveniently after Eq.A12, presently in L1067
- L1048-1059, Eq.35 is certainly not the simplest way to compute $V_H$ (with the help of a two fold integration), as well as its expansion
- L1074, the identity of Eq.A15 is exact, but what does bring the introduction of the Hurwitz-Lerch transcendental function? By the way, the definition given is only that of the Lerch transcendental function
- L1101-1112 requires to consider the exact definition of the Hurwitz-Lerch transcendental function.

New appendix B

This appendix is devoted to the special case H=1/2 that has been already investigated by Mainardi and Pironi (1996) under the term " fractional Langevin equation". The author mentions it briefly "for some early results". Unfortunately, instead of starting from these results, as well as those in appendix A, one starts more or less from scratch with the help of a tedious algebra of round-trip transforms between physical and Laplace space. Furthermore, the computed correlation does not match early results.

---

## Author Response (AR2)

12.12.20

Dear Daniel;

The paper has now benefitted from 5 referees, a personal record. These have been helpful -  and with the partial exception of referee 5, they have been uniformly positive. Although detailed comments and responses are given below, stimulated by referee 5, I would like to take this opportunity to make some general remarks about NPG.

Referee 5 is clearly a mathematician unfamiliar with Nonlinear Processes in Geophysics, in particular, the fact that fractional equations have not been much used in geoscience. His comments on fGn, fBm indicate that he is much more familiar with the random walk/ diffusion literature, referring me again to the random walk, diffusion literature that does not contain the precise results needed for the applications to fractional energy balance equation. While this may seem surprising (indeed I personally was surprised), none of the five referees have claimed that any of the key results have been published elsewhere (by this I mean the second order statistical processes of the new processes fRn, fRm). None of the referees have claimed that the results are not unoriginal. Indeed, science constantly throws up new mathematical challenges.

Referee 5 in particular seems uninterested in the geoscience applications and is insensitive to the need to develop the material with more steps and explanations than would be usual in a mathematical – or statistical physics journals. Although perhaps irritating to mathematicians, I think these will prove useful  to geophysicists.

The NPG journal was precisely developed so that papers of the present type could find a place.

You as editor repeatedly ask what is the key novelty. It remains the precise statistical properties of the fRn, fRm processes, now made more detailed thanks to the use of Fourier techniques (as suggested by referee 5, but in fact that I had already implemented while waiting for his comments).

-Shaun

**Referee 4:**

**Suggestions for revision or reasons for rejection (will be published if the paper is accepted for final publication)**

The manuscript has been strongly modified. More references have been added and the proposed model is discussed in relation with available literature in the field of fractional stochastic modeling. It is also now more self-contained and is not referring to submitted manuscripts.

I have only one comment: the FEBE is mentioned in the title, in the abstract and the introduction. But after this for about 40 pages the work and presentation is on a new proposal, called fractional relaxation noise (fRn) and fractional relaxation motion (fRm): their definition, their scaling properties, their synthesis and their prediction properties.

It seems that the FEBE (fractional energy balance equation) is only a motivation for this work but is not the main issue. I therefore suggest to change the title to "Fractional relaxation noises and motion: scaling properties and prediction", or something similar. Also I suggest to modify the abstract accordingly. In the introduction it may be indicated more clearly that FEBE is only a motivation to propose such model and that the work is more and the property of such model (which may apply to other situations that FEBE).

*Au: I thank the referee for his positive review. Upon reflection, I decided to keep the old title since on the one hand there are now four other papers on the FEBE and on the other to emphasize that this paper is intended more as a contribution to the nonlinear geophysics literature rather than the fractional differential equation literature. Also, the paper was resolutely written for a geophysics audience, not a mathematical one and there are numerous indications, references to the geoscience applications.*

**Referee 5:**

**General comments**

Despite the reference increase, this revised version has still important gaps on the state-of-art and therefore remains unclear on the truly original contributions of this manuscript. There has been indeed an abundant literature on fractional differential equations, in particular on the relaxation-oscillation equation that is the topic of this paper.

The choice of this linear equation is debatable with respect to the focus on nonlinear geophysics of this special centennial issue.

*Au: Throughout its brief history, Nonlinear Processes in Geophysics has had a constant theme of using stochastic models for strongly nonlinear systems. In the case of turbulence, the corresponding stochastic models (multifractal cascades) were themselves nonlinear, but often linear stochastic models can be could models for deterministic nonlinear systems. That is one of the arguments for applying the fractional relaxation equation to the Earth's energy balance.*

In this respect, there are several rather surprising statements, such as the second part of the sentence: "The choice of a Gaussian white noise forcing was made both for theoretical simplicity but also for physical realism" (L.171-172). This is unfortunately in direct agreement with the oft-quoted, ironic Chester Kisiel's pray to the theoretical hydrologist: "Oh, Lord, please, keep the world linear and Gaussian!".

*Au: I have spent several decades focusing on nonlinear stochastic models and have now well documented the fact that macroweather in time (but not space) is the low intermittency, quasi-Gaussian exception to otherwise strongly intermittent, multifractal regimes at higher weather frequencies or lower climate, mega-climate frequencies, see especially [Lovejoy, 2018]. In other words, the model presented in this paper is quite plausibly pertinent for the Earth Energy balance from months to (at least) decades in time scale.*

Similarly, the claim of "the paucity of mathematical literature on stochastic fractional equations (see however [*Karczewska and Lizama*, 2009])" (L.78) is in contradiction with various review papers, in particular the Physics Report of Metzler and Klafter (2000) that focuses on "various generalisations to fractional order [that] have been employed, i.e. different fractional operators [that] have been introduced to replace either the time derivative or the occurring spatial derivatives, or both" and has more than 300 references. More generally, the author's emphasis on opposing fractional vs. integer order differential equations, both linear, seems rather outdated.

*Au: Clearly in the physics and mathematics literature, there has been an explosion in fractional equations in the last decades. This is indeed underlined in the introduction. However, at present, in the geophysics literature (to which this paper is a contribution), there are actually very few papers applying fractional equations. In addition, most of the results that do exist concern random walks and other nonstationary processes. Most importantly, as far as I know, none of the key results needed for applications (the second order statistics) are available elsewhere.*

The tentative argument in favour of having an "enormous memory" with the help of a lower integration bound of the so-called Weyl fractional integration/differentiation (which in fact could be traced back to Liouville (1832)) is overstated, while it basically corresponds to an over-simplification, not only with respect to the finite date of the Big Bang, but also to Earth climate.

*Au: It is common in both physics and geophysics to ignore the big bang and take time integrals from $-\infty$. I'm not sure that it is worth bringing attention to this approximation.*

Contrary to the author's claim that "the interval between an initial time = 0 and a later time t [...] is the exclusive domain considered in Podlubny's mathematical monograph on deterministic fractional differential equations [Podlubny, 1999]" (L136-138), this monograph, as several others, does deal with the Weyl fractional integration/differentiation and a more careful reading of it might have helped to simplify and make more rigorous the present manuscript. Let us clarify that a lot of efforts had been spent for the finite $t0$ case (e.g., works of Gorenflo, Mainardi and collaborators) due to the fact it was much more difficult than the (negative) infinite case: in fact, it required to define a new fractional derivative (Caputo, 1967) to handle the initial conditions, whereas the classical Riemann-Liouville failed to do it. All these important technicalities vanish for the (negative) infinite case, basically because the Laplace convolution reduces to a Fourier convolution.

*Au: The aim of this paper is to develop the basic statistical properties of the noise driven fractional relaxation- oscillation equation.  In the new version significantly more precise results have been obtained using Fourier and Laplace techniques.  As far as I can tell, the results are original and they are needed for applications, now in several papers: [Lovejoy, 2020a; b], [Lovejoy et al., 2020; Procyk et al., 2020].*

A paradox of this paper is that it claims to be innovative by focusing on the Weyl fractional integral/ derivative ("the key novelty of this paper is therefore to consider the FEBE as a Weyl fractional Langevin equation", L.914), while being lost in many mathematical details (e.g., Mittag-Leffler functions) that are rarely necessary for this simplifying case (e.g., the composition of fractional integrals/derivatives is then commutative), as well as not taking advantage of other structural simplifications resulting from the combination of the linear and Gaussian assumptions (e.g., linear stability). On the contrary, the (potential) bringing-in of Fourier techniques is mostly limited to spectra in the short sub-section 3.5 "Spectra".

*Au: As indicated above, the aim of the paper was to elucidate the main statistical properties as simply as possible with an aim at the geophysics applications.  The more precise results in the new iteration (that are original as far as I can tell) are precisely due to the systematic use of Fourier and Laplace techniques.*

It seems that the main results are the scaling behaviours of the studied noise and motion, which the author calls fractional relaxation noise and motion, for small and large time lags. Unfortunately, it is not clear what is analytically obtained and to which level. Often information is missing on important issues, whereas there are numerous mathematical displays. Unfortunately, mathematical rigour is not always there, contrary to mathematical pitfalls that bring into question what is really $t0 = -\infty$ _-- 1obtained (see examples in detailed comments). A clear synthesis, with a comparison with their classical counterparts of fractional Brownian noise and motion, is unfortunately missing.

*Au: It was there, but not clear enough.  The new version hopefully makes the similarities and differences more obvious.  We clearly establish the difference between fRn and fGn and develop approximations for the differences in their statistics (the new appendix A.3) and in section 4 we systematically compare their predictability skills.*

The aforementioned problems are amplified in the sections on simulations and prediction and make them very difficult to evaluate before these problems will be solved. A final general comment is that the author's claim that the studied noises and motions are generalisation of the Brownian ones is not obvious. Indeed, what could be the new generality gained with their help?

*Au: In physical applications of fGn and fBm, the range over which scaling holds is always finite. In applications to the Earth's energy balance this scale has the simple interpretation as a relaxation time. We show much below this relaxation time (which is unity in the nondimensional processes studied here) - that the small scale limit of the new noises and motions studied here – that we do indeed recover fGn and fBm behaviour. The new generality is that it allows treatment of both scales smaller and larger than the relaxation time, including the transitional behaviour.*

On the contrary, an important and generic property has been lost: scaling. This is a direct consequence of the introduction of a characteristic time (presently hidden by the non-dimensionalisation of equations) due to the presence of two time derivatives of different orders ($H$ and 0) instead of a unique one for (fractional) Brownian noises ($H$ or 0). With no surprise, the case of fractional differential polynomials of "degree" $n>2$ has been already formally investigated (e.g., Podlubny, 1999). This just illustrates that there are many ways to obtain different (approximate) scaling regimes on various frequency ranges. By the way, this also points out the physics of the problem well beyond the mathematical details that submerge the present manuscript. This is also in agreement with the fact that present numerical results on scaling are disappointingly simple compared to the heavy mathematical tools used in this paper.

*Au: We have added new emphasis on the physics, especially the fact that the H=1/2 special case can be derived simply as a consequence of the classical continuum mechanics heat equation but with radiative-conductive surface boundary conditions. Thanks to the extensive use of Fourier and Laplace techniques, the analytic and numerical results are significantly stronger in the revised version.*

Overall, the aforementioned problems, as well as the sharp contrast between this long paper (56 pages, 127 equations) with much more compact and rigorous papers (e.g., Karczewska and Lizama (2009)), invite to proceed to a thorough revision that will better build upon the present state-of-the art that could produced a terser paper with more rigorous, parsimonious mathematics. However, not only the mathematics need to be considerably cleaned up, but the main and challenging issue is to define a new "key novelty".

*Au: The key novelty is the (now quite full) derivation of the statistical properties (series expansions about the origin, asymptotic expansions) including predictability properties needed for monthly and seasonal forecasting. In the previous version of the paper, only the leading terms in the expansions were given. If these results are available elsewhere, please advise.*

**A sample of detailed comments**

- introduction: the space-time fractional integration/differentiation for multifractals are surprisingly forgotten as well as others reviewed by Metzler and Klafter (2000), although being more general than the present fractional time derivatives;

*Au: Multifractals were discussed in several places, the term appeared 7 times. Metzler and Klafter (2000) discussed random walks and do not give the results discussed here, we have now referenced this paper, thank you.*

- Eq. 2 does not provide the Riemann-Liouville fractional derivative (but in fact the Caputo fractional derivative), furthermore no other equation does it;

*Au: Thanks, I have modified the definition and improved this paragraph accordingly.*

- L.250: there are many reasons that an integration is not in general the inverse of a derivative, despite this is often considered to be true, including by the author;

*Au: This property was not used in the further developments and is now noted as suggested.*

- in Eq.3 and equations that follow, the Weyl derivative symbol could be simplified in since other fractional derivatives are not used (except in appendix A);

*Au: At the referee's suggestion, we removed the old appendix A. However there are very few uses of the fractional derivative symbol so that leaving the notation is not so unwieldy, it has the advantage of underscoring the differences with respect to the usual applications.*

- L261-263: see general comment on this so-called generalisation; - step-response function of the noise: in the present case (see general comments on the simplifying case ) it is in fact an impulse-response function of the motion, while in other cases it has much less generality and is much less generic than an impulse-response function. Therefore, it would be simpler to use only impulse-response functions. In particular, Eq.13 is immediate.

*Au: Thanks for the comment. I have added the information about $G_{1,H}$ being the impulse response for the motions, although this could be a little misleading since there is the nontrivial issue of low frequency divergences that motivate the present development.*
    *An additional reason for spending time on this mundane issue is that the Energy Balance literature often uses the step response because it give direct information about approach to thermodynamic equilibrium: in energy contexts, it is easier to interpret physically than the impulse response.*

- Eqs.15-18 are classical (this should be said without any ambiguity) and are in fact a mathematical detour that is not indispensable, see below, especially comments on Appendix A;

*Au: Yes, from a strictly mathematical point of view it is not needed, but the paper was written for geoscientists for whom this will be new. The figure is also quite useful as is the asymptotic*

*expansion (eq. 17). Eq. 18 underlines the rather special nature of the usual Energy Balance Equation. I thought it was clear, but I have underlined that this is classical.*

- L.330: it is the Mittag-Leffler functions that are often called "generalized exponentials", not the Green functions; - L.332: see previous comments on step-response function and there is no difficulty to take care of possible divergences;

*Au: Thanks for the correction on the term "generalized exponentials", for the divergences, at the very least, $G_{1,H}$ rather than $G_{0,H}$ is useful for the numerical simulations described later. We have indicated this.*

- L.341, equation without label: the symbol is not defined and cannot inferred in a unique way from Eq.16), hence the origin of its r.h.s. is rather mysterious; $G_{\zeta,H}$.

*Au: Thanks, there were some words missing as well as the equation number. It should now be fine.*

- Eq. 17: the leading term is missing, the summation should begin with $n=0$ , a precise reference should be given to the corresponding theorem that yields only a limited series, not an entire function as displayed. This is particularly important for $H=1$.

*Au: For 0<H<1 and 1<H<2 (as indicated), Eq. 17 is correct: the n=0 term for $G_{1H}$ was written independently to emphasize this asymptotic limit. We have nevertheless rewritten it as suggested by the referee. The H = 1 case is given in eq. 18.*

- L.347: poor display of $t^{-H}$, the present comment is unclear since only this kind of expansion could be expected;

*Au: Yes, as expected now indicated.*

- L367-369: this claims does not seem reasonable because Karczewska and Lizama (2009) worked on the more complex case of a finite and a complex vector-valued process, hence beyond the framework/approach of the present paper;

*Au: Thanks, we revised the sentence accordingly.*

- sect.2.3: - it should rather begin with Eq. 36-c (with possible divergences) rather than from Eqs.19-20, which are furthermore written in a complicated manner to obtain a simple centralisation of the motion (Eq.21). In the latter equation, there is a one-way implication between the two equalities... which therefore should be stated in the reverse order.

*Au: I appreciate that it could be possible to start with the noise (eq. 36) and derive the motion (eq. 19). However, I have followed the usual route following Mandelbrot and Biaginni et al. I think that it makes the divergence issues more transparent. In eq. 21 , I have reversed the order as suggested.*

- Eq.34-b is a direct consequence of Eq.35 and $<dW(s)dW(s')>=ds$ - there is a change of notations ($U$-> $Y$, $Q$->$Z$) that is not so helpful (only due to the introduction of the ad-hoc pre-factor $NH$ in Eqs.19-20);

*Au: The text attempts to consistently derive the noises from the motions (rather than the inverse) so that as indicated, 35 is derived from 34. It is then explicitly stated (line 445) that the derivation could have started the other way around as the referee suggests, although the divergences would need particular care. The normalizations are introduced so that the results may be directly compared with the more familiar fGn properties; the latter are often defined in terms of their (normalized) second order statistics (not as solutions to fractional equations as here).*

- most developments constitute a mathematical detour, furthermore with too many small variations. In particular, the so-called Haar fluctuations of $Y$ are merely fluctuations of their integrals $Z$. It rather adds a distracting jargon than anything else and should be forgotten.

*Au: The aim of the paper is to develop results useful for geoscientists seeking to apply the results to the Earth energy balance. Over the years, the use of Haar fluctuations derived from Haar wavelets, has been very helpful in clarifying climate variability over huge ranges of scales. Indeed, it helped to point out a missing factor of $10^{15}$ in the standard model of atmospheric variability [Lovejoy, 2015]. Therefore the referee is therefore correct that from a strict mathematical point of view, the results on Haar fluctuations are not needed however, for NPG readers and energy balance applications, they will be valuable.*

- section 3: - Eqs.40-41: - it is rather obvious that the normalisation coefficient is an imaginary number on the range (and a real number on the range, contrary to the claim of L.490), which is at odds with the non-negativeness of the structure function;

*Au: The problem was a sign error in eq. 40, 41, they have been corrected. Clearly we agree: it was indeed correctly stated (line 490) that for $H>1/2$ $K_H^2<0$ which is the same as saying $K_H$ is imaginary.*

- one then wonders what was really done in the following, since this expression of could not have been used;

*Au: Due to Gamma function identities, there are numerous ways to write the normalization factor, converting from one form to another led to the sign typo. The correct sign was used throughout.*

- as well as what is done outside of the range and for the structure function of the noise (L.468 is rather ambiguous and/or inconsistent);

*Au: The normalization is quite standard for fGn, fBm; as indicated often the latter are defined with this normalization. This has now been indicated.*

- it seems the sign error results from an error on the argument of the sinus, which is indeed different according to another approach;

*Au: Yes, see above.*

- in any case, a lot of information is missing on how Eq.40 is obtained; - most developments of the section 3, particularly those around Eqs.40-41, as well as those of Appendix B, would be greatly simplified if it would start with a developed sub-section 3.5 on Fourier space;

*Au: The new more complete derivations (starting in Fourier space) in the new appendix A should answer this question.*

- Eq.60 does not display relevant functions.

*Au: It was for applications, but now eliminated.*

- Appendix A - its goal is questionable since it aims to "use the R-L Green's functions to solve the Weyl fractional derivative equation" (L.1039), i.e., why to use a more complex approach than needed? - it seems mostly based on a circular reasoning: and R-L is a special case of , but forgetting that it is in fact used for a fixed . By the way, the Green function (Eq. 87 and others) is not defined;

*Au: OK, I have eliminated the old appendix A.*

- Appendix B - to go from Eq.90 (improper integral of a series) to Eq.92 (series of improper integrals) requires conditions that are not discussed. They are a priori not satisfied. This explains the divergences of the resulting series; $t{-}H$ $t0$ $KH$ $-1/2 < H < 1/2$ $1/2 < H < 3/2$ $KH$ $-1/2 < H < 1/2$ $-\infty D = \lim t0\rightarrow-\infty$ _$(t0D)$ $t0D$ $t0 = 0$ $G0 - 3$
- obviously, of Eq.94 does not correspond to of Eq.93: it rather corresponds to , which is not relevant;
- there is no justification to sum in Eq.94 only the integrals that converge at infinity (L.1069)
- Eq.96 is obviously wrong: its r.h.s. should correspond to the summation of a geometric series (as foreseen from the l.h.s.), which is easy to obtain and quite different;
- it is extremely difficult to accept the statement "Since the series is divergent, the accuracy decreases if we use more than one term in the sum" (L.1080).
 - the above inconsistencies bring into question all the claims that follow. $Ak,m$ $An,m$ $Am,m$ $\Gamma(m + 1)2 - 4$

*Au:  I have completely replaced the old appendix B with new material based on Fourier and Laplace techniques that enabled me to obtain full series (including asymptotic series) expansions of both $R_H$, $V_H$.  The new material is now also included in a highly revised section 3.2.*

**References**

Lovejoy, S., A voyage through scales, a missing quadrillion and why the climate is not what you expect, *Climate Dyn.*, *44*, 3187-3210 doi: doi: 10.1007/s00382-014-2324-0, 2015.

Lovejoy, S., The spectra, intermittency and extremes of weather, macroweather and climate, *Nature Scientific Reports*, *8*, 1-13 doi: 10.1038/s41598-018-30829-4, 2018.

Lovejoy, S., The Half-order Energy Balance Equation, Part 1: The homogeneous HEBE and long memories, *Earth Syst .Dyn. Disc.* doi: https://doi.org/10.5194/esd-2020-12, 2020a.

Lovejoy, S., The Half-order Energy Balance Equation, Part 2: The inhomogeneous HEBE and 2D energy balance models, *Earth Sys. Dyn. Disc.* doi: https://doi.org/10.5194/esd-2020-13, 2020b.

Lovejoy, S., Procyk, R., Hébert, R., and del Rio Amador, L., The Fractional Energy Balance Equation, *Quart. J. Roy. Met. Soc.* , *(under revision)*, 2020.

Procyk, R., Lovejoy, S., and Hébert, R., The Fractional Energy Balance Equation for Climate projections through 2100, *Earth Sys. Dyn. Disc.*, *under review* doi: org/10.5194/esd-2020-48 2020.

---

## Author Response (AR3)

*Responses in Italics;*

Dear Shaun,

First of all, I would like to clarify that there were only 4 referees and that the first 3 were not "uniformly positive".

*My only information was that the referees were numbered 1 -5 and I responded to each, sometimes more than once. It would have been helpful to know that there were only four different people involved!*

For example, referee 3 was very critical and referee 2 actually suggested a major revision. Referees 3 and 4 both complained bitterly about the lack of key references to the literature and actually helped to improve the state of the art considerably,

*This was helpful but was addressed in the January 2020 revision.*

although reviewer 4 is still not fully satisfied and believes that this remains a source of problems for your paper. I can also mention that I have received abrupt refusals to review.

*I appreciate that it is not easy to find reviewers!*

The referee 4 had strongly suggested to use the Fourier transform to derive the statistical characteristics of the fractional Langevin equation. The new Appendix A tries to respond to this and I am happy to acknowledge that it is the core of the present revision. However, the referee 4 raises several issues around the main question: how to do it in a very simple way? It has taken me some time to assess the extent to which this calls into question current algebraic developments. I am afraid this is the case since the beginning of the new Appendix A with its unfortunate rush to mathematical details of a given approach instead of trying to optimise the approach and to avoid cumbersome developments. Indeed, it should rely more on physics than a series of "convenient" equations.

*The adoption of Fourier techniques had been done after January 2020 but before referee 4 suggested it: it was indeed a good idea to go beyond the old approaches used by Mandelbrot. Appendix A rushes to mathematical details because the physics is dealt with in the main text (and now in numerous publications, see below). Following your suggestion, this has now been augmented and the entire approach based on Fourier and Laplace methods. Unfortunately, the real space Green's function approach is needed for the predictability results, so they are also given. They also allow an easier entry to the topic for geoscienctists.*

*Over the last years, the science has also advanced so that results on the fGn forced fractional relaxation process are also needed for applications. Therefore, the new version has been slightly extended to cover this possibility.*

This may partly explain why Appendix B relies on a different technique than Appendix A.

*Yes, now, appendix B is more clearly cast the half order case in the same Fourier/Laplace framework as the rest and has been (mildly) extended to the fGn forcing case.*

But the main problem with Appendix B is that it actually addresses a distinct problem. This is an example of the inconsistencies briefly mentioned in the report, some of which I fear are generated by repeated use of incomplete notations, which, for example, do not make explicit the fact that only part of the solution is addressed.

*The new version addressed this (see above).*

The referee's report raises other important questions, such as the relationship between this fractional Langevin equation and the actual Budyko-Sellers model, in fact, the actual predictive capabilities of this equation.

*Since this paper was submitted in July 2019, there have been many publications addressing this in great detail. Several new references have been included in the new version and more physical discussion has been given.*

***Theory:***
*Starting with the classical continuum equation (and then its fractional generalizations), the exact link with the Budyko-Sellers approach was given in a two part series: [Lovejoy, 2021a; b]. The FEBE was also derived phenomenologically in [Lovejoy et al., 2021]; this paper included a summary of the dimensional FEBE statistics (citing to the 2019 version of this paper).*

***Applications:***
 *First, there was [Del Rio Amador and Lovejoy, 2019] (building on [Lovejoy, 2015], [Lovejoy et al., 2015]) who investigated the predictability of the global temperature using the high frequency fGn approximation to fRn. There was also [Hébert et al., 2021] who investigated the low frequency approximation for climate projections to 2100 with [Procyk et al., 2020], [Procyk, 2021] directly using the full FEBE to improve on this further (notably using several of the results of the NPG paper; the projections had much lower uncertainties than the GCMs). More recently, [Del Rio Amador and Lovejoy, 2021a; Del Rio Amador and Lovejoy, 2021b] addressed the regional application of the FEBE model for monthly and seasonal predictions (using the fGn approximation) showing that they were comparable to or better than the GCM alternatives.*

Due to the above-mentioned inconsistencies, especially in the new materials, the paper still needs thorough revisions. The safest, simplest and most satisfactory way would be to modify the approach followed in Appendix A. But this would require a major rethink and an alternative might be to eliminate the inconsistencies in the current approach and make it more accessible despite its inherent complexity.

*The need to quantify the statistics in real space while nevertheless using Fourier/Laplace techniques makes this paper complex. However, the new version eliminates the last traces of the Mandelbrot approach and is much more streamlined. Putting the technical part in appendices makes the main text lighter and more accessible. Thanks for the suggestions.*

To speed up the process, I choose the option that the next review will be done only by the editor despite the expected thorough revisions.

*I appreciate that it is not easy to find reviewers but the current version should be relatively glitch free!*

Best regards,
Daniel

**References:**

Del Rio Amador, L., and Lovejoy, S., Predicting the global temperature with the Stochastic Seasonal to Interannual Prediction System (StocSIPS) *Clim. Dyn.*, *53*, 4373–4411 doi: org/10.1007/s00382-019-04791-4., 2019.

Del Rio Amador, L., and Lovejoy, S., Using regional scaling for temperature forecasts with the Stochastic Seasonal to Interannual Prediction System (StocSIPS), *Clim. Dyn.* doi: 10.1007/s00382-021-05737-5, 2021a.

Del Rio Amador, L., and Lovejoy, S., Long-range Forecasting as a Past Value Problem: Untangling Correlations and Causality with scaling, *Geophys. Res. Lett.*, 1-12 doi: 10.1029/2020GL092147, 2021b.

Hébert, R., Lovejoy, S., and Tremblay, B., An Observation-based Scaling Model for Climate Sensitivity Estimates and Global Projections to 2100, *Climate Dynamics*, *56*, 1105–1129 doi: doi.org/10.1007/s00382-020-05521-x, 2021.

Lovejoy, S., Using scaling for macroweather forecasting including the pause, *Geophys. Res. Lett.*, *42*, 7148–7155 doi: DOI: 10.1002/2015GL065665, 2015.

Lovejoy, S., The Half-order Energy Balance Equation, Part 1: The homogeneous HEBE and long memories, *Earth Syst .Dyn.* , *(in press)* doi: https://doi.org/10.5194/esd-2020-12, 2021a.

Lovejoy, S., The Half-order Energy Balance Equation, Part 2: The inhomogeneous HEBE and 2D energy balance models, *Earth Sys. Dyn.*, *in press* doi: https://doi.org/10.5194/esd-2020-13, 2021b.

Lovejoy, S., del Rio Amador, L., and Hébert, R., The ScaLIng Macroweather Model (SLIMM): using scaling to forecast global-scale macroweather from months to Decades, *Earth Syst. Dynam.*, *6*, 1–22 doi: www.earth-syst-dynam.net/6/1/2015/,  doi:10.5194/esd-6-1-2015, 2015.

Lovejoy, S., Procyk, R., Hébert, R., and del Rio Amador, L., The Fractional Energy Balance Equation, *Quart. J. Roy. Met. Soc.* , 1–25 doi: https://doi.org/10.1002/qj.4005, 2021.

Procyk, R. (2021), The Fractional Energy Balance Equation: the Unification of Externally Forced and Internal Variability, MSc thesis, 111 pp, McGill University, Montreal, Canada.

Procyk, R., Lovejoy, S., and Hébert, R., The Fractional Energy Balance Equation for Climate projections through 2100, *Earth Sys. Dyn. Disc.*, *under review* doi: org/10.5194/esd-2020-48 2020.

---

## Author Response (AR4)

**Comments to the author**:

Dear Shaun,

To speed up the review process, I had chosen the option that "the next review will be done only by the editor despite the expected thorough revisions". Unfortunately, I gradually became aware that there was no response to the referee and the track change copy is unusable, providing only a grey rectangular on the right side of the text. I initially thought that this could be due to bugs of the review system, but after some investigation it does not seem so. Therefore, please provide these documents, slightly updated if necessary.

*I would have appreciated an acknowledgement of the substantial revisions that were made in the previous (fourth) version, they seem to have been undetected. In any case, with the exception of the first four referees which were answered in January 2020 (the first revision), the other comments I have received have been of the very general, very vague and slightly negative variety.*

*In other words, the reviews and comments since January 2020 seem to (barely) conceal a desire to kill the paper, they refrain from giving enough specifics to allow the paper to be improved or defended. We are both familiar with the genre! This contrasts with the attitude of the first four referees that were generally quite positive: the only one to suggest major changes had rather mild specifics that were addressed nearly two years ago. I went along with the three (post January 2020) major revisions because the process was so slow that in the meantime I was able to make my own improvements with virtually no guidance.*

*Certainly your latest round of comments will only allow me to add the three sentences indicated in the following. I will make these changes if you agree to publish the paper when they are done.*

They may help to provide more complete answers to my own report that was primarily trying to highlight the main points of the referee's report, which is more detailed.

*Until now, the only specific comments that I had were several that were addressed in the first round of responses and revisions in January 2020. My second round of changes were made while waiting for the new $5^{th}$ referee who responded 13 months into the process. He noticed a sign error (an obvious typo with no consequence) and suggested Fourier techniques. By then I had already added the Fourier approach (without his suggestion), but I still kept the Mandelbrot approach in the main part of the paper with the Fourier approach in the appendix. Following your comments (25 months into the process), I put the Fourier approach into the main part of the paper and removed the Mandelbrot approach altogether. However, real space results are still needed for the important predictability part (section 4) that must be done in real space.*

For instance:
The reference made to physics for this Appendix A (instead of a rush to mathematical details) was misunderstood as a justification of the fractional Langevin equations, while the referee specifically challenged the Fourier technique chosen.

> *It would have been nice to see something specific in this direction since it isn't obvious what an alternative Fourier approach might consist of! At the moment I can't even guess!*

However, I have accepted that the present paper could remain in the same approach despite the resulting complexity.
There is still a loss of symmetry along the early developments of Appendix A that should at least noted.

> *There is already a sentence explaining why both Fourier and real space results are needed, but I can add an appropriate sentence apologizing for the resulting lack of symmetry.*

The strong reference to the Budyko-Sellers model is not justified, as its main physical process is absent.

> *This is a bit ridiculous! When the paper was originally submitted, I admit that there was only an "announcement" of the result that had been properly published. However over the 28 months since submission, there have been 4 publications with "FEBE" (or the h=1/2 case, HEBE) in the title! One of the latter- the application of the FEBE to temperature projections to 2100 (submitted in March 2020) is now waiting the referees after "minor revisions". Three additional published papers already reference the FEBE (a full list was also appended to the previous response). These publications outline two quite different derivations of the FEBE including one that is identical to Budyko-Sellers except that it uses the correct radiative-conductive surface boundary condition that Budyko-Sellers got wrong (see the two HEBE papers).*

> *In other words, the strong reference to Budyko-Sellers is fully justified by the cited literature and needs no further support in the present paper!*

The referee also strongly questioned several aspects of the Sect.4, including they considers it as limited to comparisons of prediction skill indicators, themselves discussed.

> *There are numerous skill indicators and I gave the most popular and easy to interpret one (the MSS), but there are clearly others (I will reference the extensive discussion and of these in the Del Rio Amador, L., and Lovejoy, S., Using regional scaling for temperature forecasts with the Stochastic Seasonal to Interannual Prediction System (StocSIPS), Clim. Dyn. doi: 10.1007/s00382-021-05737-5, 2021a which explicitly uses the high frequency FEBE limit for state of the art forecasting).*

*In any case, it would have been nice to see specific comments explaining why the MSS is unacceptable (it might even help the GCM modellers that use it routinely)!*

I apologise for miscounting the referees: they were 5, not 4 and my numbering must be increased by one and you can claim a record. Referee 5 can be thanked for his constructive comments, especially on the role of Fourier techniques, but more generally contributed to this paper being modified.

*Unfortunately his comments were in fact not much help since they were so late that the transition to Fourier had already been done.  I will nevertheless acknowledge him in a final version.*

Best                                                                                                                regards,
Daniel

---

## Author Response (AR5)

Dec. 29, 2021

Dear Daniel;

Thanks for the acceptance.  I have made the promised modifications.  I'm disappointed that you are dissatisfied, but the comments were not specific enough to be more actionable.  The positive side was that the long review process gave me the time to improve numerous aspects and widen its scope.

As for the justification of the link with Budyko – Sellers, clearly the titles of the papers are not in themselves very relevant: the links with the FEBE are established by the contents.  In the end, the acceptance of this paper is timely since in the next week, the FEBE climate projections through 2100 (they use some of the results in this paper) will be published and there will be a press release.

Thanks and best wishes,

Shaun